# Regulation of Mitochondrial Metabolism by *Mfn1* Gene Encoding Mitofusin Affects Cellular Proliferation and Histone Modification

**DOI:** 10.3390/cells14131015

**Published:** 2025-07-02

**Authors:** Han Xu, Xiaoyu Zhao, Yuan Yun, Yuxin Gao, Chunjie Bo, Lishuang Song, Chunling Bai, Lei Yang, Guangpeng Li, Guanghua Su

**Affiliations:** 1State Key Laboratory of Reproductive Regulation and Breeding of Grassland Livestock (R2BGL), Inner Mongolia University, 24 Zhaojun Rd., Hohhot 010070, China; xuhan971225@163.com (H.X.); zhaoxiaoyu3233@163.com (X.Z.); souw261146@163.com (Y.Y.); m18535078392@163.com (Y.G.); shengwubcj@163.com (C.B.); xiaoshuang2000@126.com (L.S.); chunling1980_0@163.com (C.B.); mrknowall@126.com (L.Y.); 2College of Life Sciences, Inner Mongolia University, 24 Zhaojun Rd., Hohhot 010070, China

**Keywords:** mitochondrial dynamics, *Mfn1*, mitochondrial metabolism, cell proliferation, epigenetic modifications

## Abstract

Mitochondria maintain cellular homeostasis through the dynamic balance of fusion and fission, which relies on nuclear-encoded mitochondrial fusion proteins, mitofusins 1 and 2 (Mfn1, Mfn2). Changes in *Mfn1* and *Mfn2* expression significantly affect mitochondrial fusion and fission, thereby affecting cellular metabolism. This study investigated the effect of *Mfn1* expression on cell proliferation, apoptosis, and mitochondrial function by overexpressing *Mfn1* (in OE-Mfn1 cells) and silencing *Mfn1* using short hairpin RNA (shRNA) (in shMfn1 cells). Cell proliferation capacity, mitochondrial membrane potential, and mitochondrial ATP content were measured. To investigate the effects of Mfn1 on cellular metabolism and epigenetic modifications, the levels of metabolites α-KG, A-CoA, and SAM, as well as the levels of cellular methylation and acetylation, were detected by ELISA. Differentially expressed genes and metabolites were assessed by RNA-seq and LC-MS. This study demonstrates that alterations in *Mfn1* gene expression can significantly affect mitochondrial metabolism and cell proliferation and apoptosis. In addition, Mfn1 affects the expression of genes encoding enzymes that are responsible for histone methylation and acetylation, thereby regulating these modifications. These findings provide a theoretical basis for further elucidation of the mechanisms by which Mfn1 affects cell proliferation, regulates metabolites, and modulates chromatin epigenetic modification.

## 1. Introduction

The energy requirements and metabolism of the cell are in a state of dynamic balance, and changing any single component can disrupt this balance, thus affecting the function and state of the cell. As the power source of cells, mitochondria play a crucial role in regulating cellular metabolism and energy production through the dynamic balance of fusion and fission [1]. The biogenesis of mitochondrial dynamics depends on nuclear-encoded mitofusin 1 (Mfn1) and Mfn2. Mfn1 regulates mitochondrial fusion, allowing “unhealthy” mitochondria to resume their normal function, thereby affecting other mitochondrial function and maintaining cell homeostasis [2]. The function of mitochondria is closely related to their morphology. Enhanced mitochondrial fusion can promote the process of oxidative phosphorylation (OXPHOS), increase the production of adenosine triphosphate (ATP), and restore the function of mitochondrial DNA (mtDNA) [3]. Conversely, inhibiting mitochondrial fusion or promoting mitochondrial fission can lead to a surge in the levels of reactive oxygen species (ROS) within the mitochondria, changes in the permeability of the outer mitochondrial membrane (OMM), and the activation of apoptotic proteins such as caspases. These changes can lead to DNA damage and cell death [4].

Studies have found that metabolites produced by mitochondria, such as acetyl CoA (A-CoA), alpha-ketoglutarate (α-KG), nicotinamide adenine dinucleotide (NADH), and S-adenosyl methionine (SAM), often act as cofactors to regulate DNA and histone methylation in the promoter regions’ epigenetic modifications, thereby affecting gene expression [5,6,7,8]. In addition, intermediates such as citrate and succinate produced by the metabolism of the tricarboxylic acid cycle in mitochondria also participate in the regulation of epigenetic modification, thus regulating the epigenetic profile of cells [9]. These mitochondrial metabolites not only provide energy for cell growth, but also provide donor groups for chromatin modification. Changes in mitochondrial metabolism lead to alterations in expression levels in a variety of small metabolite molecules such as A-CoA, SAM, and α-KG, resulting in changes in metabolite content, thereby affecting the levels of chromatin modifications [10]. Epigenetic modifications, such as DNA and histone methylation and acetylation, which closely depend on the supply of mitochondrial metabolites, affect cell reprogramming efficiency [11].

In the process of somatic cell reprogramming, abnormal epigenetic modifications such as DNA and histone methylation and acetylation, which are closely related to mitochondrial metabolites, can affect reprogramming efficiency [12]. Our previous studies have shown that the overexpression of *Mfn1* can promote zygotic genome activation and significantly improve the developmental efficiency of the cloned embryos [13]. Currently, few studies have been reported on whether mitochondrial fusion affects the production of these metabolites that are associated with chromatin’s apparent modification, and thus the epigenetic modification of chromatin in cells. The aim of this study was to further investigate cell proliferation, gene expression, and metabolic regulation after the overexpression or silencing of *Mfn1* by RNA-seq and LC-MS analysis. This study tries to elucidate the mechanism by which the mitochondrial fusion gene *Mfn1* regulates cellular metabolites and chromatin epigenetic modifications.

## 2. Materials and Methods

### 2.1. Cell Culture

The bovine fibroblasts used in this study were derived from a Luxi cattle fetus. The cells were cultured in DMEM-F12 (Gibco, Waltham, MA, USA, A4192001) supplemented with 10% fetal bovine serum (Gibco, Waltham, MA, USA, 10099141) and 1% penicillin/streptomycin (Gibco, Waltham, MA, USA, 15140122) in a humidified incubator at 37 °C with 5% CO_2_. When the cell density reached 90%, passage was performed at a ratio of 1:3. These cells were used as control cells (CON) in further study.

### 2.2. Transfection

Cells with a good growth state and a logarithmic growth stage were inoculated in 10 cm dishes. When the cells grew to 80–90% density, the constructed *Mfn1* overexpression or silencing plasmids were added to transfect the cells by Lipofectamine 2000 (Invitrogen, Shanghai, China, 11668019). The overexpression and silencing vectors used in this study were both constructed in our laboratory previously. Among them, the backbone of the *Mfn1* overexpression vector is pmCherry2-N1 (Addgene, Watertown, MA, USA), and the backbone of the silencing vector is pGPU6-GFP-Neow. Transfected bovine fetus fibroblasts overexpressing the *Mfn1* gene (OE-Mfn1) and fibroblasts with silenced *Mfn1* (shMfn1) were used in further study, and incubated at 37 °C and 5% CO_2_ for 48 h. Transfection experiments were performed three times and transfection efficiency was statistically analyzed.

### 2.3. 3-(4,5-Dimethylthiazol-2-yl)-2,5-diphenyltetrazolium Bromide (MTT) Assay for Cell Proliferation Detection

Cell proliferation capacity was measured by the MTT Cell Proliferation and Cytotoxicity Detection Kit (Solarbio, Beijing, China, M1020) following the manufacturer’s instructions. Cells in the logarithmic growth phase were collected, seeded on a 96-well plate at a density of 1 × 10^4^ cells per 180 µL per well, and then cultured in an incubator at 37 °C and 5% CO_2_ for 24 h. The supernatant was carefully removed, 90 µL of fresh culture medium and 10 µL of MTT solution were added, and samples were incubated for 4 h. The supernatant was removed and 110 µL of Formazan dissolution solution was added to each well. Samples were gently shaken on a microplate for 10 min, and the absorbance was measured at 490 nm with a microplate reader.

### 2.4. 5-Ethynyl-2′-deoxyuridine (EdU) Detection of Cell Proliferation Ability

Cell proliferation was detected using Edu Imaging Kits (Cy3) (Apexbio, Houston, TX, USA, K1075) following the manufacturer’s instructions. Specifically, cells were seeded on 12-well plates pre-coated with cell-adhesive slides at a density of 2 × 10^5^ cells per 1 mL. After incubation for 4 h, 8 h, and 12 h, the pre-prepared EdU working solution was added and incubated at 37 °C and 5% CO_2_ for 2 h. The medium was discarded, and 500 µL of 4% paraformaldehyde was added to each well, and it was left to stand at room temperature for 15 min. The fixing solution was removed and the wells were washed three times with PBS containing 3% BSA for 5 min each time. A total of 500 µL of 0.05% Triton X-100 were added to each well and incubated at room temperature for 20 min for permeabilization, then the wells were washed three times with PBS containing 3% BSA for 5 min each time. A 500 µL Click reaction solution was added to each well and they were then incubated at room temperature for 30 min away from light. A 500 µL DAPI solution was added and samples were then incubated at room temperature for 3 min. The DAPI solution was removed, the anti-fade reagent was dropped onto the glass slides, and the slides were covered with coverslips. The images were captured with a Nikon confocal microscope.

### 2.5. Annexin V-FITC Detection of Cell Apoptosis Ability

According to the manufacturer’s instructions, the Annexin V-FITC Cell Apoptosis Detection Kit (Beyotime, C1062S) was used to detect the apoptosis ability of CON, OE-Mfn1, and shMfn1 cells. The specific procedures were as follows: 5 × 10^5^ resuspended cells of CON, OE-Mfn1, and shMfn1 were collected and centrifuged at 1000× *g* for 5 min. The supernatant was discarded. We added 195 μL of Annexin V-FITC binding solution to the cell samples and mixed them to gently resuspend the cells. Next, we added 5 μL of Annexin V-FITC to the cell samples and mixed well. Subsequently, 10 μL of propidium iodide staining solution was added to each sample and mixed. The samples were incubated at room temperature in the dark for 10 min and then centrifuged at 1000× *g* for 5 min. The cells were collected and resuspended in 50 μL of Annexin V-FITC binding solution. The cells were placed on a glass slide and observed under a fluorescence microscope.

### 2.6. RNA Extraction and Quantitative RT-PCR

Total mRNA was extracted from the transfected cells by RNAiso Plus. The total mRNA was reverse-transcribed into cDNA by HiScript^®^ Ⅱ Q RT SuperMix (+gDNA wiper) (Vazyme, Nanjing, China, R223-01). RT-qPCR was performed according to the instructions of the ChamQ Universal SYBR qPCR Master Mix (Vazyme, Nanjing, China, Q711-02) kit. The sequences of primers used for RT-qPCR in this study are shown in Appendix A.

### 2.7. Mitochondrial Membrane Potential Assay

Cells overexpressing *Mfn1*, with silenced *Mfn1* and CON cells, were stained with JC-1, a fluorescent dye that accumulates in mitochondria and indicates the membrane potential across the matrix membrane. Red fluorescence indicates high membrane potential and green fluorescence indicates low membrane potential. Mitochondrial activity was evaluated by the ratio of red/green fluorescence intensity. The cells were collected, washed once with PBS, and incubated in JC-1 staining working solution for 20 min. At the end of the incubation, the supernatant was aspirated and washed twice with JC-1 staining buffer (1×). The green JC-1 signal was measured at 485/535 nm, while the red signal was measured at 590/610 nm.

### 2.8. ATP Content Assay

Intracellular ATP levels were detected using the ATP Detection Kit (Beyotime Biotechnology, Shanghai, China). Cells were collected and disrupted by ultrasound at a ratio of 10^4^ cells per 1 mL of distilled water (1000:1). According to the number of samples, an appropriate amount of ATP assay working solution was added to a 12-well plate. After incubation at 37 °C for 30 min, a color developing reagent was added. After 20 min in a 37 °C water bath, the absorbance was measured at 700 nm.

### 2.9. Superoxide Dismutase (SOD) Activity Assay

According to the manufacturer’s instructions, the Superoxide Dismutase (SOD) Detection Kit (SOD-1-W, Keming Biotechnology Co., Ltd., Suzhou, China) was used to detect the intracellular SOD content. The specific operations were as follows: Cells were collected, and extraction buffer was added according to the ratio specified in the instructions. We sonicated the cell samples on ice. Subsequently, the samples were centrifuged at 8000× *g* for 15 min at 4 °C, and the supernatant was collected. According to the reagents provided in the kit, we added 100 μL of Reagent 2 to Reagent 1 and mixed them thoroughly to prepare the working solution. The pre-prepared working solution was added in the specified ratio. The samples were left to stand in the dark at room temperature for 30 min, and then the absorbance at 450 nm was measured using a microplate reader.

### 2.10. Catalase (CAT) Activity Assay

According to the manufacturer’s instructions, the CAT detection kit (CAT-1-W, Keming Biotechnology Co., Ltd., Suzhou) was used to detect the intracellular CAT content. Specifically, the operations were as follows: The cells were collected, and we added the extraction solution according to the ratio specified in the instructions. We disrupted the cells by sonication on ice. Then the samples were centrifuged at 8000× *g* for 10 min at 4 °C, and the supernatant was collected. The detection reagents were successively added per the instructions and mixed well. Specifically, for the experimental group, we added 150 μL of the sample and 90 μL of Reagent 1, respectively, to each group, mixed them well, and then incubated at 25 °C for 10 min. Subsequently, we added 300 μL of Reagent 2 and 795 μL of Reagent 3 to the samples. For the control group, 90 μL of Reagent 1, 300 μL of Reagent 2, and 795 μL of Reagent 3 were added, respectively, mixed thoroughly, and then 150 μL of the sample was added. The content of H_2_O_2_ in the samples was detected. The absorbance at 450 nm was measured using a microplate reader.

### 2.11. Nicotinamide Adenine Dinucleotide Hydrogen (NADH/NAD+) Content Assay

According to the manufacturer’s instructions, the NAD^+^/NADH Assay Kit (WST-8 method) was used to detect the ratio of NAD^+^/NADH in cells. CON, OE-Mfn1, and shMfn1 cells were collected and resuspended. We lysed the cells at a ratio of adding 200 μL of extraction solution per 1 × 10^6^ cells. Standards were prepared according to the kit’s instructions, and a standard curve was plotted. Pipette 100 μL of the sample to be tested into a centrifuge tube and heat it in a 60 °C water bath for 30 min. We pipetted 20 μL of the sample to be tested into a 96-well plate, with 3 replicates set for each group. Blank control wells, standard wells, and sample wells were set up, respectively, per the instructions. Add ethanol dehydrogenase working solution to each well and mix thoroughly. Incubate the plate in the dark at 37 °C for 10 min. Add 10 μL of chromogenic solution to each well, mix, and then incubate in the dark at 37 °C for 10–20 min. Measure the absorbance at 450 nm using a microplate reader.

### 2.12. Western Blot Analysis

Cells were collected and washed with PBS. Each sample of 1 × 10^7^ cells was treated with 100 µL of protein lysis buffer (990 µL RIPA + 10 µL PMSF) and kept in ice for 30 min. The supernatant was collected after centrifugation at 12,000 rpm for 30 min at 4 °C. The Pierce BCA protein assay kit (Thermo Fisher Scientific, Waltham, MA, USA, 23225) was used to determine the protein concentration. The protein supernatant was mixed with 5× protein loading buffer containing DTT at a ratio of 4:1, and heated in boiling water for 10 min to fully denature the proteins. According to the sodium dodecyl sulfate–polyacrylamide gel electrophoresis (SDS-PAGE) gel preparation kit instruction, a gel concentration of 12% was selected for electrophoresis. After electrophoresis, the proteins were transferred to the membrane. Then the polyvinylidene fluoride (PVDF) membrane was blocked with 5% skim milk (2.5 g of skim milk powder + 50 mL of TBST) at room temperature for 1 h. Anti-Mfn1 antibody (Santa Cruz Biotechnology, Santa Cruz, CA, USA, SC-166644, diluted 1:500, Mfn1 (D-10) is a mouse monoclonal antibody raised against amino acids 10–74 mapping within an N-terminal cytoplasmic domain of Mfn1 of human origin), rabbit anti-α-tubulin antibody (Proteintech, Chicago, IL, USA, 11224-1-AP, diluted 1:1000, raised against full length α-tubulin of human origin), rabbit anti-Histone H3 (tri methyl K9) antibody (Abcam, Cambridge, UK, ab8898, diluted 1:1000, raised against the human Histone H3 (tri-methyl K9) polypeptide), mouse anti-Histone H3 (tri methyl K27) antibody (Abcam, Cambridge, UK, ab6002, diluted 1:1000, raised against the human Histone H3 (tri-methyl K27) polypeptide), rabbit anti-Histone H3 (acetyl K9) antibody (Abcam, Cambridge, UK, ab32129, diluted 1:1000, a recombinant monoclonal antibody), rabbit anti-acetyl lysine antibody (Abcam, Cambridge, UK, 190479, diluted 1:1000, a recombinant monoclonal antibody), and rabbit anti-Histone H3 antibody (Proteintech, Chicago, Illinois, USA, 17168-1-AP, diluted 1:1000, raised against the human Histone H3 polypeptide) were diluted with primary antibody dilution buffer and incubated at 4 °C overnight. After the removal of the primary antibodies, the membranes were washed three times with 1× TBST washing buffer for 10 min each time. The membranes were then placed in 5 mL of 1× TBST-diluted secondary antibody, horseradish peroxidase (HRP)-labeled affinipure goat anti-mouse IgG (H + L) (Proteintech, Chicago, Illinois, USA, SA000011, 1:5000, purified from antisera by immunoaffinity chromatography using antigens coupled to agarose beads), and HRP-conjugated affinipure goat anti-rabbit IgG (H + L) (Proteintech, Chicago, Illinois, USA, SA00001-2, diluted 1:1000, purified from antisera by immunoaffinity chromatography using antigens coupled to agarose beads), and incubated at room temperature with shaking for 2 h. After the removal of the secondary antibody, the membrane was washed three times with 1× TBST washing buffer on a shaking bed for 10 min each. Then the developing solutions A and B (1:1) were added and left to stand in a dark box for 5 min. Then, the membranes were wiped dry with filter paper and exposed. Samples were analyzed by ImageJ software (Version 1.8.0) to analyze the images and obtain the grayscale values. Each experiment was repeated three times, and the significance analysis is performed using GraphPad Prism (Version 8) software.

### 2.13. Determination of Enzyme-Linked Immunosorbent Assay (ELISA) for α-KG, A-CoA, and SAM Content

According to the manufacturer’s instructions, the levels of α-KG, SAM, and A-CoA in cells were measured using the Bovine α-KG ELISA Kit (Enzyme Immunoassay, Jiangsu, China, MM-5107801), Bovine SAM ELISA Kit (Enzyme Immunoassay, Jiangsu, China, MM-5108101), and Bovine A-CoA ELISA Kit (Enzyme Immunoassay, Jiangsu, China, MM-5096101), respectively. Briefly, columns were equilibrated at room temperature for 20 min. First, standard wells and sample wells (for CON, OE—Mfn1, and shMfn1 cells) were established, with three replicates set for each sample. Fifty microliters of standards at different concentrations were added to the standard wells. Subsequently, 10 µL of the samples to be tested were added to the sample wells, followed by the addition of 40 µL of the sample diluent. The blank wells were left empty. Except for the blank wells, 100 µL of HRP-conjugated detection antibody was added to each well, and the wells were incubated at 37 °C for 60 min. The liquid was aspirated, and the plate was washed 5 times with the wash buffer, with each well completely filled for 1 min each time. Fifty microliters of substrate A and 50 µL of substrate B were added to each well, and the wells were incubated at 37 °C in the dark for 15 min. Finally, the stop solution was added to each well, and the optical density (OD) was measured at a wavelength of 450 nm.

### 2.14. RNA Sequencing Method

Cells were collected from CON, OE-Mfn1, and shMfn1 cell cultures. Total RNA was extracted using the RNA-easy Isolation Reagent (Vazyme, Nanjing, China, R701) according to the manufacturer’s instructions, and the RNA-seq libraries were constructed. The Illumina HiSeq2000TM platform was used for sequencing. Quality control of raw RNA-seq data was performed using Fastx_toolkit (Version 0.014) and Fastp (0.19.5) software. The Reads were mapped to the genome using Bowtie alignment software (Version 2). Principal Component Analysis (PCA) was performed on RNA-seq data using the sklearn package in Python (Version 1.4.1) for multivariate dimensionality reduction analysis to intuitively express the similarities and differences in gene expression among samples. DESeq2 software (Version 1.38.0) was used for differential analysis, and the standard for identifying differentially expressed genes was set as |log2FC| ≥ 1 and *p*-value < 0.05. The Gene Ontology (GO) and Kyoto Encyclopedia of Genes and Genomes (KEGG) analyses were conducted using the DAVID database (https://david.ncifcrf.gov/).

### 2.15. Metabolome Analysis

Total metabolites were extracted from CON, OE-Mfn1, and shMfn1 cells. The metabolites were separated using chromatography columns and analyzed using mass spectrometry in both positive and negative ion modes. The resulting raw data were preprocessed, including filtering, imputation, normalization, and log-transformation. Differentially synthesized metabolites were identified based on univariate statistics (*t*-tests) combined with multivariate statistics (OPLS-DA/PLS-DA) and fold change (FC) values, with the criteria as *p* < 0.05, VIP > 1, and FC < 1 or FC > 1. KEGG and GO analyses were performed to identify signaling pathways and biological processes which can affect the synthesis of differentially produced metabolites.

### 2.16. Statistical Analysis

All values were expressed as mean ± SEM. GraphPad Prism 8 software was used to analyze the significance of differences (one-way ANOVA was used for multiple group comparisons, and Student’s *t* test was used for two-group comparisons), and *p* < 0.05 was considered to indicate a significant difference (* *p* < 0.05, ** *p* < 0.01, *** *p* < 0.001). The experiment was repeated three times for both technical and biological replicates.

## 3. Results

### 3.1. Changes in Mfn1 Gene Expression Affect Cell Proliferation and Apoptotic Capacity

Based on current research, it is known that the *Mfn1* gene plays an important role in mitochondrial function. However, there are few studies on the effects of *Mfn1* overexpression and silencing on cell fate. To investigate the role of OE-Mfn1 and short hairpin RNA (shMfn1) in bovine fetal fibroblasts, we transfected the cells with OE-Mfn1 and shMfn1 vectors (Appendix A). The expression of *Mfn1* and the levels of Mfn1 in OE-Mfn1 and shMfn1 fibroblasts were detected by RT-qPCR and Western blot. The results showed that the expression of *Mfn1* gene was significantly increased in OE-Mfn1 cells and was significantly decreased in shMfn1 cells, when compared with CON cells (Appendix A). To explore the effect of the *Mfn1* gene on cell functions, we used EdU, MTT, and Annexin V-FITC to detect and analyze the proliferation, apoptosis, and viability of CON, OE-Mfn1, and shMfn1 cells. The EdU and MTT cell proliferation assays showed that OE-Mfn1 significantly increased the cell proliferation rate compared to the control cells, while shMfn1 significantly decreased the cell proliferation capacity (Figure 1A,B). Analysis of apoptosis results showed that OE-Mfn1 significantly reduced the number of apoptotic cells, while shMfn1 significantly increased the number of apoptotic cells (Figure 1C). To further investigate the effect of the *Mfn1* gene on cell proliferation and apoptosis, we analyzed mRNA expression levels of the following genes: proliferating cell nuclear antigen (*Pcna*), cyclin dependent kinase 1 (*Cdk1*), *Cdk2*, Marker of proliferation Ki-67 (*Mk167*), proliferation inhibitors Cyclin-dependent kinase inhibitor 1a (*Cdkn1a*), *Cdkn2a*, pro-apoptotic genes Bcl-xl/Bcl-2 associated death promoter (*Bad*), Cytochrome C (*Cycs*), Caspase3 (*Casp3*), *Casp9*, anti-apoptotic genes B-cell lymphoma-2 (*Bcl2*), and the X-linked inhibitor of apoptosis (*Xiap*). The results showed that, in OE-Mfn1 cells, the mRNA expression levels of *Pcna*, *Cdk1*, *Cdk2*, *Mk167*, *Cdkn2a*, *Bcl2,* and *Xiap* were significantly increased, while the expression of *Cdkn1a* and the pro-apoptotic gene *Bad* were significantly decreased (Figure 1D–G). In shMfn1 cells, the expressions of *Pcna*, *Cdk1*, and *Bcl2* were significantly decreased, while the expressions of *Cdkn1a*, *Bad*, *Cycs*, *Casp3*, and *Casp9* were significantly increased. The expression of the anti-apoptotic gene *Xiap* did not show a significant difference (Figure 1D–G).

### 3.2. Mfn1 Gene Affects Mitochondrial Function

To investigate the effect of a change in the expression of *the Mfn1* gene on mitochondrial function, we measured mitochondrial membrane potential, energy production and antioxidant capacity in OE-Mfn1- and shMfn1-transfected cells. JC-1 mitochondrial membrane potential detection showed that the overexpression of *Mfn1* significantly increased the mitochondrial membrane potential (*p* < 0.001), while the downregulation of *Mfn1* significantly decreased the mitochondrial membrane potential (*p* < 0.05) (Figure 2A). The levels of ATP and NADH in OE-Mfn1 cells were significantly increased (*p* < 0.01), while there was no significant difference in shMfn1 cells (Figure 2B,C). The ROS content in OE-Mfn1 cells decreased, while the ROS production in shMfn1 cells increased, when compared with CON cells (Figure 2D). The activities of CAT and SOD in cells with altered *Mfn1* gene expression significantly increased (*p* < 0.001), while shMfn1 significantly decreased the activities of CAT and SOD (*p* < 0.001) (Figure 2E,F). The above results indicate that after the overexpression of the *Mfn1* gene, mitochondrial activity increases, mitochondrial energy production is enhanced, and the antioxidant capacity of cells is strengthened. After the silencing of the *Mfn1* gene, mitochondrial activity decreases, and the antioxidant capacity of cells is weakened.

### 3.3. Overexpression and Downregulation of Mfn1 Gene Affects Expression of Many Genes

To find how changes in the expression of *Mfn1* affect the expression of nuclear genes, the RNA-seq analysis was performed in OE-Mfn1, shMfn1, and CON cells. RNA-seq analysis of OE-Mfn1 and shMfn1 cells showed that the transcriptional correlation coefficient was consistent with PCA data. The samples within each group had good clustering and high correlation, while the samples between groups were scattered, indicating high similarity between biological replicates (Figure 3A,B). Compared with CON cells, 1679 differentially expressed genes were identified in OE-Mfn1 cells, including 587 upregulated genes and 1092 downregulated genes (*p*-Value < 0.05 and |Log2 FC| ≥ 1) (Figure 3C). In shMfn1 cells, 1433 differentially expressed genes were identified, including 502 upregulated and 931 downregulated genes (Figure 3D). The KEGG results showed that the differentially expressed genes in both OE-Mfn1 and shMfn1 cells mainly enriched genes involved in cell growth and apoptosis, lipid metabolism, energy metabolism, and glycan biosynthesis and metabolism biological processes (Appendix A). The GO database analysis indicated that differentially expressed genes are mainly enriched in genes involved in cell proliferation regulation, reactive oxygen metabolism, glucose metabolism, lipid metabolism, ATP synthesis, and electron transfer biological processes (Figure 3E and Appendix A).

*Mfn1* is a key protein in the regulation of mitochondrial fusion, and plays an important role in cell anabolism, energy metabolism, and the maintenance of mitochondrial homeostasis [14]. Therefore, we performed KEGG pathway enrichment analysis for differentially expressed genes involved in metabolic signaling pathways. The results showed that, compared with CON cells, the differentially expressed genes in OE-Mfn1 cells were mainly enriched in genes involved in cAMP, PI3K-Akt, and Wnt signaling pathways, while the differentially expressed genes in shMfn1 cells were mainly enriched in cAMP, PI3K-Akt, Wnt, and PPAR signaling pathways (Figure 3F and Appendix A).

### 3.4. Metabolomic Analysis of OE-Mfn1 and shMfn1 Cells

To investigate the effect of the *Mfn1* gene on cellular metabolic processes and metabolite synthesis, LC-MS metabolites were measured and analyzed in CON, OE-Mfn1, and shMfn1 cells. Metabolic correlation coefficient and PCA results showed little difference and high correlation between metabolites in CON, OE-Mfn1, and shMfn1 cells, and a significant separation of metabolites between cells with large differences (Figure 4A,B). Using the criteria of *p*-Value < 0.05, VIP-PLS-DA > 1, and fold change ≥ 1, compared to CON cells, 249 metabolites show different levels on OE-Mfn1 cells, including 177 that were upregulated and 72 that were downregulated. In shMfn1 cells, 223 differentially synthesized levels of metabolites were identified with 87 upregulated and 136 downregulated (Figure 4C,D). GO data analysis showed that the differentially synthesized compounds were mainly enriched in metabolism, organismal systems, genetic information processing, and cellular processes. The metabolic pathways were mainly enriched in energy metabolism, amino acid metabolism, and glycan synthesis. The cellular processes involved are mainly enriched in cell growth and apoptosis pathways (Figure 4E and Appendix A). KEGG pathway enrichment analysis showed that the differentially synthesized metabolites are mainly enriched in glucose metabolism, pantothenate and CoA biosynthesis, the pentose phosphate pathway, glutathione metabolism, niacin metabolism, folate synthesis, and lipid metabolism pathways (Figure 4F and Appendix A).

The loss of Mfn1 affects the function of mitochondria [15]. Mitochondrial energy metabolism is a key part of cellular metabolism and provides energy for cell growth [16]. Therefore, we analyzed the content of key mitochondrial metabolites α-KG, A-CoA, and SAM in CON, OE-Mfn1, and shMfn1 cells. ELISA results showed (Figure 4G–I) that α-KG and A-CoA levels in OE-Mfn1 cells were significantly increased (*p* < 0.001), and SAM was significantly decreased (*p* < 0.05). In shMfn1 cells, α-KG was significantly decreased (*p* < 0.001), while A-CoA was significantly increased (*p* < 0.01), and the SAM level was not significantly different (Figure 4G–I).

### 3.5. Mfn1 Gene Affects Protein Methylation and Acetylation

Increasing evidence has indicated that mitochondrial metabolites are not only involved in cellular metabolism, but also regulate gene expression through epigenetic modifications as a donor group [16]. α-KG acts as a cofactor for dioxygenases such as Ten eleven translocation 1–3 (TET1-3) and KDMs containing the Jumonji C domain, SAM acts as a cofactor for the methylation of proteins, and acetyl-CoA can be used as a cofactor for histone acetylation [17,18,19,20]. The results mentioned above indicate that changes in *Mfn1* gene expression can affect the synthesis and levels of mitochondrial metabolites such as α-KG, A-CoA, and SAM. To further explore the relationship between Mfn1 and methylation and acetylation modifications, a Heatmap cluster analysis of differentially expressed genes associated with methylation and acetylation was performed. The results showed that, compared with CON cells, the demethylase genes *Kdm3a*, *Kdm4a*, and *Tet3* were upregulated and the methyltransferase genes Euchromatic histone lysine N-methyltransferase 2 (*Ehmt3*), Suppressor of variegation 3–9 homolog 1 (*Suv39h1*), and Set domain containing 1a (*Setd1a*) were downregulated, while the Sirtuins (*Sirts*) and Hdac deacetylase gene family were downregulated and the acetyltransferase genes Lysine acetyltransferase 2b (*Kat2b*) and Glucosaminyl (N-acetyl) transferase 4 (*Gcnt4*) were upregulated in OE-Mfn1 cells. In shMfn1 cells, the methyltransferase genes *Ehmt2*, *Suv39h1*, and *Setd1a* were upregulated, the demethylase genes *Kdm3a*, *Kdm4a*, and *Tet3* were downregulated, the Sirts deacetylase gene family was upregulated, and the expression of the *Gcnt4* acetyltransferase gene was increased (Figure 5A,B). At the cellular level, in OE-Mfn1 cells, compared with CON cells, the genes of histone demethylases *Kdm1a*, *Kdm1b*, *Kdm6a*, and *Kdm6b*, and the genes of histone acetyltransferases *Kat2a*, *Kat2b*, and *Kat7* were significantly upregulated, while the genes of histone deacetylases *Sirt1*, *Sirt4*, *Sirt6*, and *Sirt7* were significantly downregulated. In shMfn1 cells, the genes of histone acetyltransferases *Kat2a*, *Kat2b*, and *Kat7* and the gene of histone demethylase *Kdm6a* were significantly downregulated, while the gene of histone deacetylase *Sirt2* was significantly upregulated (Figure 5C–E).

To explore the effect of the *Mfn1* gene on histone methylation and acetylation, the histone modifications in OE-Mfn1 and shMfn1 cells were assessed by Western Blot analysis. The results showed that the levels of H3K9me3 and H3K27me3 modifications were significantly decreased in OE-Mfn1 cells, while the levels of H3K9me3 and H3K27me3 methylation modifications were increased in shMfn1 cells (Figure 5F,G). The pan-acetylation and H3K9 acetylation in OE-Mfn1 cells were increased, while both of them were significantly decreased in shMfn1 cells (Figure 5H,I).

## 4. Discussion

Mitochondrial membrane fusion is mainly regulated by the *Mfn1* gene and the Mfn1 protein. Mutations in the *Mfn1* gene lead to impaired mitochondrial fusion, resulting in mitochondrial dysfunction, improving autophagy, and increasing cell apoptosis [21]. Low expression of *Mfn1* not only reduces the efficiency of mitochondrial fusion, but also affects mitochondrial motility and changes in mitochondrial morphology [22]. In mice, the knockout of *Mfn1* resulted in spherical and short tubules of mitochondrial morphologies. Recovery of *Mfn1* expression can reshape the fragmented mitochondria to their original normal shape. Although the overexpression of *Mfn2* can also partially recover damaged mitochondria, this recovery effect is far less significant than that of *Mfn1* [23]. Our previous studies have found that low expression of *Mfn*1 is one of the reasons for the failure of interspecific embryonic genome activation (EGA). High expression of *Mfn1* can promote interspecific embryonic development and induce EGA-related gene expression by influencing mitochondrial function [13]. In the present study, it was found that the alteration in *Mfn1* expression significantly affected cell proliferation and apoptosis. This may be related to the expression changes in cell proliferation marker genes *Pcna*, *Cdk1*, *Cdk2*, and *Mk167*, anti-apoptosis genes *Bcl2* and *Xiap*, proliferation inhibition gene *Cdkn1a*, and pro-apoptosis genes *Bad*, *Cycs*, *Casp3*, and *Casp9*. In addition, this study found that changes in *Mfn1* gene expression affected cell apoptosis regulatory pathways such as PI3K-Akt, cAMP, and MAPK. The cAMP signaling pathway regulates oxidative metabolism and mitochondria-initiated cell apoptosis by influencing mitochondrial protein phosphorylation [24]. The PI3K-Akt signaling pathway is involved in the regulation of glucose metabolism, cell proliferation, and apoptosis [25]. The MAPK signaling pathway is involved in a variety of physiological processes and disease, and affects gene transcription, cell proliferation and apoptosis, and oxidative stress by activating protein kinase cascades [26]. The wnt signaling pathway mediates mitochondrial stress responses [27].

The mitochondrial morphology is one of the key factors that determine mitochondrial function [28]. During periods of high energy demand, mitochondrial fusion increases and the mitochondrial network expands to meet the energy needs of the cell, thereby maintaining cell homeostasis. In contrast, when cells are in a state of excess energy, abnormal mitochondrial fusion leads to mitochondrial fission [29]. Studies have shown that promoting mitochondrial fusion can improve cellular energy metabolism and prevent the overproduction of ROS [30]. Accelerated mitochondrial fission weakens the mitochondrial antioxidant defense system, causing mitochondrial dysfunction, ROS accumulation, affecting mitochondrial membrane potential, and leading to a decline in mitochondrial activity [31]. Ying et al. found that the overproduction of *Mfn1* increased mitochondrial length and connectivity, and cells with elongated mitochondria maintained cellular ATP levels by increasing ATP synthase activity and the number of cristae [32]. Deficiencies of *Mfn1* or *Mfn2* disrupt mitochondrial fusion, resulting in the loss of mitochondrial membrane potential, reduced oxidative phosphorylation, and decreased ATP production [33]. Similarly, results of this study indicate that changes in *Mfn1* gene expression affect mitochondrial membrane potential, ATP and NADH production, as well as cellular ROS content and CAT and SOD activity. These results confirm that the overexpression of the *Mfn1* gene increases mitochondrial activity, enhances energy production and strengthens cellular antioxidant capacity, while the knockdown of *Mfn1* decreases mitochondrial activity and weakens the cellular antioxidant capacity.

Mitochondria are the sites of various biological processes such as β-oxidation, the tricarboxylic acid (TCA) cycle, and oxidative phosphorylation. The present data of RNA-seq indicate that the differentially expressed genes regulated by *Mfn1* expression are mainly involved in cellular energy metabolism, lipid metabolism, amino acid metabolism, and glycan synthesis. Metabolomics data showed that the differentially produced metabolites were primarily enriched in glucose metabolism, AMPK signaling pathways, pantothenate and CoA biosynthesis, glutathione metabolism, and folate synthesis. Excessive accumulation of ROS can damage mitochondria, thereby directly affecting the synthesis of SAM, and thus DNA methylation [34]. Gao et al. reported that the addition of α-KG and its derivatives could extend the lifespan of *Caenorhabditis elegans* by inhibiting ATP synthase activity [16]. By activating mitochondrial enzymes, dichloroacetate (DCA) converts pyruvate into A-CoA and increases the level of cellular NAD+ [35], subsequently activating the mitochondrial unfolded protein response, reducing the mismatch rate between mtDNA-encoded proteins and nuclear-encoded proteins, restoring communication between mitochondria and the nucleus, and extending the lifespan of *C. elegans* [36]. In this study, we found that OE-Mfn1 significantly increased the levels of α-KG and A-CoA, while decreasing SAM levels. shMfn1 significantly decreased α-KG and increased A-CoA expression, but did not affect SAM content. Changes in α-KG, A-CoA, and SAM may be the cause of increased apoptosis.

Studies have evidenced that mitochondrial metabolites such as ROS, A-CoA, α-KG, SAM, and NAD+ are involved in the regulation of chromatin epigenetic modifications [5]. In rat liver, elevated levels of adenosine and homocysteine inhibit the activity of SAM-dependent methyltransferases [37], leading to a decrease in global DNA methylation levels. Administration of metformin in rats increases SAM levels, which significantly increases DNA methylation levels [38]. The ATP-citrate lyase (ACL) enzyme can affect DNA methylation levels through an A-CoA-dependent mechanism [39]. In human lung cells, increased H_2_O_2_ and ROS affect histone methylation levels, resulting in increased H3K4me3 levels [40]. In senescent cells, elevated α-KG levels affect histone demethylases, thereby modulating the chromatin epigenetic state [41]. In aged mouse bone marrow stromal cells (BMSCs), elevated α-KG reduces H3K9me3 and H3K27me3 modification levels [42]. Galdieri and Vancura reported that the inhibition of A-CoA carboxylase can induce global histone acetylation, and the dysregulation of A-CoA homeostasis may affect chromatin epigenetics [43]. In mouse muscle stem cells, lowering the NAD+ concentration significantly increases H4K16 acetylation [44]. In mouse embryonic stem cells, A-CoA promotes histone acetylation and maintains cell pluripotency [45]. Our study found that genes related to histone methylation and acetylation showed significant expression differences in CON, OE-Mfn1, and shMfn1 cells. Compared to the CON cells, the expression of *KDMs* in OE-Mfn1 cells was significantly increased, while the expression levels of H3K9me3 and H3K27me3 were significantly decreased. In contrast, in shMfn1 cells, the expression of *KDM6B* histone demethylase was significantly decreased, and the levels of H3K9me3 and H3K27me3 were increased. Similarly, compared with CON cells, the OE-Mfn1 cells exhibited a significantly higher expression of histone acetyltransferases and a lower expression of histone deacetylases, resulting in significantly elevated pan-acetylation levels and increased H3K9 acetylation. The shMfn1 cells showed a significantly lower expression of histone acetyltransferases and a higher expression of histone deacetylases, resulting in significantly reduced pan-acetylation levels and decreased H3K9 acetylation. In summary, the *Mfn1* gene indirectly regulates the expression of histone methyltransferases and histone acetyltransferase genes by influencing mitochondrial metabolism, thereby regulating histone methylation and histone acetylation. This study has revealed the functions and mechanisms of action of the *Mfn1* gene in bovine fetal fibroblasts from multiple aspects, providing a research foundation and new research directions for fields such as cell biology, mitochondrial research, metabolic regulation, and epigenetics.

## 5. Study Limitations

In this study, some of the data are solely sourced from bovine fetal fibroblasts. Therefore, the generalizability of the research results may be somewhat limited, and the conclusions may not be directly applicable to other types of cells or organisms. In addition, in the control setting of the transfection experiments, we used bovine fetal fibroblasts that were not transfected with any vectors as control cells, rather than using cells transfected with the same plasmid containing a scrambled version of *Mfn1* for overexpression and silencing. Although non-transfected cells can largely present the natural state of various indicators and provide a relatively pure baseline for the study, we conducted comprehensive and in-depth comparative analyses between the non-transfected group and the experimental group through multiple methods such as transcriptome sequencing and metabolomics. Moreover, the data met the requirements for relevant statistical analyses in terms of sample size and variance, endowing the research conclusions with a certain degree of reliability and scientific value. However, this control setting method may have limitations. Since cells transfected with the same plasmid containing a scrambled version of *Mfn1* were not used as controls, it may not be possible to completely rule out the non-specific effects on cells caused by the plasmid transfection operation itself. This, to some extent, may affect the precise evaluation of the overexpression or silencing effects of *Mfn1*. In future studies, we will fully consider the suggestions of the reviewers and use cells transfected with the same plasmid containing a scrambled version of *Mfn1* as controls. This will enable us to more accurately isolate the cell biological effects caused by changes in *Mfn1* expression and improve the accuracy and reliability of the research results. Meanwhile, we will also continuously monitor the research trends in this field and constantly optimize the experimental design and methods.

## Figures and Tables

**Figure 1 cells-14-01015-f001:**
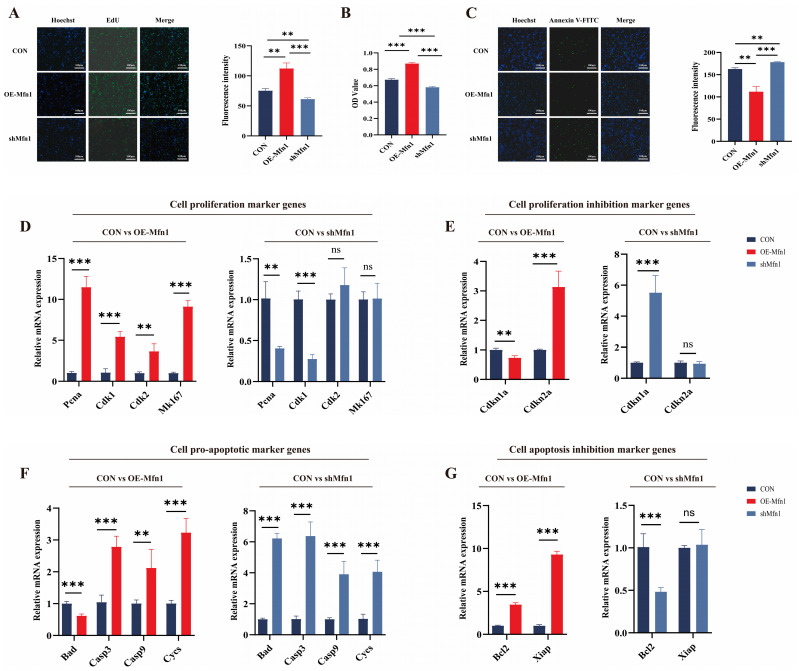
Cell proliferation and apoptosis ability is changed upon the overproduction or downregulation of the *Mfn1* gene. (**A**) Changes in the expression of the *Mfn1* gene can affect the cell proliferation ability. (**B**) MTT assay of cell proliferation abilities in CON, OE-Mfn1, and shMfn1 cells. (**C**) Changes in the expression of the *Mfn1* gene can have an impact on cell apoptosis. (**D**) RT-qPCR detection of mRNA expression levels of cell proliferation marker genes *Pcna*, *Cdk1*, *Cdk2*, and *Mk167* in CON, OE-Mfn1, and shMfn1 cells. (**E**) RT-qPCR detection of mRNA expression levels of proliferation inhibitory genes *Cdkn1a* and *Cdkn2a* in CON, OE-Mfn1, and shMfn1 cells. (**F**) RT-qPCR detection of mRNA expression levels of pro-apoptotic genes *Bad*, *Casp3*, *Casp9*, and *Cycs* in CON, OE-Mfn1, and shMfn1 cells. (**G**) RT-qPCR detection of mRNA expression levels of apoptosis inhibitory genes *Bcl2* and *Xiap* in CON, OE-Mfn1, and shMfn1 cells. ** *p* < 0.01; *** *p* < 0.001; ns, no statistically significant difference.

**Figure 2 cells-14-01015-f002:**
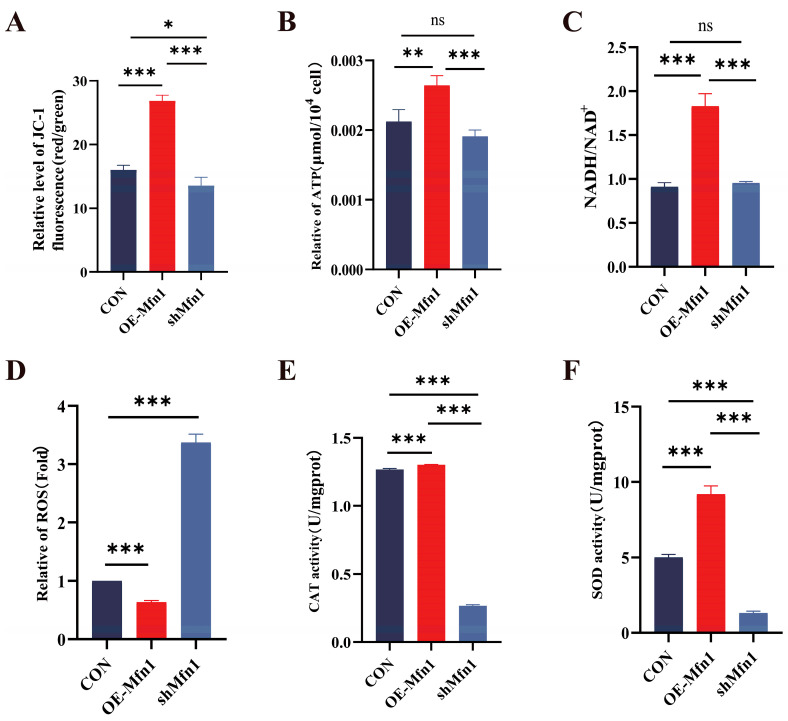
Expression of *Mfn1* gene affects mitochondrial functions. (**A**) Detection of mitochondrial membrane potential in CON, OE-Mfn1, and shMfn1 cells by JC-1. (**B**) Measurement of ATP content in CON, OE-Mfn1, and shMfn1 cells. (**C**) Detection of NADH/NAD+ content in CON, OE-Mfn1, and shMfn1 cells. (**D**) Measurement of ROS content in CON, OE-Mfn1, and shMfn1 cells. (**E**) Detection of CAT activity in CON, OE-Mfn1, and shMfn1 cells. (**F**) Measurement of SOD activity in CON, OE-Mfn1, and shMfn1 cells. * *p* < 0.05; ** *p* < 0.01; *** *p* < 0.001; ns, no statistically significant difference.

**Figure 3 cells-14-01015-f003:**
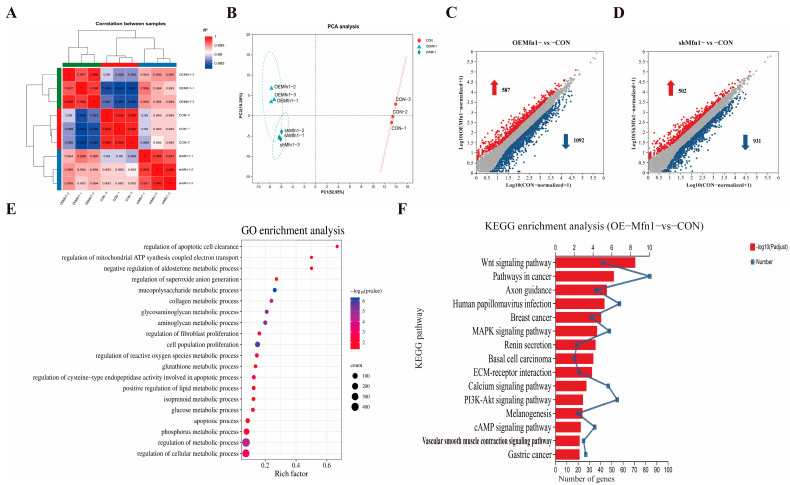
RNA-seq analysis of CON, OE-Mfn1, and shMfn1 cells. (**A**) Correlation coefficient analysis of CON, OE-Mfn1, and shMfn1 cells. (**B**) PCA clustering analysis of CON, OE-Mfn1, and shMfn1 cells. (**C**) Differential gene statistics in CON and OE-Mfn1 cells. (**D**) Differential gene statistics in CON and shMfn1 cells. (**E**) GO functional enrichment analysis of differentially expressed genes between CON and OE-Mfn1 cells. (**F**) KEGG pathway enrichment analysis of differentially expressed genes between CON and OE-Mfn1 cells.

**Figure 4 cells-14-01015-f004:**
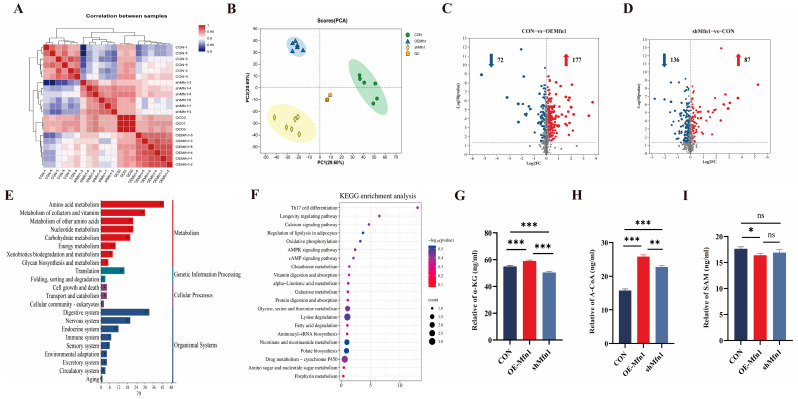
CON, OE-Mfn1, and shMfn1 cells for metabolomics analysis and key metabolite content detection. (**A**) Correlation coefficient analysis of metabolites in CON, OE-Mfn1, and shMfn1 cells, respectively. (**B**) PCA clustering of metbolomics in CON, OE-Mfn1, and shMfn1 cells. (**C**) Differential metabolite statistics in CON and OE-Mfn1 cells. (**D**) Differential metabolite statistics in CON and shMfn1 cells. (**E**) KEGG functional pathway enrichment analysis of differential metabolites between CON and OE-Mfn1 cells. The red bars represent the enriched signal pathways related to metabolism, the green bars represent the enriched signal pathways related to genetic information processing, the purple bars represent the enriched signal pathways related to cellular processes, and the blue bars represent the enriched signal pathways related to organismal systems. (**F**) KEGG pathway enrichment analysis of differential metabolites between CON and OE-Mfn1. (**G**) Detection of α-KG content in CON, OE-Mfn1, and shMfn1 cells. (**H**) Detection of A-CoA content in CON, OE-Mfn1, and shMfn1 cells. (**I**) Detection of SAM content in CON, OE-Mfn1, and shMfn1 cells. * *p* < 0.05; ** *p* < 0.01; *** *p* < 0.001; ns, no statistically significant difference.

**Figure 5 cells-14-01015-f005:**
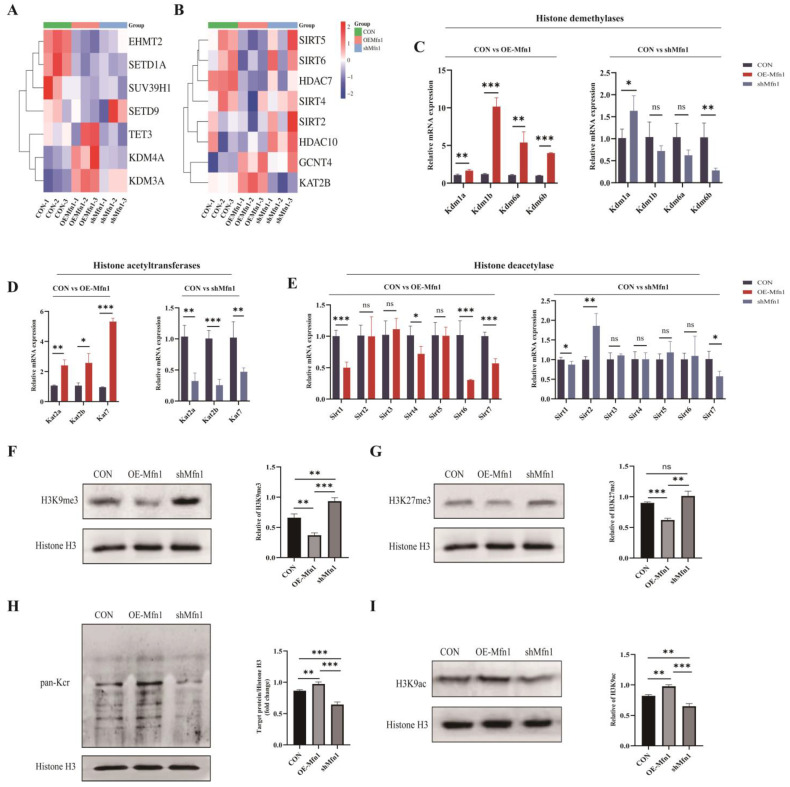
Detection of expression levels of methyltransferases and acetyltransferases related to methylation and acetylation in CON, OE-Mfn1, and shMfn1 cells, as well as histone methylation and acetylation modification levels. (**A**) Cluster analysis of methylation-related genes in CON, OE-Mfn1, and shMfn1 cells. (**B**) Cluster analysis of acetylation-related genes in CON, OE-Mfn1, and shMfn1 cells. (**C**) RT-qPCR detection of mRNA expression levels of histone demethylases Kdm1a, Kdm1b, Kdm6a, and Kdm6b in CON, OE-Mfn1, and shMfn1 cells. (**D**) RT-qPCR detection of mRNA expression levels of histone acetyltransferases Kat2a, Kat2b, and Kat7 in CON, OE-Mfn1, and shMfn1 cells. (**E**) RT-qPCR detection of mRNA expression levels of histone deacetylases Sirt1-6 in CON, OE-Mfn1, and shMfn1 cells. (**F**) Western blot detection of H3K9me3 modification levels in CON, OE-Mfn1, and shMfn1 cells and the grayscale analysis thereof. (**G**) Western blot detection of H3K27me3 modification levels in CON, OE-Mfn1, and shMfn1 cells and the grayscale analysis thereof. (**H**) Western blot detection of pan-Kcr modification levels in CON, OE-Mfn1, and shMfn1 cells and the grayscale analysis thereof. (**I**) Western blot detection of H3K9ac modification levels in CON, OE-Mfn1, and shMfn1 cells and the grayscale analysis thereof. * *p* < 0.05; ** *p* < 0.01; *** *p* < 0.001; ns, no significant difference.

## Data Availability

The original contributions presented in the study are included in the article. Further inquiries can be directed to the corresponding authors. The raw RNA-seq data analyzed in this study are available from Science Data Bank database (https://doi.org/10.57760/sciencedb.19947). The raw LC-MC metabolomic data analyzed in this study are available from the China National Center for Bioinformation OMIX database (OMIX ID: OMIX008662).

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
