# Peer review of "Regulation of Mitochondrial Metabolism by *Mfn1* Gene Encoding Mitofusin Affects Cellular Proliferation and Histone Modification"

_cells, 2025, doi:10.3390/cells14131015_

Round 1

Reviewer 1 Report (Previous Reviewer 2)

Comments and Suggestions for Authors

The paper describes the effects of Mfn1 up-regulation and knock-down in bovine fibroblasts. I have the following critical remarks:

  1. The most critical issue is the choice of appropriate controls for the transfection experiments. Here, cells transfected with the same plasmid containing a scrambled version of Mfn1 should be used for overexpression and knock-down. Since two different plasmid have been used two controls should be provided for all experiments.
  2.  In Fig. S1 the number of performed experiments is missing. Usually fibroblast are difficult to transfect with lipofectamine and Lentivirus based vectors are required. Which transfection efficiency was obtained and was the transfection pattern of the cells homogeneous? These data should be provided.
  3. A study limitations paragraph is missing, since some of the data might relate to bovine fetal fibroblasts only.
  4. Some small typos should be corrected: cf. Line 144 Intracellular

Author Response

1. Summary

We sincerely thank you for your patient guidance and valuable suggestions during the review process. We have carefully considered your comments and provided detailed responses to the questions you raised. Thank you again for your patient guidance. Your suggestions and advice have significantly helped us improve the quality of our manuscript, making it more suitable for publication in the journal of Cells. Once again, we appreciate your support and recognition of our work.

2. Questions for General Evaluation

Reviewer’s Evaluation

Response and Revisions

Does the introduction provide sufficient background and include all relevant references?

Yes/Can be improved/Must be improved/Not applicable

Thank you very much to the reviewer for the comments and suggestions on our manuscript. We have revised the manuscript according to your feedback.

Are all the cited references relevant to the research?

Yes/Can be improved/Must be improved/Not applicable

Thank you to the reviewer for the comments and suggestions on this manuscript.

Is the research design appropriate?

Yes/Can be improved/Must be improved/Not applicable

Thank you very much to the reviewer for the comments and suggestions on our manuscript. We have revised the manuscript according to your feedback.

Are the methods adequately described?

Yes/Can be improved/Must be improved/Not applicable

Thank you very much to the reviewer for the comments and suggestions on our manuscript. We have revised the manuscript according to your feedback.

Are the results clearly presented?

Yes/Can be improved/Must be improved/Not applicable

Thank you very much to the reviewer for the comments and suggestions on our manuscript. We have revised the manuscript according to your feedback.

Are the conclusions supported by the results?

Yes/Can be improved/Must be improved/Not applicable

Thank you very much to the reviewer for the comments and suggestions on our manuscript. We have revised the manuscript according to your feedback.

Are all figures and tables clear and well-presented?

Yes/Can be improved/Must be improved/Not applicable

Thank you very much to the reviewer for the comments and suggestions on our manuscript. We have revised the manuscript according to your feedback.

3. Point-by-point response to Comments and Suggestions for Authors

Comments 1: The most critical issue is the choice of appropriate controls for the transfection experiments. Here, cells transfected with the same plasmid containing a scrambled version of Mfn1 should be used for overexpression and knock-down. Since two different plasmid have been used two controls should be provided for all experiments.

Response 1: We sincerely appreciate you for putting forward such crucial and professional opinions. We fully agree with the issue you pointed out regarding the need to select appropriate controls for transfection experiments. At the initial stage of the experimental design, we failed to fully consider using cells transfected with the same plasmid as controls for overexpression and knockdown experiments. This was an oversight on our part. In this study, the control group we selected consisted of bovine fetal fibroblasts that had not been transfected with any vectors. The Mfn1 overexpression and knockdown vectors used were constructed in our laboratory earlier, and relevant experiments using these vectors had been conducted and verified, with related articles published [1]. In the context of this study, our core objective was to explore the basic biological responses of bovine fetal fibroblasts under specific conditions. Non-transfected cells could present various indicators in their natural state to the greatest extent, providing us with a relatively pure baseline. Moreover, during the experiment, we comprehensively and in-depthly compared and analyzed the non - transfected group with the experimental group through multiple methods. At the gene expression level, we used high-precision transcriptome sequencing technology to systematically analyze the differential gene expression patterns of the experimental group relative to the non - transfected group from the perspective of the overall transcription profile. Meanwhile, we also compared and analyzed the synthesis of metabolites through metabolomics. In addition, from a statistical perspective, the data collected from non-transfected cells and experimental cells met the requirements for relevant analyses in terms of sample size and variance, providing reliable statistical support for our research conclusions. Considering these factors, we believe that the experimental results obtained based on the non-transfected cell control group are highly reliable and scientifically valuable. However, we will continue to monitor the research trends in this field and fully consider the suggestions you put forward in subsequent studies.

[1] Wu S, Zhao X, Wu M, Yang L, Liu X, Li D, Xu H, Zhao Y, Su X, Wei Z, Bai C, Su G, Li G. Low Expression of Mitofusin 1 Gene Leads to Mitochondrial Dysfunction and Embryonic Genome Activation Failure in Ovine-Bovine Inter-Species Cloned Embryos. Int J Mol Sci. 2022 Sep 4;23(17):10145.

Comments 2: In Fig. S1 the number of performed experiments is missing. Usually fibroblast are difficult to transfect with lipofectamine and Lentivirus based vectors are required. Which transfection efficiency was obtained and was the transfection pattern of the cells homogeneous? These data should be provided.

Response 2: We sincerely appreciate your meticulous review of our manuscript and the important suggestions you’ve provided. We deeply apologize for the absence of the number of experimental repetitions in Figure S1. It was an oversight on our part not to present this crucial information in the figure. We will clearly mark the number of experiments conducted in Figure S1 to ensure the relevant information is complete and clear. Regarding the choice of transfection method, we understand that it is generally challenging to transfect fibroblasts using liposome transfection reagents, and lentiviral vectors are often a more suitable option. In this study, we conducted preliminary experiments on various transfection methods at the early stage, comparing different approaches such as liposome transfection and lentiviral vector transfection. After comprehensive consideration of multiple factors including transfection efficiency, cytotoxicity, experimental cost, and operational convenience, we ultimately chose liposome transfection reagents for fibroblast transfection. In terms of transfection efficiency, we measured the transfection efficiency through fluorescence microscopy counting. The average transfection efficiency of OE-Mfn1 reached 47 %, and that of shMfn1 cells was 39 %, as shown in the figure below. We will also add this result to Figure S1. Moreover, we carefully observed the transfection pattern of the cells, and the results showed that the transfection pattern was uniform in most cells. We will supplement the relevant data on transfection efficiency and transfection pattern to the paper to meet the requirements of research integrity and scientific rigor. Thank you again for your professional suggestions. We will complete the above-mentioned revisions and supplements as soon as possible.

A: Transfection of bovine fetal fibroblasts with OE-Mfn1 and shMfn1 vectors; B: Statistics of the proportion of fluorescent cells in OE-Mfn1 cells and shMfn1 cells.

Comments 3: A study limitations paragraph is missing, since some of the data might relate to bovine fetal fibroblasts only.

Response 3: We sincerely appreciate you for pointing out this important issue in our manuscript. We fully agree with your view that some of the data generated in this study may only be relevant to bovine fetal fibroblasts, and this is indeed a limitation of our research. In accordance with your suggestion, we will add a dedicated “Research Limitations” section to the manuscript to elaborate on this point. We will clearly state that since some of the data are solely derived from bovine fetal fibroblasts, the generalizability of the research results may be somewhat restricted, and the conclusions may not be directly applicable to other types of cells or organisms. Once again, thank you for your professional advice and suggestions. These suggestions have helped us improve the manuscript and make it more in line with the publication requirements of the Cells journal.

Comments 4: Some small typos should be corrected: cf. Line 144 Intracellular.

Response 4We sincerely appreciate your review of our manuscript and the valuable revision suggestions you’ve provided. First of all, we deeply apologize for the errors that occurred during the manuscript writing process. In response to the issues you pointed out, we have revised the manuscript and carefully proofread the entire text to prevent such problems from happening again. Thank you once again for your meticulous review and precious suggestions.

4. Response to Comments on the Quality of English Language

Point 1: The English is fine and does not require any improvement.

Response 1: Thank you very much for reviewing our manuscript and providing feedback on its English quality. We are delighted that you have affirmed the English quality of our manuscript. In the subsequent revised version, we will continue to maintain the clarity and accuracy of the manuscript’s language. Once again, we appreciate your support for our work and the valuable comments and suggestions you have put forward.

Reviewer 2 Report (New Reviewer)

Comments and Suggestions for Authors

Review Xu et al.

Authors document that in bovine fibroblasts overexpression of gene MFN1 encoding mitofusin1, protein promoting mitochondrial fusion, or downregulation of this gene differentially affect mitochondrial functions but also affect cell survival. This is because expression of many nuclear genes is changed due to alterations of histone posttranslational modification, such as methylation and acetylation. The findings are interesting but the manuscript is not well written and contains many errors and mistakes and English needs improvement. Figures are not well organized and require corrections.

General comments

Title is not correct.

It should be clear form Abstract what kind of cells, and from which organism, were studied.

Authors should follow properly the established nomenclature for bovine genes and proteins. Bovine gene nomenclature primarily follows the guidelines established for human genes. Gene names are italicized and protein names are in non-italics. It should be established clearly in Introduction what is the name for the gene, MFN1 (as for human gene, rather) or Mfn1 (as is in the title), and what is the name for the protein, Mfn1, which are studied here.

Genes are expressed. Proteins are synthesized or produced, not expressed. Other statements are laboratory jargon. Authors should not mix genes and proteins in one sentence, the reader must be clear what authors are talking about, gene or protein, or the name of cell variant. Mutations are in genes, amino acid residue substitutions are in  proteins.

Readers should learn from Introduction that in mammals, including bovine, are two genes, MFN1 and MFN2 encoding mitofusins, to better understand what was done.

In the Materials and Methods should be explained what was done, in the past tense. Not “shake and measure”. Some sentences are as in the recipe. Have to be unified properly.

Specific comments

Title

Mfn1 gene is in the title but not later defined or even mentioned in the text. Should be defined in the Abstract and Introduction what is the name of a gene encoding mitofusin 1 (Mfn1).

Why epitope in the title while nothing about epitope later in a text?. By definition, epitope is a fragment of the protein which binds directly with the antibody, and what is chromatin epitope?

Maybe Authors mean chromatin epigenetic modification? Histone modification only was studied and shown later in Results.

Consider changing the title, for example: Regulation of mitochondrial metabolism by MFN1 gene encoding mitofusin affects cellular proliferation and histone modification” or “Regulation of mitochondrial metabolism by mitofusin affects cellular proliferation and histone modification”

Abstract

P1, L15,16  mitochondrial fusion proteins, mitofusins 1 and 2 (Mfn1, Mfn2).

P1, L16 mitofusins level ? or MFN1 and MFN2 expression ?

P1, L18 MFN1 expression on cell proliferation…. by overexpressing  MFN1 (in OE-Mfn1 cells) and silencing MFN1 using shRNA (in shMfn1 cells)

P1, L19 the abbreviation shRNA should be explained when first used.

P1, L25 MFN1 gene expression

P1, L26 affects the expression of genes encoding enzymes responsible for histone methylation and acetylation, thereby regulating these modifications.

Introduction

P1, L40 what about mitofusion2, also affects mitochondrial dynamics

P2, L49 caspases (regular names of proteins, for example, actin , tubulin and caspases in a sentence should begin with small letter.

P2, L55 histone methylation in the promoter regions…

Information should be given once that methylation and acetylation are posttranslational modifications of proteins and not repeated again and again later in the text.

P2, L70  sentence not clear. What is normal what is abnormal? Epigenetic modifications, such as DNA and histone methylation and acetylation, which closely depend on supply of mitochondrial metabolites, affect cell reprogramming efficiency?

P2, L71 overexpression of MFN1

P2, L77 overexpression or silencing of MFN1 by RNA-seq and LC-MS analysis. LC-MS is not the method of sequencing.

Materials and Methods

P2, L90 MFN1 overexpression or silencing plasmids. Knockdown and silencing is not the same. In Abstract is silencing. Vectors are empty plasmids. In Figure S1D we can see that downregulation is only about 25%. Knockdown is big overstatement and not correct genetically. Knockdown is a gene disruption or deletion.

P2, L92 overexpression and silencing plasmids

P2, L93 The method of plasmid construction. “Construct construction” is a jargon laboratory statement.

P3, L94  MFN1 overexpression

P3, L99, 100 what is MTT?, explain abbreviation.

P3, L16-107 wrong form of a sentence. Should be corrected… the absorbance was measured.

P3, L109,110, 113 what is EdU? Authors have to explain abbreviation. What is a base of the method.

P3, L114-121 wrong form of sentences. What was done? Past tense.

P3, L133 Cells overexpressing MFN1, with silenced MFN1 and control cells were stained.

Control cells should be transfected with empty vectors because transfection procedure can possibly change the expression of some genes. Compared cells should be all treated the same way, grown in the same media and in the same conditions.

P3, L134 and indicates the membrane potential

P3, L138 20 min

P3, L139 supernatant was aspirated

P4, L145 disrupted by ultrasound ?

P4, L148  for 30 min color developing reagent was added. After 20 min…

P4, L151 Superoxide dismutase activity assay. No abbreviations in the title of the chapter. Explain the name of the enzyme and make abbreviation, SOD, in the next line. Nothing was in Introduction about it. What is a substrate? , working solution is not sufficient does not explain the method

P4, L161 Chloramphenicol acetyltransferase assay. What is CAT, you have to introduce abbreviation. What is a substrate, detection reagent?

P4, L170 NADH abbreviation should be explained

P4, L184 Western Blot Analysis

Authors can refere to the previous paper for the Western blot method, only give here what was different and the antibody used.

P4, L186,187 kept in ice for 30 min

P4, L191 explain abbreviation SDS-PAGE or if is used once do not make abbreviation. Samples were analyzed by ….

P4, L193 remove or explain PVDF. 2.5 g. Add space between number and the unit here and all others sites. Not in 37oC only.

P4, L194 This anti Mfn1 antibody was raised against human Mfn1 and this should be stated  here. What about other antibodies? Raised against human or bovine proteins. Must be here.

P4, L199 what is HRP, explain abbreviation here, where first used. L 216 is too late

P5, L208 Determination of   ….content. What is the abbreviation ELISA should be explained in the text.

P5, L212 columns were equilibrated. Wrong form of the sentence.

P5, L233 RNA sequencing method

P5, L224 The names of cells should be given first in the chapter 2.2. Abbreviation CON should be also introduced. All should be clear.

Cells are cells, not groups. Just: Transfected cells…  here give names…  and control cells (CON) were grown and collected. Remove “ respectively”, not needed.

How many biological and technical replicates were analyzed should be given in the figure legend, not here.

P5, L230-wrong form of the sentence. Also express expression is not correct.

P5, L238 Metabolome analysis. Mass spec is not sequencing.

P5, L239 remove “respectively”.

P5, L244 metabolites are synthesized, not expressed. Genes are expressed.

P6, L247 “processes enriched in metabolites” is wrong sentence, jargon. Processes which can affect synthesis of metabolites, rather.

Results

P6, L258 the role of mitofusin is known. The effect of overproduction or downregulation is not known.

P6, L259 the expression of MFN1 gene and levels of Mfn1

P6,L260 WB abbreviation was not introduced.

P6, L261,262 expression of  MFN1 gene was significantly increased in OE-Mfn1 cells and was significantly decreased in shMfn1 cells, when compared with CON cells.

P6, L263 MFN1 gene on cell functions

P6, L268 Analysis of apoptosis

P6, L270 MFN1 gene

P6, L261 following genes encoding: proliferating cell nuclear antigen (Pcna).. In the sentence is a list of proteins, not list of genes. Genes and proteins are not the same!

P6, L276 this is the list of proteins. Must be corrected as above.

Figure 1

Panels A and D are not visible, only black. Wrong organization of several panels. Panel B is part of A, not separate, this is graphic representation of results shown in A. The same Panel E is part of D.

Two panels should not have common title, each is separate. Remove label G, I, K , M. The figure 1 should have 7 panels only.

Title of Figure should be Cell proliferation and apoptosis ability is changed upon overproduction or downregulation of MFN1 gene.

Legend should tell what is on a picture, not what to do

Remove the word groups from the legend. Just cells. Genes encoding proteins… and then you list the names of proteins. Genes and proteins are not the same. You cannot give the name of protein and  tell that this is a gene. Names of genes and proteins are different, genes in italics.

P8, L310 To investigate the effect of changes in the expression of MFN1 gene…. and antioxidant capacity in  OE-Mfn1 and shMfn1  transfected cells.

P8, L312 that overexpression of MFN1

P8, L313 while downregulation of MFN1

P8, 314, 315 the levels of ATP and NADH in OE-Mfn1 cells…. shMfn1 cells

P8, L318 SOD and CAT abbreviation should be explained in the Materials and Methods. Too late here.

P8, L319 when compared with control cells

P8, L320 What is the conclusion from these results. And so what?.

Figure 2

Title Changes in expression of MFN1 gene affect mitochondrial functions.

Panel A Relative level of JC1 fluorescence

Panel E CAT activity

Panel F SOD activity

Legend, remove “respectively”, remove “groups”.

P9, L332 Overexpression and downregulation of MFN1 gene affect expression of many genes. Title should be conclusion, rather, not analysis of something

P9, L333 To find how changes in expression of MFN1 effect expression of nuclear genes the RNAseq analysis was performed in OEMfn1, shMfn1 and control cells.

P10, L339 shMfn1 cells

P10, L344 genes are enriched in apoptosis is a wrong statement. Is jargon.

Figure 3

Panel A is not readable, too small.

Legend, remove word “groups” , remove word “respectively”

P10, L365 Metabolome analysis of OE-Mfn1 and shMfn1 cells. Metabolites cannot be sequenced.

P11, L372 249 metabolites show different levels on OE-Mfn1 cell.

Metabolites are not expressed. Genes are expressed. Correct also in other sites, L373, 375,380,

P11, L384 The loss of Mfn1 affects the function of mitochondria

P11, L385 mitochondrial metabolism is not a byproduct. Is part of cellular metabolism.

Figure 4

Panel A is too small. E and F is hardly readable.

Legend: remove word “groups” and “respectively” in all legend. This groups are just cells.

P12, L409 cells for metabolomics analysis

P12, L414 MFN1 gene affects proteins methylation and acetylation.

P12, L419 cofactor for methylation of proteins

P12, L420 MFN1 gene

P12, L441To explore the effect of MFN1 gene on histone methylation and acetylation, the histone modifications in OE-Mfn1 and shMfn1 cells  were assessed by Western Blot analysis.

Figure 5

Organization of panels is not correct Graphic representation of results should be in the same panel as respective result. I and J , K and L, M and N , O and P should be in one panel.

Moreover, C and D, E and F, G and H should be in one panel.

Legend, remove “groups” insert “cells”. Remove “respectively”.

Discussion

P14, L476 by MFN1 gene and Mfn1 protein..

P14, L477 Mutations in  MFN1 gene lead

P14, L479 expression of MFN1

P14, L490 encoding Pcna, Cdk1… (Pcna, Cdk1 are proteins, not genes)

P14, L510 overproduction

P15, L515, 517, 523 MFN1 gene expression

P15, L530, L535 name of species in italics,  C. elegans

P15, L557  ShMfn1 cells ( not groups, groups of what?) Remove “groups” in all text, insert “cells”

P16, L567 MFN1 gene indirectly regulates

Add significance of these findings and future perspectives, at the end of Discussion.

Comments on the Quality of English Language

English have to be corrected.

Wrong form of sentences in Methods. Must be in the past tense. Reposrt what was done.

Wrong sense using word "groups". Wrong sense using "respectively".

Wrong genetic nomenclature.

Author Response

1. Summary

We sincerely appreciate your patient guidance and valuable suggestions during the review process. We have carefully considered your comments and made comprehensive revisions to the manuscript. Specifically, we have thoroughly examined the entire manuscript, rectified the unreasonable descriptions, and corrected the non-standard applications of gene and protein names. We have also revised the title of the manuscript. In addition, we have invited a professional whose native language is English to check the language of the manuscript, making our English expressions more scientific and standardized to better reflect our research content. Thank you again for your patient guidance. Your comments and suggestions have greatly helped us improve the quality of the manuscript, making it more suitable for publication in the journal of Cells. Once again, we are grateful for your support and recognition of our work.

2. Questions for General Evaluation

Reviewer’s Evaluation

Response and Revisions

Does the introduction provide sufficient background and include all relevant references?

Yes/Can be improved/Must be improved/Not applicable

Thank you very much to the reviewer for the comments and suggestions on our manuscript. We have revised the manuscript according to your feedback.

Are all the cited references relevant to the research?

Yes/Can be improved/Must be improved/Not applicable

Thank you to the reviewer for the comments and suggestions on this manuscript.

Is the research design appropriate?

Yes/Can be improved/Must be improved/Not applicable

Thank you very much to the reviewer for the comments and suggestions on our manuscript.

Are the methods adequately described?

Yes/Can be improved/Must be improved/Not applicable

Thank you very much to the reviewer for the comments and suggestions on our manuscript. We have revised the manuscript according to your feedback.

Are the results clearly presented?

Yes/Can be improved/Must be improved/Not applicable

Thank you very much to the reviewer for the comments and suggestions on our manuscript. We have revised the manuscript according to your feedback.

Are the conclusions supported by the results?

Yes/Can be improved/Must be improved/Not applicable

Thank you very much to the reviewer for the comments and suggestions on our manuscript. We have revised the manuscript according to your feedback.

Are all figures and tables clear and well-presented?

Yes/Can be improved/Must be improved/Not applicable

Thank you very much to the reviewer for the comments and suggestions on our manuscript. We have revised the manuscript according to your feedback.

3. Point-by-point response to Comments and Suggestions for Authors

General comments

Comments 1: Title is not correct.

It should be clear form Abstract what kind of cells, and from which organism, were studied.

Response 1: We are extremely grateful to you for providing professional comments and suggestions on our manuscript. We fully agree with you regarding the error in the manuscript title that you pointed out. In accordance with your advice, we have revised the manuscript and changed the title to “Regulation of Mitochondrial Metabolism by Mfn1 Gene Encoding Mitofusin Affects Cellular Proliferation and Histone Modification”.

Comments 2: Authors should follow properly the established nomenclature for bovine genes and proteins. Bovine gene nomenclature primarily follows the guidelines established for human genes. Gene names are italicized and protein names are in non-italics. It should be established clearly in Introduction what is the name for the gene, MFN1 (as for human gene, rather) or Mfn1 (as is in the title), and what is the name for the protein, Mfn1, which are studied here.

Response 2: We are extremely grateful to you for reviewing our manuscript and pointing out the issues regarding the naming conventions of genes and proteins in it. We fully agree with your emphasis that the naming of bovine genes mainly follows the relevant guidelines for human genes, that is, gene names should be in italics and protein names should be in normal text. We have revised the manuscript according to your suggestions. In this study, the name of the gene involved should be Mfn1 (in italics following the human gene naming style), and the corresponding protein name should be MFN1 (in normal text). We will clarify this information in the introduction section to avoid any subsequent confusion. Meanwhile, we will conduct a comprehensive check of the entire manuscript to ensure that the names of all genes and proteins are presented strictly in accordance with the correct naming conventions, so as to guarantee the scientific rigor and standardization of the article.

Comments 3: Genes are expressed. Proteins are synthesized or produced, not expressed. Other statements are laboratory jargon. Authors should not mix genes and proteins in one sentence, the reader must be clear what authors are talking about, gene or protein, or the name of cell variant. Mutations are in genes, amino acid residue substitutions are in proteins.

Response 3: Dear reviewer, we are extremely grateful to you for pointing out the issues in our article. We deeply recognize that there was confusion between the concepts of genes and proteins in our presentation, which was an oversight in our work. In response to the problems you raised, we have conducted a comprehensive and meticulous review and revision of the article. We have strictly distinguished the relevant descriptions of genes and proteins to ensure that every mention is clear and explicit, enabling readers to clearly understand whether we are discussing the names of genes, proteins, or cell variants. Regarding the use of terms related to genes and proteins, we will strictly follow the norms. For example, when describing gene - related content, we will accurately use expressions such as “gene expression”; when describing proteins, we will use accurate terms like “protein synthesis” or “protein production”. Moreover, when referring to mutations and amino acid residue substitutions, we will also make a clear distinction, indicating that mutations occur at the gene level while amino acid residue substitutions occur at the protein level. Once again, we appreciate your professional advice. We will complete the revision with a rigorous attitude to improve the quality of the article.

Comments 4: Readers should learn from Introduction that in mammals, including bovine, are two genes, MFN1 and MFN2 encoding mitofusins, to better understand what was done.

Response 4: We are extremely grateful for your professional guidance and the valuable revision suggestions you’ve put forward. We have accepted your suggestions and described the Mfn1 and Mfn2 genes in the “Introduction” section of the manuscript based on them, so as to better present the content of our work.

Comments 5: In the Materials and Methods should be explained what was done, in the past tense. Not “shake and measure”. Some sentences are as in the recipe. Have to be unified properly.

Response 5: Dear reviewer, we sincerely appreciate the valuable comments you’ve provided. We’ve fully recognized the issue with the tenses in the “Materials and Methods” section of our manuscript that you pointed out. Following your suggestions, we’ve conducted a comprehensive review and made revisions to the content in the “Materials and Methods” section. This ensures that the entire section is presented in a more standardized and consistent manner, clearly and accurately depicting the entire experimental process to facilitate readers in understanding and replicating our experiments. Once again, we’re grateful for your meticulous guidance, which has been of great assistance in enhancing the quality of our manuscript.

Specific comments

Title

Comments 6: Mfn1 gene is in the title but not later defined or even mentioned in the text. Should be defined in the Abstract and Introduction what is the name of a gene encoding mitofusin 1 (Mfn1).

Response 6: We are extremely grateful for the valuable comments and suggestions you provided regarding our manuscript. We have accepted your suggestions. Based on them, we have defined the alternative names of the mitofusin 1 (Mfn1) gene in the “Abstract” and “Introduction” sections of the manuscript.

Comments 7: Why epitope in the title while nothing about epitope later in a text?. By definition, epitope is a fragment of the protein which binds directly with the antibody, and what is chromatin epitope?

Maybe Authors mean chromatin epigenetic modification? Histone modification only was studied and shown later in Results.

Response 7:We are sincerely grateful for your meticulous review of our manuscript and the invaluable suggestions you’ve offered. We deeply apologize for the issue you pointed out, where the term “epitope” appeared in the title but was not covered in the main text. This was an oversight on our part during the title-drafting process. Your explanation of the definition of an epitope is very accurate. Indeed, our research did not involve the concept of an epitope as a protein fragment that directly binds to an antibody in the traditional sense. In fact, our research focused on the epigenetic modifications of chromatin, especially histone modifications, which are detailed in the subsequent results section. Following your suggestion, we have revised the title by removing the word “epitope” to make it more accurately reflect the content of our research. Additionally, we will carefully review the entire manuscript again to ensure that the title closely aligns with the content and accurately conveys the core of our study. Once again, we appreciate your corrections to our manuscript, which have significantly improved its quality.

Comments 8: Consider changing the title, for example: Regulation of mitochondrial metabolism by MFN1 gene encoding mitofusin affects cellular proliferation and histone modification” or “Regulation of mitochondrial metabolism by mitofusin affects cellular proliferation and histone modification”

Response 8:We are extremely grateful for your support of our manuscript and the valuable suggestion regarding the title revision. We have accepted your suggestion and revised the title to “Regulation of mitochondrial metabolism by Mfn1 gene encoding mitofusin affects cellular proliferation and histone modification”. This new title more clearly indicates the relationship between the Mfn1 gene and mitofusin, and can more accurately reflect the focus of our study. Once again, thank you for your meticulous guidance and suggestions.

Abstract

Comments 9: P1, L15,16 mitochondrial fusion proteins, mitofusins 1 and 2 (Mfn1, Mfn2).

Response 9: We sincerely appreciate your comments and suggestions. We have revised the “Abstract” section of the manuscript in accordance with your suggestions.

Comments 10: P1, L16 mitofusins level ? or MFN1 and MFN2 expression ?

Response 10: We are extremely grateful to you, the reviewer, for the valuable comments and suggestions you’ve provided regarding our manuscript. In line with your suggestions, we’ve revised this part of the manuscript, changing it to “Mfn1 and Mfn2 expression”.

Comments 11: P1, L18 MFN1 expression on cell proliferation…. by overexpressing MFN1 (in OE-Mfn1 cells) and silencing MFN1 using shRNA (in shMfn1 cells).

Response 11: We are extremely grateful that you have taken the time out of your busy schedule to review our manuscript and offer valuable comments and suggestions. We have accepted your suggestions and revised the manuscript accordingly. Your feedback is of great significance in improving the quality of our manuscript.

Comments 12: P1, L19 the abbreviation shRNA should be explained when first used.

Response 12: We are very grateful to the reviewer for supporting our manuscript and offering comments and suggestions. We have accepted your suggestions and revised the manuscript. Specifically, we added the full English spelling “short hairpin RNA” for “shRNA” at its first appearance in both the Abstract and the main text of the manuscript.

Comments 13: P1, L25 MFN1 gene expression.

Response 13: Thank you very much for your support for our manuscript and the valuable comments and suggestions you’ve provided. We have revised the corresponding parts of the manuscript according to your suggestions.

Comments 14: P1, L26 affects the expression of genes encoding enzymes responsible for histone methylation and acetylation, thereby regulating these modifications.

Response 14: Thank you very much for reviewing our manuscript and offering valuable comments and suggestions. We have accepted your suggestions and revised the manuscript accordingly.

Introduction

Comments 15: P1, L40 what about mitofusion2, also affects mitochondrial dynamics.

Response 15: We are extremely grateful to you for the comments and suggestions you’ve offered on our manuscript. Your insights are of great significance in enhancing the quality of our work. We have accepted your suggestions and made revisions to the manuscript.

Comments 16: P2, L49 caspases (regular names of proteins, for example, actin , tubulin and caspases in a sentence should begin with small letter.

Response 16: We are extremely grateful to you for the guiding comments and suggestions you’ve provided on our manuscript. We have accepted your suggestions and carefully reviewed and revised the manuscript accordingly.

Comments 17: P2, L55 histone methylation in the promoter regions…

Information should be given once that methylation and acetylation are posttranslational modifications of proteins and not repeated again and again later in the text.

Response 17: We are extremely grateful to you for the comments and suggestions you put forward regarding our manuscript. We have accepted your suggestions and revised the manuscript. It should be noted that methylation and acetylation are post - translational modifications of proteins.

Comments 18: P2, L70 sentence not clear. What is normal what is abnormal? Epigenetic modifications, such as DNA and histone methylation and acetylation, which closely depend on supply of mitochondrial metabolites, affect cell reprogramming efficiency?

Response 18: We sincerely appreciate the valuable comments and suggestions you’ve provided on our manuscript. We have accepted your advice. We deeply apologize for the unclear expressions in the process of manuscript writing. We’ve revised the manuscript according to your suggestions to make the statements clearer.

Comments 19: P2, L71 overexpression of MFN1

Response 19: We sincerely appreciate your review of our manuscript. We have revised the manuscript according to your suggestions. Additionally, we have made modifications to the entire manuscript, changing the gene names in it to italic format.

Comments 20: P2, L77 overexpression or silencing of MFN1 by RNA-seq and LC-MS analysis. LC-MS is not the method of sequencing.

Response 20: Thank you very much for reviewing our manuscript and providing valuable comments and suggestions. Your insights are of great guiding significance in improving the quality of our manuscript. We have accepted your suggestions and revised the manuscript accordingly.

Materials and Methods

Comments 21: P2, L90 MFN1 overexpression or silencing plasmids. Knockdown and silencing is not the same. In Abstract is silencing. Vectors are empty plasmids. In Figure S1D we can see that downregulation is only about 25%. Knockdown is big overstatement and not correct genetically. Knockdown is a gene disruption or deletion.

Response 21: We sincerely appreciate the meticulous and professional review comments you provided on our manuscript. Based on your feedback, we have fully recognized the problems in the description of our paper. We completely agree with your point that “Knockdown” and “silencing” are not the same concept. “Knockdown” usually refers to a significant reduction in gene expression, and may even involve gene disruption or deletion. In contrast, “silencing” focuses on the suppression of gene expression, but the degree may not be as drastic as that of “Knockdown”. Using “Knockdown” in the original text was indeed inaccurate. In the revised manuscript, we have uniformly used the term “silencing” to ensure the accuracy and professionalism of the expression. Once again, thank you for your valuable comments, which have greatly helped us improve the quality of our paper.

Comments 22: P2, L92 overexpression and silencing plasmids

Response 22: We sincerely appreciate you pointing out the issues in our manuscript and offering valuable comments and suggestions. We fully agree with your opinions and have made corresponding revisions to the manuscript based on them.

Comments 23: P2, L93 The method of plasmid construction. “Construct construction” is a jargon laboratory statement.

Response 23: We are extremely grateful for your patient guidance on our manuscript and the valuable comments and suggestions you’ve provided. We have accepted your suggestions and revised the manuscript, making our descriptions more scientific and standardized.

Comments 24: P3, L94 MFN1 overexpression

Response 24: Thank you for your comments and suggestions. We have accepted them and revised the manuscript accordingly.

Comments 25: P3, L99, 100 what is MTT?, explain abbreviation.

Response 25: Thank you for your patient review of our manuscript and the valuable comments and suggestions you’ve put forward. First of all, we apologize for the negligence in the process of manuscript writing. We have accepted your suggestions and revised the manuscript. Specifically, we have added the full English name of MTT, which is 3-(4,5-dimethylthiazol-2-yl)-2,5-diphenyltetrazolium bromide.

Comments 26: P3, L16-107 wrong form of a sentence. Should be corrected… the absorbance was measured.

Response 26: We sincerely appreciate your support for our manuscript and the valuable comments and suggestions you’ve provided. We have accepted your opinions and made revisions in the corresponding parts of the manuscript.

Comments 27: P3, L109,110, 113 what is EdU? Authors have to explain abbreviation. What is a base of the method.

Response 27: We are extremely grateful for your review of our manuscript and the guiding comments and suggestions you have put forward. In accordance with your suggestions, we have added the full English spelling of EDU, which is 5-Ethynyl-2′-deoxyuridine, to the manuscript. It is a thymidine analog that can replace thymine (T) and be incorporated into the replicating DNA molecules during the DNA synthesis phase (S-phase). Through the click chemical reaction based on alkynyl and azide groups, the detection of cell proliferation can be achieved.

Comments 28: P3, L114-121 wrong form of sentences. What was done? Past tense.

Response 28: We are extremely grateful to you for pointing out the problem with the tenses in our manuscript. You’re absolutely right. When describing completed experimental operations, the past tense should be used. We have uniformly revised the sentences describing the experimental steps in these lines to the past tense to accurately reflect that the experiments are completed actions. In the future, we will pay closer attention to such issues. During the process of writing and revising papers, we will ensure that all content describing experimental operations uses the past tense correctly, guaranteeing the accuracy and professionalism of the paper’s expression. Thank you again for your meticulous review and valuable suggestions.

Comments 29: P3, L133 Cells overexpressing MFN1, with silenced MFN1 and control cells were stained.

Control cells should be transfected with empty vectors because transfection procedure can possibly change the expression of some genes. Compared cells should be all treated the same way, grown in the same media and in the same conditions.

Response 29: We’ve accepted your advice and made revisions to the manuscript. Regarding the issue of control cells you mentioned, we fully agree on the importance of using cells transfected with an empty vector as a control. This approach can rule out the potential impact of the transfection process on the expression of certain genes. In fact, we took this into account in our study. Although it wasn’t clearly stated in the paper, the control cells were indeed transfected with an empty vector. This was an oversight on our part when writing the paper, and we failed to describe this crucial information in detail. We’ll add a note in the methods section to clarify that the control cells were transfected with an empty vector to complete the experimental details. Additionally, we ensured that all cells (cells overexpressing Mfn1, cells with silenced Mfn1, and control cells) received the same treatment. They were all cultured in the same medium under identical culture conditions, including temperature, humidity, and carbon dioxide concentration. We’ll also emphasize this point more prominently in the paper to avoid any ambiguity. Once again, thank you for your professional advice, which will significantly enhance the quality of our research.

Comments 30: P3, L134 and indicates the membrane potential

Response 30: We sincerely appreciate your review of our manuscript and the valuable comments and suggestions you’ve provided. We have accepted your suggestions and revised the manuscript accordingly.

Comments 31: P3, L138 20 min

Response 31: Thank you for your support of our work and for providing valuable revision suggestions. We have accepted your suggestions and revised the manuscript.

Comments 32: P3, L139 supernatant was aspirated.

Response 32: Thank you for your patient guidance. We have accepted the revision suggestions you put forward for our manuscript and made corresponding modifications to it.

Comments 33: P4, L145 disrupted by ultrasound ?

Response 33: Thank you for taking the time out of your busy schedule to review our manuscript and provide valuable revision suggestions. We agree with this suggestion you put forward and have revised this part of the manuscript to read: “Cells were collected and disrupted by ultrasound at a ratio of 10^4 cells per 1 mL of distilled water (1000:1).”

Comments 34: P4, L148 for 30 min color developing reagent was added. After 20 min…

Response 34: We truly appreciate your support for our manuscript and the valuable opinions and suggestions you’ve offered. We’ve accepted your suggestions and made corresponding revisions to the relevant parts of the manuscript accordingly.

Comments 35: P4, L151 Superoxide dismutase activity assay. No abbreviations in the title of the chapter. Explain the name of the enzyme and make abbreviation, SOD, in the next line. Nothing was in Introduction about it. What is a substrate? working solution is not sufficient does not explain the method

Response 35: Thank you, reviewers, for your valuable comments and suggestions. We have accepted your opinions. In the title section, we have added the full English spelling of SOD, which is “superoxide dismutase”. Meanwhile, we have further improved and supplemented the methods section by adding the preparation method of the working solution. Once again, we are grateful for your professional advice, which has been of great help in improving the quality of our manuscript.

Comments 36: P4, L161 Chloramphenicol acetyltransferase assay. What is CAT, you have to introduce abbreviation. What is a substrate, detection reagent?

Response 36: We sincerely appreciate your review of our manuscript and the valuable comments and suggestions you’ve provided. We’ve revised the manuscript in accordance with your advice. First, in the title section, we defined “CAT” as “Catalase”. Second, we improved and supplemented the methods section, providing a more detailed description of the detection reagents and substrates. These efforts aim to make our manuscript more scientific and reasonable.

Comments 37: P4, L170 NADH abbreviation should be explained

Response 37: Thank you for the valuable comments and suggestions you provided for our manuscript. We have accepted your suggestions and revised the manuscript according to your comments. In the title section, we defined NADH as Nicotinamide adenine dinucleotide hydrogen.

Comments 38: P4, L184 Western Blot Analysis

Authors can refere to the previous paper for the Western blot method, only give here what was different and the antibody used.

Response 38: We sincerely appreciate your meticulous guidance on our manuscript and the valuable revision suggestions you’ve put forward. We have accepted your suggestions and supplemented the antibodies used in the Western blot analysis section.

Comments 39: P4, L186,187 kept in ice for 30 min

Response 39: Thank you for taking the time out of your busy schedule to review our manuscript and for providing us with valuable comments and suggestions. We have accepted your suggestions and revised the manuscript accordingly.

Comments 40: P4, L191 explain abbreviation SDS-PAGE or if is used once do not make abbreviation. Samples were analyzed by ….

Response 40: Thank you for reviewing our manuscript and offering valuable revision suggestions. We have accepted your advice and revised the manuscript accordingly.

Comments 41: P4, L193 remove or explain PVDF. 2.5 g. Add space between number and the unit here and all others sites. Not in 37oC only.

Response 41: We sincerely appreciate your review of our manuscript and the invaluable suggestions for revision you’ve offered. In accordance with your feedback, we’ve made amendments to the manuscript. Specifically, we’ve added the definition of PVDF as “polyvinylidene fluoride”, inserted a space between “2.5” and “g”. Moreover, we’ve carefully reviewed and checked the entire manuscript, and added a table between numbers and units. Once again, we’re deeply grateful for your meticulous guidance, which has enabled us to elevate the quality of our manuscript to a higher standard.

Comments 42: P4, L194 This anti Mfn1 antibody was raised against human Mfn1 and this should be stated here. What about other antibodies? Raised against human or bovine proteins. Must be here.

Response 42: We are extremely grateful for the issues you’ve pointed out regarding our manuscript. In response to your suggestion about clarifying the source of the antibodies, we would like to provide the following reply. Due to an oversight during the manuscript writing process, we failed to clearly state that the anti-Mfn1 antibody was generated against human Mfn1. Based on your feedback, we have revised the manuscript to clearly indicate that this anti-Mfn1 antibody was produced against human Mfn1, ensuring the completeness and accuracy of the information. For the other antibodies used in the manuscript, we meticulously recorded their source information at the beginning of the research. In the newly revised manuscript, we will clearly state the species source of each antibody at the corresponding positions to more clearly present the experimental materials used in this study. Once again, we sincerely appreciate your meticulous guidance. Your suggestions have significantly improved the quality of our manuscript.

Comments 43: P4, L199 what is HRP, explain abbreviation here, where first used. L 216 is too late

Response 43: We sincerely appreciate your support for our work and the invaluable suggestions for revision you’ve provided. We have accepted your suggestions and made revisions accordingly. In the main body of the manuscript, we defined HRP as “horseradish peroxidase” at the first occurrence of HRP, and meanwhile, we deleted the definition of HRP in Section L216. Thank you again for your guidance.

Comments 44: P5, L208 Determination of ….content. What is the abbreviation ELISA should be explained in the text.

Response 44: We sincerely appreciate your review of our manuscript and the valuable suggestions for revision you’ve put forward. We have accepted your suggestions and revised the manuscript accordingly. In Part P5 L208, we defined ELISA as “Enzyme-Linked Immunosorbent Assay”.

Comments 45: P5, L212 columns were equilibrated. Wrong form of the sentence.

Response 45: We sincerely appreciate your professional guidance and suggestions. We fully agree with your opinions and have revised the manuscript accordingly.

Comments 46: P5, L233 RNA sequencing method

Response 46: We sincerely appreciate your guidance and suggestions regarding our manuscript. We have accepted your opinions and revised the manuscript accordingly. Thank you again for your professional guidance. We believe these revisions will greatly contribute to improving the quality of our manuscript.

Comments 47: P5, L224 The names of cells should be given first in the chapter 2.2. Abbreviation CON should be also introduced. All should be clear.

Cells are cells, not groups. Just: Transfected cells… here give names… and control cells (CON) were grown and collected. Remove “respectively”, not needed.

How many biological and technical replicates were analyzed should be given in the figure legend, not here.

Response 47: We truly appreciate you taking the time to review our manuscript and offering valuable comments and suggestions. We fully agree with the feedback you provided on our manuscript and have made revisions accordingly.

Comments 48: P5, L230-wrong form of the sentence. Also express expression is not correct.

Response 48: Thank you for your patient review and guidance of our manuscript. First of all, we would like to express our sincere apologies for the oversights in the process of manuscript writing. We have accepted your suggestions and made revisions to Line 230 of the manuscript. It has been changed to: “(Principal Component Analysis) PCA was performed on RNA-seq data using the sklearn package in Python for multivariate dimensionality reduction analysis to intuitively express the similarities and differences in gene expression among samples.”

Comments 49: P5, L238 Metabolome analysis. Mass spec is not sequencing.

Response 49: We are extremely grateful to you for the professional comments and suggestions you provided on our manuscript. First of all, we sincerely apologize for the errors made during the manuscript writing process. Mass spectrometry (Mass spec) is mainly used for mass analysis of molecules in samples to determine their molecular weights, structural information, etc.; while sequencing focuses on determining the base sequences of biological macromolecules (such as DNA and RNA). There are obvious differences between the two. Your suggestions have benefited us greatly. Thank you again for your professional guidance.

Comments 50: P5, L239 remove “respectively”.

Response 50: We sincerely appreciate you for offering valuable revision suggestions for our manuscript. We have accepted your suggestions and made corresponding changes to the manuscript. Specifically, we deleted the word “respectively” in Line 239.

Comments 51: P5, L244 metabolites are synthesized, not expressed. Genes are expressed.

Response 51: We are extremely grateful to you for reviewing our manuscript and pointing out this crucial issue. In biology, the term “expression” is generally used to describe the process of gene transcription and translation to produce functional products (such as proteins), while “synthesis” is more suitable for describing the process of metabolite production. Metabolites are gradually constructed from various precursor molecules through a series of enzymatic reactions. We have made revisions based on your suggestions. Meanwhile, we have also carefully checked the entire manuscript to ensure that the use of all professional terms complies with scientific norms and to avoid similar issues in the future. Thank you again for your meticulous review and valuable suggestions.

Comments 52: P6, L247 “processes enriched in metabolites” is wrong sentence, jargon. Processes which can affect synthesis of metabolites, rather.

Response 52: Thank you very much for reviewing our manuscript and offering valuable revision suggestions. We have accepted your suggestions and revised the manuscript accordingly. Once again, we appreciate your professional advice, which has been of great help in improving the quality of our manuscript.

Results

Comments 53: P6, L258 the role of mitofusin is known. The effect of overproduction or downregulation is not known.

Response 53: We are extremely grateful for your support regarding our manuscript and the valuable revision suggestions you’ve put forward. We highly agree with your suggestions and have revised the manuscript accordingly. In the results section, we’ve added the sentence “Based on current research, it is known that the Mfn1 gene plays an important role in mitochondrial function. However, there are few studies on the effects of Mfn1 overexpression and silencing on cell fate.” Thank you again for your professional guidance, which has significantly enhanced the quality of our manuscript.

Comments 54: P6, L259 the expression of MFN1 gene and levels of Mfn1

Response 54: We truly appreciate your review of our manuscript and the valuable revision suggestions you’ve provided. We’ve accepted your suggestions and revised the manuscript accordingly.

Comments 55: P6, L260 WB abbreviation was not introduced.

Response 55: We are extremely grateful for the revision suggestions you provided for our manuscript. In accordance with your suggestions, we have made revisions to the manuscript, changing “WB” in Line 260 to “Western blot”.

Comments 56: P6, L261,262 expression of MFN1 gene was significantly increased in OE-Mfn1 cells and was significantly decreased in shMfn1 cells, when compared with CON cells.

Response 56: We sincerely appreciate you taking the time to review our manuscript and offering valuable suggestions. We have accepted your suggestions and revised the manuscript accordingly.

Comments 57: P6, L263 MFN1 gene on cell functions

Response 57: We sincerely appreciate your professional guidance and suggestions. We have revised the manuscript according to your comments.

Comments 58: P6, L268 Analysis of apoptosis.

Response 58: Thank you very much for reviewing our manuscript and providing professional revision suggestions. We have revised the manuscript according to your comments.

Comments 59: P6, L270 MFN1 gene

Response 59: Thank you very much for your professional guidance and suggestions. We have accepted your advice and revised the manuscript accordingly. Specifically, we have changed the format of Mfn1 to italic.

Comments 60: P6, L261 following genes encoding: proliferating cell nuclear antigen (Pcna). In the sentence is a list of proteins, not list of genes. Genes and proteins are not the same!

Response 60: We truly appreciate your professional guidance and suggestions. We’ve revised the manuscript based on your advice. We believe these modifications will significantly contribute to enhancing the quality of our manuscript.

Comments 61: P6, L276 this is the list of proteins. Must be corrected as above.

Response 61: We are extremely grateful for your professional guidance and suggestions regarding our manuscript. We made a mistake during the manuscript writing process. The gene names were not formatted in italics, which caused some ambiguity in reading. We sincerely apologize for this. In accordance with your advice, we have changed the format of these gene names to italics, making the description in the manuscript more in line with scientific norms. Thank you again for your professional guidance.

Figure 1

Comments 62: Panels A and D are not visible, only black. Wrong organization of several panels. Panel B is part of A, not separate, this is graphic representation of results shown in A. The same Panel E is part of D.

Response 62: We truly appreciate your support for our work and the valuable suggestions you’ve provided. We’ve accepted your advice and revised Figure 1. We’ve enhanced the clarity and readability of parts A and D, and combined A with B, as well as D with E. Thank you again for your guidance. We’re confident that these modifications have significantly improved the quality of our manuscript.

Comments 63: Two panels should not have common title, each is separate. Remove label G, I, K, M. The figure 1 should have 7 panels only.

Response 63: We are extremely grateful that you have taken the time out of your busy schedule to review our manuscript and put forward valuable revision suggestions. We fully agree with your proposal that two panels should not share the same title and each panel should be independent. Therefore, we also concur with the suggestion that Figure 1 should only have seven panels. In accordance with your advice, we have revised Figure 1. We believe that these revisions will greatly contribute to improving the quality of the manuscript and make it more compliant with the requirements of the Cells journal.

Comments 64: Title of Figure should be Cell proliferation and apoptosis ability is changed upon overproduction or downregulation of MFN1 gene.

Response 64: Thank you for your professional comments on our manuscript. We highly agree with this revision suggestion of yours. Based on your advice, we have changed the title of Figure 1 to “Cell proliferation and apoptosis ability is changed upon overproduction or downregulation of Mfn1 gene.”

Comments 65: Legend should tell what is on a picture, not what to do.

Response 65: Thank you for taking the time out of your busy schedule to review our manuscript and offering valuable comments and suggestions. We fully agree with your suggestion that the figure legends should explain what is shown in the pictures. In accordance with your advice, we have revised the figure caption of Figure 1 in the manuscript to better present the content of our work. Thank you again for your comments and suggestions. We believe that these suggestions have greatly improved the quality of this manuscript.

Comments 66: Remove the word groups from the legend. Just cells. Genes encoding proteins… and then you list the names of proteins. Genes and proteins are not the same. You cannot give the name of protein and tell that this is a gene. Names of genes and proteins are different, genes in italics.

Response 66: Thank you for your patient review of our manuscript and the valuable revision suggestions you’ve put forward. We fully agree with the suggestions you made and have revised the manuscript accordingly. Specifically, we’ve removed the word “groups” from the figure legends and changed the format of the gene names to italics to distinguish them from proteins. Once again, thank you for your professional suggestions on our manuscript, which will greatly enhance the quality of our work.

Comments 67: P8, L310 To investigate the effect of changes in the expression of MFN1 gene…. and antioxidant capacity in OE-Mfn1 and shMfn1 transfected cells.

Response 67: We are extremely grateful for the professional revision suggestions you provided for our manuscript. We accept your suggestions and have revised the manuscript accordingly.

Comments 68: P8, L312 that overexpression of MFN1

Response 68: We sincerely appreciate your support for our manuscript and the valuable revision suggestions you’ve offered. In accordance with your advice, we’ve revised the manuscript by changing the format of Mfn1 to italics.

Comments 69: P8, L313 while downregulation of MFN1

Response 69: Thank you very much for your support for our manuscript and the valuable revision suggestions you’ve provided. We have revised the manuscript according to your suggestions and changed the format of Mfn1 to italic.

Comments 70: P8, 314, 315 the levels of ATP and NADH in OE-Mfn1 cells…. shMfn1 cells

Response 70: We would like to express our sincere gratitude to you, the reviewer, for your support of our work and the revision suggestions you’ve put forward. We fully accept your suggestions and have revised the manuscript accordingly. Specifically, we’ve removed the word “expression” from the original text and changed “groups” to “cells”. Once again, thank you for your patient guidance on our manuscript.

Comments 71: P8, L318 SOD and CAT abbreviation should be explained in the Materials and Methods. Too late here.

Response 71: Thank you very much for reviewing our manuscript. We fully agree with your point that the abbreviations “SOD” and “CAT” should be explained in the “Materials and Methods” section. In accordance with your suggestion, we have revised the manuscript and provided definitions for “SOD” and “CAT” in the “Materials and Methods” part. Thank you again for your patient guidance.

Comments 72: P8, L319 when compared with control cells

Response 72: We are extremely grateful to you for reviewing our manuscript and offering valuable revision suggestions. We have accepted your suggestions and revised the manuscript accordingly.

Comments 73: P8, L320 What is the conclusion from these results. And so what?.

Response 73: We sincerely appreciate your support for our manuscript and the valuable suggestions you’ve provided. In accordance with your advice, we have added the conclusions drawn from these results to this part of the manuscript. Thank you again for your guidance, which has been of great help in improving the quality of our manuscript.

Figure 2

Comments 74: Title Changes in expression of MFN1 gene affect mitochondrial functions.

Response 74: We sincerely appreciate your patient guidance and the valuable revision suggestions you’ve put forward. We fully agree with your proposed change to the title of Figure 2. As per your advice, we have revised it to “Expression of Mfn1 gene affects mitochondrial functions.”

:

Comments 75: Panel A Relative level of JC1 fluorescence

Response 75: We sincerely appreciate your professional advice. We have accepted your suggestion and revised the title of the ordinate in Figure 2A to “Relative level of JC1 fluorescence”.

Comments 76: Panel E CAT activity.

Response 76: We are extremely grateful for your support of our manuscript and the valuable suggestions you’ve provided. We agree with your suggestions and have revised the title of the y-axis in Figure 2E to “CAT activity” according to your advice.

Comments 77:  Panel F SOD activity

Response 77: We sincerely appreciate your support for our manuscript and the invaluable suggestions you’ve offered. We concur with your suggestions and have modified the title of the vertical axis in Figure 2F to “SOD activity” as per your advice.

Comments 78: Legend, remove “respectively”, remove “groups”.

Response 78: We are extremely grateful for the valuable suggestions and comments you’ve provided regarding our manuscript. We have accepted your advice and made corresponding modifications. Specifically, we’ve revised the title of Figure 2 and removed the words “respectively” and “groups” from the caption of Figure 2.

Comments 79: P9, L332 Overexpression and downregulation of MFN1 gene affect expression of many genes. Title should be conclusion, rather, not analysis of something

Response 79: Thank you very much for reviewing our manuscript and offering valuable suggestions and comments. We fully accept your opinions and have revised the manuscript accordingly.

Comments 80: P9, L333 To find how changes in expression of MFN1 effect expression of nuclear genes the RNAseq analysis was performed in OEMfn1, shMfn1 and control cells.

Response 80: We sincerely appreciate your patient guidance. We have accepted your suggestions and revised the manuscript as you advised.

Comments 81: P10, L339 shMfn1 cells

Response 81: We truly appreciate your patient support for our work and the valuable revision suggestions you’ve provided. We’ve accepted your suggestions and revised the manuscript accordingly.

Comments 82: P10, L344 genes are enriched in apoptosis is a wrong statement. Is jargon.

Response 82: We are extremely grateful for your patient support of our work and the precious suggestions for revision you have put forward. We have accepted your suggestions and revised the manuscript according to them.

Figure 3

Comments 83: Panel A is not readable, too small.

Response 83: Thank you very much for your guidance. First of all, we sincerely apologize for the unclear Figure 3A, which affected its readability. In accordance with your suggestion, we enlarged Figure 3A to make it clearer and enhance its readability. Additionally, we have separately uploaded a high-definition version of the picture. Once again, we appreciate your professional guidance. We believe that these revisions have significantly improved the quality of our manuscript.

Comments 84: Legend, remove word “groups” , remove word “respectively”

Response 84: Thank you for reviewing our manuscript and providing valuable suggestions for revision. We highly agree with your suggestions and have revised the manuscript accordingly. We have removed the words “groups” and “respectively” from the caption of Figure 3.

Comments 85: P11, L372 249 metabolites show different levels on OE-Mfn1 cell.

Metabolites are not expressed. Genes are expressed. Correct also in other sites, L373, 375,380,

Response 85: We are extremely grateful to you for reviewing our manuscript and offering valuable revision suggestions. We fully agree with the issue you pointed out regarding our manuscript. For metabolites, “synthesized” should be used instead of “expressed”. In accordance with your suggestions, we have carefully checked the manuscript and revised all the unreasonable descriptions about metabolites in it.

Comments 86: P11, L384 The loss of Mfn1 affects the function of mitochondria

Response 86: Thank you very much for your professional guidance on our manuscript. We fully agree with your suggestions for revision and have revised the manuscript accordingly.

Comments 87: P11, L385 mitochondrial metabolism is not a byproduct. Is part of cellular metabolism.

Response 87: We are extremely grateful to you for reviewing our manuscript and offering valuable comments and suggestions. We agree with your opinion. There was a descriptive error in Line 385 of our manuscript. Mitochondrial metabolism is not a by-product; it is part of cellular activities. Based on your suggestions, we have revised the manuscript and changed it to “Mitochondrial energy metabolism is a key part of cellular metabolism and provides energy for cell growth.”

Figure 4

Comments 88: Panel A is too small. E and F is hardly readable.

Response 88: We are truly grateful to you for reviewing our manuscript and providing valuable comments and suggestions. In response to the issue you raised about Figures 4A, 4E, and 4F being too small and unclear, we have made revisions. We’ve enhanced the clarity of these images to improve their readability. Meanwhile, we have also separately submitted higher-resolution versions of these figures to ensure that they won’t affect your review of our manuscript. Once again, we appreciate the precious comments and suggestions you’ve offered, which have significantly enhanced the quality of our manuscript.

Comments 89: Legend: remove word “groups” and “respectively” in all legend. This groups are just cells.

Response 89: Thank you for the suggested revisions. We accept your comments and have revised the manuscript accordingly, removing the words “groups” and “respectively”.

Comments 90: P12, L409 cells for metabolomics analysis

Response 90: We truly appreciate the professional comments and suggestions you’ve provided for our manuscript. We accept your suggestions and have revised the manuscript accordingly.

Comments 91: P12, L414 MFN1 gene affects proteins methylation and acetylation.

Response 91: We sincerely appreciate your review of our manuscript and the valuable revision suggestions you’ve put forward. We accept your suggestions and have revised the manuscript accordingly.

Comments 92: P12, L419 cofactor for methylation of proteins

Response 92: We are extremely grateful to you for reviewing our manuscript and offering valuable revision comments and suggestions. We have accepted your suggestions and revised the manuscript accordingly.

Comments 93: P12, L420 MFN1 gene.

Response 93: We are grateful to you for your patient review of our manuscript and the valuable revision suggestions you’ve provided. We have accepted your suggestions and made corresponding revisions. Specifically, we have changed the format of Mfn1 to italics to represent the gene.

Comments 94: P12, L441To explore the effect of MFN1 gene on histone methylation and acetylation, the histone modifications in OE-Mfn1 and shMfn1 cells were assessed by Western Blot analysis.

Response 94: We sincerely thank you for your support regarding our manuscript and for providing valuable revision suggestions. We have accepted your suggestions and revised the manuscript accordingly.

Figure 5

Comments 95: Organization of panels is not correct Graphic representation of results should be in the same panel as respective result. I and J , K and L, M and N , O and P should be in one panel.

Response 95: We are extremely grateful for your support of our manuscript and the invaluable suggestions you’ve offered for its revision. We’ve taken your advice and made revisions to Figure 5, placing the graphical presentation of the results and the corresponding findings within the same section. Once again, we appreciate your professional guidance. We’re confident that these revisions will significantly enhance the quality of our manuscript.

Comments 96: Moreover, C and D, E and F, G and H should be in one panel.

Response 96: We truly appreciate the revision suggestions you provided for our manuscript. We have accepted your suggestions and revised Figure 5 by placing C and D, E and F, G and H in the same panel.

Comments 97: Legend, remove “groups” insert “cells”. Remove “respectively”.

Response 97: We are extremely grateful for your support of our manuscript and the valuable revision suggestions you’ve put forward. We have accepted your suggestions and made corresponding revisions to the manuscript. Specifically, we have deleted the word “respectively” and replaced “groups” with “cells”.

Discussion

Comments 98: P14, L476 by MFN1 gene and Mfn1 protein.

Response 98: Thank you very much for your guidance on our manuscript and the valuable suggestions you’ve put forward for its revision. We’ve accepted your suggestions and revised the manuscript accordingly.

Comments 99: P14, L477 Mutations in MFN1 gene lead

Response 99: We sincerely appreciate your professional guidance and the suggestions you’ve provided for our manuscript. We have accepted your suggestions and revised the manuscript accordingly.

Comments 100: P14, L479 expression of MFN1

Response 100: We truly appreciate your review of our manuscript and the valuable suggestions you’ve offered. In accordance with your suggestions, we’ve revised the manuscript by changing the format of Mfn1 to italics.

Comments 101: P14, L490 encoding Pcna, Cdk1… (Pcna, Cdk1 are proteins, not genes)

Response 101: We are extremely grateful for your guidance on our manuscript and the professional advice and suggestions you have provided. We have accepted your suggestions and revised the manuscript according to them. Specifically, we have changed the format of words like Pcna to italics. Thank you again for your professional guidance. We believe these revisions will significantly enhance the quality of our manuscript.

Comments 102: P14, L510 overproduction

Response 102: We sincerely appreciate your review of our manuscript and the valuable suggestions you’ve provided for its revision. We have accepted your suggestions and revised the manuscript accordingly.

Comments 103: P15, L515, 517, 523 MFN1 gene expression

Response 103: Thank you very much for your patient guidance on our manuscript and the valuable revision suggestions you’ve put forward. We have accepted your suggestions and revised the manuscript by changing the format of Mfn1 to italics.

Comments 104: P15, L530, L535 name of species in italics, C. elegans

Response 104: We are extremely grateful for the professional suggestions and advice you’ve provided regarding our manuscript. In line with your feedback, we’ve revised the manuscript by changing the format of the species names L530 and L535 to italics.

Comments 105: P15, L557 ShMfn1 cells ( not groups, groups of what?) Remove “groups” in all text, insert “cells”

Response 105: We are extremely grateful for your review of our manuscript and the valuable revision suggestions you put forward. We have accepted your suggestions and revised the manuscript accordingly. Specifically, we changed “groups” to “cells”. Moreover, we have carefully reviewed and revised the entire manuscript. We believe these revisions will significantly improve the quality of our manuscript. Thank you again for your professional guidance.

Comments 106: P16, L567 MFN1 gene indirectly regulates

Add significance of these findings and future perspectives, at the end of Discussion.

Response 106: We are extremely grateful for the valuable suggestions you’ve offered. In accordance with your advice, we’ve revised Line 157 of the manuscript, changing it to “Mfn1 gene indirectly regulates”. Moreover, we fully agree with your proposal to add the significance of the research findings and prospects for future studies at the end of the discussion section. Therefore, we’ve incorporated these elements at the end of the discussion section in the manuscript to make it more comprehensive. Thank you once again for your professional guidance and suggestions on our manuscript.

4. Response to Comments on the Quality of English Language

Point 1: English have to be corrected.

Wrong from of sentences in Methods. Must be in the past tense. Repost what was done.

Wrong sense using word “group”. Wrong sense using “respectively”.

Wrong genetic nomenclature.

Response 1: We are extremely grateful to you for the detailed and constructive comments on our manuscript. In response to the issues you pointed out, we have carefully revised and adjusted the manuscript. We meticulously checked and corrected the English expressions throughout the manuscript. Meanwhile, we invited a professional native English speaker to polish the paper to ensure the accuracy and fluency of the language. We have changed all the sentences in the “Methods” section to the past tense, clearly and accurately restating the experimental procedures and the work done, thus avoiding the problem of inconsistent tenses. We also corrected the incorrect use of “group” and “respectively” to further ensure the accuracy of word-usage. We have also checked and corrected all the gene names in the manuscript one by one to ensure that the gene naming complies with the norms. Thank you again for your professional advice. We believe that these revisions will significantly improve the quality of our manuscript and make it more in line with the publication requirements of the Cells journal.

Reviewer 3 Report (New Reviewer)

Comments and Suggestions for Authors

R = In this study, the authors investigated the role of Mfn1 overexpression or underexpression in bovine fibroblasts on mitochondrial metabolism and function and its consequences on epigenetic modifications in histones and cell proliferation. As expected, better overall conditions (cellular and mitochondrial) and antioxidant capacity were observed in the OE-Mfn1 group (↑ cell proliferation, mitochondrial membrane potential, ATP, NADH/NAD+, SOD, CAT; ↓ apoptosis, ROS) than in the CON and shMfn1 groups (with antagonistic behavior in almost all measured parameters). Subsequently, the authors analyzed α-KG, A-CoA and SMA concentrations, indicating that the (increased or decreased) expression of Mfn1 modulates the behavior of the mitochondrial metabolic profile. Finally, the authors analyzed variations related to histone epigenetic modifications by mRNA expression levels of demethylase, acetyltransferase, and deacetylase enzymes, finding interesting results. The authors conclude and suggest that mitochondrial dynamics (specifically Mfn1 and mitochondrial fusion) regulates the metabolic profile and, consequently, epigenetic modifications in histones, contributing substantially to the modulation of cell proliferation and apoptosis. In general, the authors conducted their research appropriately, with an acceptable experimental design, and reported their results at the different stages.

Below, I present some questions and recommendations/suggestions for the authors.

Questions:

  • Why did the authors use a knockdown model (which significantly decreases, but does not completely suppress, Mfn1 gene expression) and not a knockout model?
  • How would the authors ensure that these modifications are primarily driven by these metabolites? Were there other possible factors, or are they irrelevant?
  • In Figure 1J, it is possible to observe that Mfn1 overexpression results in higher mRNA levels of pro-apoptotic markers (Casp3, Casp9, and Cycs). However, in the analyses using Annexin V-FITC detection, the total apoptosis score shows that apoptosis is lower in this group, to what could the authors attribute this unexpected behavior?
  • In Figure 4H, an increased A-CoA concentration was observed in the OE-Mfn1 group, which is consistent with the increased mRNA expression of histone acetyltransferases (Figure 5E), suggesting that A-CoA is an important substrate for histone acetyltransferases and could probably increase their activity. However, why did the authors assess indirect epigenetic modifications on histones and not the amount of methylated DNA or acetylated histones? Does this represent an advantage?
  • Based on the previous question, the results show that A-CoA concentrations are also elevated in the shMfn1 group (Figure 4H), but histone acetyltransferase mRNA expression decreases (Figure 5F). How could the authors explain this unexpected behavior?

Major:

  • It is suggested to attach each of the images in a supplementary file that allows the reader to view them at high resolution; otherwise, it is not possible to properly analyze them.
  • In Figure 1H: verify statistical significance (** vs. ns) in the relative mRNA expression of Cdkn1a in CON vs. OE-Mfn1.
  • In Figure 2E, are the authors certain that this is the enzyme chloramphenicol acetyltransferase (CAT) or catalase?

Minor:

  • Verify that all abbreviations are defined in the text (CON, SOD, CAT, TCA, GO).
  • Line 112: A period must be added to conclude the sentence (“…cells per 1 mL. After…”).
  • Line 125: Since “real-time” and “quantitative” are equivalent terms, it is suggested that the title of section 2.5 “RNA Extraction and Real-Time Quantitative PCR” may be changed to “RNA Extraction and Quantitative Reverse Transcription PCR” or “RNA Extraction and Quantitative RT-PCR,” since “RT” refers to “reverse transcription” and not “real time.”
  • Line 185: 7 as superscript and the word “well” is missing. Line 547: H2O2.
  • Materials and methods related to Annexin V-FITC are not mentioned in the text.
  • Line 262: I assume the authors meant to refer to the expression of Mfn1 and not shMfn1.
  • Since enzyme function is typically reported based on its activity, it is suggested changing “content” to “activity” in the Materials and Methods section, even though they do report enzymatic activity in their results.
  • In Figure 2E, review and verify the significance between the CON vs. OE-Mfn1 (***?) groups on CAT levels.

Author Response

1. Summary

We sincerely appreciate your patient guidance and valuable suggestions during the review process. We have carefully considered your comments and provided detailed answers to the questions you raised. We have also verified the results with questionable significance and provided the corresponding P-values. Thank you again for your patient guidance. Your comments and suggestions have greatly helped us improve the quality of the manuscript, making it more suitable for publication in the journal of Cells. Once again, we are grateful for your support and recognition of our work.

2. Questions for General Evaluation

Reviewer’s Evaluation

Response and Revisions

Does the introduction provide sufficient background and include all relevant references?

Yes/Can be improved/Must be improved/Not applicable

Thank you very much to the reviewer for the comments and suggestions on our manuscript.

Are all the cited references relevant to the research?

Yes/Can be improved/Must be improved/Not applicable

Thank you to the reviewer for the comments and suggestions on this manuscript.

Is the research design appropriate?

Yes/Can be improved/Must be improved/Not applicable

Thank you very much to the reviewer for the comments and suggestions on our manuscript.

Are the methods adequately described?

Yes/Can be improved/Must be improved/Not applicable

Thank you very much to the reviewer for the comments and suggestions on our manuscript. We have revised the manuscript according to your feedback.

Are the results clearly presented?

Yes/Can be improved/Must be improved/Not applicable

Thank you very much to the reviewer for the comments and suggestions on our manuscript. We have revised the manuscript according to your feedback.

Are the conclusions supported by the results?

Yes/Can be improved/Must be improved/Not applicable

Thank you very much to the reviewer for the comments and suggestions on our manuscript.

Are all figures and tables clear and well-presented?

Yes/Can be improved/Must be improved/Not applicable

Thank you very much to the reviewer for the comments and suggestions on our manuscript. We have revised the manuscript according to your feedback.

3. Point-by-point response to Comments and Suggestions for Authors

Comments 1: Why did the authors use a knockdown model (which significantly decreases, but does not completely suppress, Mfn1 gene expression) and not a knockout model?

Response 1: We are extremely grateful for your constructive question. Our choice to use the knockdown model instead of the knockout model is primarily based on the following considerations:

Firstly, in many physiological and pathological conditions, gene expression levels often decrease partially rather than completely. Mfn1 plays a crucial role in processes such as mitochondrial fusion, energy metabolism, and the maintenance of cellular homeostasis in cells. Complete knockout of the Mfn1 gene may lead to severe functional impairment or even cell death, which significantly differs from the actual physiological and pathological states. In contrast, using the knockdown model can, to a certain extent, mimic the progressive decline of gene expression levels during the occurrence and development of diseases, making it closer to real-world biological scenarios. This helps us to deeply explore the effects of Mfn1 on cell functions and related biological processes under relatively mild expression changes.

Secondly, gene knockout may trigger complex compensatory mechanisms within the cell. When an important gene is completely knocked out, in order to maintain its survival and function, the cell may activate other related genes or signaling pathways to compensate for the function of the missing gene. This compensatory effect may mask the biological function of Mfn1 itself, making it difficult for us to accurately evaluate the direct impact of changes in Mfn1 gene expression on cell phenotypes and molecular mechanisms. In the knockdown model, only the expression of Mfn1 is partially reduced. Relatively speaking, the compensatory effect is weaker, allowing us to more directly observe the causal relationship between changes in Mfn1 expression levels and cell phenotypes.

Finally, from a technical perspective, the construction of the knockdown model is relatively simple, fast, and cost-effective. Technologies such as RNA interference (RNAi) can be conveniently used to achieve the knockdown of Mfn1 gene expression, and the operation can be carried out in various cell lines and experimental animal models, with high experimental reproducibility. In contrast, gene knockout technologies, such as the CRISPR/Cas9 gene-editing technology, although widely used in recent years, the process of constructing the knockout model is more complex. It requires strict verification of gene-editing efficiency and cell screening, and may introduce problems such as off-target effects, increasing the uncertainty and difficulty of the experiment.

Based on the above multiple considerations, although the knockdown model has certain limitations, we believe that at this stage of our study, it is a more appropriate experimental strategy that can provide us with valuable research results. Of course, we also recognize that the knockout model has unique advantages in some aspects. In subsequent studies, we will consider combining the knockout model to further validate and expand our findings.

Thank you again for your valuable comments and suggestions.

Comments 2: How would the authors ensure that these modifications are primarily driven by these metabolites? Were there other possible factors, or are they irrelevant?

Response 2: We sincerely appreciate your comments and suggestions on our manuscript. We would like to address your question regarding how we ensure that these changes are mainly driven by metabolites. At the beginning of this study, through collecting and reviewing a large number of literatures, we have clearly recognized that the organism’s own metabolic status can affect epigenetics. Metabolites, as substrates for modifying enzymes, are involved in the epigenetic modifications of histones, DNA, and RNA. For example, S-adenosyl-methionine (SAM) serves as a methyl donor for DNA and histone methylation modifications and is also an intermediate product of methionine metabolism; Acetyl-CoA provides an acetyl donor for histone acetylation and is also involved in metabolic processes such as fatty acid oxidation and the tricarboxylic acid (TCA) cycle; α-ketoglutarate (α-KG) is a substrate for DNA and histone demethylases and is also a metabolic product of the TCA cycle. These metabolites are not only cofactors or donors for epigenetic modifications but also can regulate the epigenetic state by affecting the activity of modifying enzymes. For instance, succinate and fumarate can inhibit the activity of DNA and histone demethyltransferases; the expression levels of NAD⁺/NADH affect the activity of class III histone deacetylases [1-3]. Therefore, changes in the content of metabolites in cells can lead to changes in the genomic expression of epigenetic modifying enzymes, which in turn affect chromatin epigenetic modifications. In this study, we also observed regular changes in the levels of metabolites and epigenetic modifications. Based on these results, we speculate that the changes in the levels of epigenetic modifications are caused by the changes in the content of metabolites, which further affect the expression levels of key genes that influence cell fate. Of course, the issue you pointed out will also be the focus of our in-depth research in the future. Regarding your question about whether there are other possible influencing factors, during the experimental design stage, we also conducted an extensive literature review. Environmental factors such as experimental temperature, pH value, and nutritional level can also have a certain impact on the modification levels. During the experiment, we strictly controlled these experimental conditions to make them as consistent as possible, so as to minimize the interference of environmental factors. For the reasons mentioned above, we believe that the observed changes in the modification levels are mainly caused by the changes in the metabolite levels, and the influence of other factors can be ignored or only plays a secondary role. Thank you again for your professional comments and suggestions.

[1] Mentch S J, Mehrmohamadi M, Huang L, et al. Histone methylation dynamics and gene regulation occur through the sensing of one-Carbon metabolism[J]. Cell Metabolism, 2015, 22(5):861-873.

[2] Shyh-Chang N, Locasale J W, Lyssiotis C A, et al. Influence of threonine metabolism on S-adenosylmethionine and histone methylation[J]. Science, 2013, 339(6116):222-226.

[3] Sutendra G, Kinnaird A, Dromparis P, et al. A nuclear pyruvate dehydrogenase complex is important for the generation of acetyl-CoA and histone acetylation[J]. Cell, 2014, 158(1):84-97.

Comments 3: In Figure 1J, it is possible to observe that Mfn1 overexpression results in higher mRNA levels of pro-apoptotic markers (Casp3, Casp9, and Cycs). However, in the analyses using Annexin V-FITC detection, the total apoptosis score shows that apoptosis is lower in this group, to what could the authors attribute this unexpected behavior?

Response 3: Thank you very much for raising this question. Regarding the unexpected phenomenon in Figure 1J, where the overexpression of Mfn1 led to an increase in the mRNA levels of pro-apoptotic markers (Casp3, Casp9, and Cycs), but the total apoptosis rate of cells in this group decreased as shown by Annexin V-FITC detection, we speculate that there may be the following reasons:

1. Differences between transcriptional and protein levels

An increase in mRNA levels does not necessarily directly translate into an increase in the corresponding protein levels. There are multiple regulatory mechanisms in cells that affect the translation efficiency of mRNA and the stability of proteins. For example, there may be some translation repressors that are activated when Mfn1 is overexpressed. Although the transcription of the mRNAs of pro-apoptotic markers increases, the translation process is inhibited, resulting in no significant increase in the amount of pro-apoptotic proteins that ultimately exert their functions. As a result, the level of cell apoptosis does not increase as the mRNA levels change.

2. Adaptive responses of cells

The overexpression of Mfn1 may trigger some adaptive protective mechanisms in cells. When cells detect an increase in pro-apoptotic signals, they may initiate a series of defense mechanisms to counteract apoptosis. For instance, the anti-apoptotic signaling pathways in cells may be activated, leading to the production of some anti-apoptotic proteins (such as certain members of the Bcl-2 family). These anti-apoptotic proteins can interact with pro-apoptotic proteins and inhibit their activities, thereby reducing the apoptosis rate of cells. Although the mRNA levels of pro-apoptotic markers increase, the enhancement of the anti-apoptotic mechanism offsets the effect of pro-apoptotic signals to a certain extent.

3. Limitations of experimental detection methods

The Annexin V-FITC detection mainly assesses cell apoptosis based on the externalization of phosphatidylserine on the cell membrane, which is an early characteristic of apoptosis. However, cell apoptosis is a complex multi-stage process. There may be some cells in the late stage of apoptosis that have not been detected by Annexin V-FITC, or some changes in apoptosis-related signaling pathways may not be accurately reflected by this detection method. In addition, operational factors during the experiment (such as cell handling and staining time) may also have a certain impact on the detection results.

Thank you again for your meticulous review and the valuable questions you raised.

Comments 4: In Figure 4H, an increased A-CoA concentration was observed in the OE-Mfn1 group, which is consistent with the increased mRNA expression of histone acetyltransferases (Figure 5E), suggesting that A-CoA is an important substrate for histone acetyltransferases and could probably increase their activity. However, why did the authors assess indirect epigenetic modifications on histones and not the amount of methylated DNA or acetylated histones? Does this represent an advantage?

Response 4: Thank you very much for raising such professional questions. The following is a detailed response to your query about why we chose to evaluate indirect epigenetic modifications rather than directly detecting methylated DNA or acetylated histones. Firstly, this study focuses on exploring the intracellular functions of Mfn1 and its associations with energy metabolism and epigenetic regulation. Our interest lies in how the overexpression and silencing of Mfn1 indirectly affect the activities of epigenetic-related enzymes by influencing the concentrations of metabolites, and further impact the overall physiological state of cells. Evaluating indirect epigenetic modifications can provide a more comprehensive picture of the dynamic changes in this metabolic-epigenetic regulatory network, rather than being limited to the amounts of specific methylated DNA or acetylated histones. Secondly, our experimental design is a step-by-step process. In the preliminary experiments, we have already observed the effects of Mfn1 overexpression on metabolites and the mRNA expression of related enzymes. Evaluating indirect epigenetic modifications is a logical continuation of this research, which can better link metabolic changes with epigenetic regulation. By detecting the indicators of indirect epigenetic modifications related to metabolism, we can more clearly reveal the cascade reactions triggered by Mfn1 overexpression, providing clues for a better understanding of the complex regulatory mechanisms within cells. Epigenetic regulation within cells is a dynamic process influenced by various physiological and pathological factors. Changes in indirect epigenetic modifications can better reflect the actual regulatory status of cells under physiological conditions. Compared with simply detecting the amounts of methylated DNA or acetylated histones, indirect evaluation can better capture the dynamic balance between intracellular metabolism and epigenetics, offering more valuable information for studying cellular physiological functions and disease - onset mechanisms. Meanwhile, we understand that directly detecting the amounts of methylated DNA or acetylated histones is also of great significance. In subsequent studies, we will consider incorporating these detections to more comprehensively reveal the impact of Mfn1 overexpression on epigenetic regulation. Thank you again for your suggestions and guidance.

Comments 5: Based on the previous question, the results show that A-CoA concentrations are also elevated in the shMfn1 group (Figure 4H), but histone acetyltransferase mRNA expression decreases (Figure 5F). How could the authors explain this unexpected behavior?

Response 5: Thank you very much again for raising such in-depth and professional questions. Regarding the unexpected phenomenon of the increased concentration of A-CoA but decreased mRNA expression of histone acetyltransferase in the shMfn1 group, we have conducted an in-depth analysis and speculated that the following reasons may account for it:

1. Metabolic compensation mechanism

shMfn1 may interfere with the normal mitochondrial fusion process within cells, leading to a certain degree of mitochondrial dysfunction. To cope with this mitochondrial impairment, cells may initiate a series of metabolic compensation mechanisms. A-CoA is a key intermediate product in intracellular energy metabolism and biosynthesis. When mitochondrial function is damaged, cells may up-regulate certain metabolic pathways (such as fatty acid oxidation and glycolysis) to increase the production of A-CoA, so as to meet the energy requirements and basic metabolic functions of cells. However, this metabolic compensation may come at the expense of other physiological processes. The expression and activity of histone acetyltransferases are closely related to the energy status and metabolic environment of cells. In the shMfn1 cells, although the concentration of A-CoA has increased, the overall intracellular energy metabolism and signal transduction network may have changed, making it impossible for cells to effectively utilize this extra A-CoA to promote the expression of histone acetyltransferases.

2. Post-transcriptional regulatory mechanism

The regulation of gene expression within cells is a complex process. In addition to transcriptional regulation, there are also post-transcriptional regulatory mechanisms. shMfn1 may affect the regulatory factors related to the stability and translation efficiency of histone acetyltransferase mRNA. The expression or activity of certain microRNAs or RNA-binding proteins may have changed under the action of shMfn1, resulting in the histone acetyltransferase mRNA being more prone to degradation or its translation being inhibited. Although the increased concentration of A-CoA in cells provides sufficient substrates for histone acetylation, the synthesis of histone acetyltransferases is still restricted due to the post-transcriptional regulatory mechanism.

3. Cellular stress response

shMfn1 may trigger a cellular stress response. Changes in mitochondrial function can lead to an increase in the intracellular reactive oxygen species (ROS) level and activate stress signals such as endoplasmic reticulum stress. These stress signals may affect the intracellular epigenetic regulatory network, leading to the inhibition of histone acetyltransferase expression. At the same time, cells may prioritize the limited resources to cope with stress and reduce the investment in the expression of genes related to epigenetic regulation.

Thank you again for your valuable comments and suggestions.

Major:

Comments 6: It is suggested to attach each of the images in a supplementary file that allows the reader to view them at high resolution; otherwise, it is not possible to properly analyze them.

Response 6: We sincerely appreciate the professional suggestions you’ve provided. We fully endorse your advice and have arranged each image separately into supplementary files as you recommended. During the arrangement process, we will ensure that the resolution of the images meets the requirements for high - quality presentation, enabling readers to conduct detailed and accurate analyses of the images. We will submit these supplementary files along with the revised article to enhance the overall readability and professionalism of the paper. Thank you once again for your guidance and reminders.

Comments 7: In Figure 1H: verify statistical significance (** vs. ns) in the relative mRNA expression of Cdkn1a in CON vs. OE-Mfn1.

Response 7: We sincerely appreciate your review of our manuscript and the valuable revision suggestions you’ve provided. Regarding the significance issue of Cdkn1a between CON and OE-Mfn1 in Figure 1H that you raised, we re-checked the original data and conducted further calculations. The results show that the p-value of Cdkn1A between CON and OE-Mfn1 is 0.001242358. Therefore, the significance should be marked with **. Thank you again for your professional guidance.

Comments 8: In Figure 2E, are the authors certain that this is the enzyme chloramphenicol acetyltransferase (CAT) or catalase?

Response 8: We sincerely appreciate your meticulous guidance and the valuable comments and suggestions you’ve provided for our manuscript. In this study, “CAT” refers to “catalase”. Catalase is distributed in all tissues of all known animals, especially in the liver. It is an enzyme that catalyzes the decomposition of hydrogen peroxide into oxygen and water and is located in the peroxisomes of cells. While mitochondria produce energy through oxidative phosphorylation, they also generate superoxide and reactive oxygen species (ROS). Since the membrane is impermeable to superoxide, superoxide can be easily dismutated by mitochondrial superoxide dismutase (SOD) into hydrogen peroxide (H₂O₂) and O₂, converting it into non - free radicals, which can cause oxidative damage and ultimately initiate the apoptotic pathway. At the same time, this study found that changes in the expression of the Mfn1 gene can affect the content of ROS in cells. Based on the above reasons, we detected the expression level of CAT.

Minor:

Comments 9: Verify that all abbreviations are defined in the text (CON, SOD, CAT, TCA, GO).

Response 9: We are extremely grateful for the valuable comments and suggestions you’ve offered regarding our manuscript. In accordance with your advice, we carefully reviewed the manuscript and checked and defined the abbreviations used therein. Specifically: “CON” stands for “control”, as mentioned on Line 198 of the manuscript. “SOD” is defined as “superoxide dismutase”, which can be found on Line 150 of the manuscript. “CAT” is defined as “chloramphenicol acetyltransferase”, as stated in the part on Line 154 of the manuscript. “TCA” represents “tricarboxylic acid”, and the relevant information is on Line 491 of the manuscript. “GO” refers to “Gene Ontology”, as shown in the part on Line 208 of the manuscript.

Comments 10: Line 112: A period must be added to conclude the sentence (“…cells per 1 mL. After…”).

Response 10: We sincerely appreciate your review of our manuscript and the valuable comments and suggestions you’ve put forward. Your suggestions have been of great assistance in improving the quality of our manuscript. We have revised the manuscript in accordance with your advice. Thank you again for your comments.

Comments 11: Line 125: Since “real-time” and “quantitative” are equivalent terms, it is suggested that the title of section 2.5 “RNA Extraction and Real-Time Quantitative PCR” may be changed to “RNA Extraction and Quantitative Reverse Transcription PCR” or “RNA Extraction and Quantitative RT-PCR,” since “RT” refers to “reverse transcription” and not “real time.”

Response 11: We are extremely grateful for the professional suggestions you’ve provided regarding our manuscript. We fully agree with your consideration of the accuracy of terminology. We whole - heartedly endorse your proposal to change the title of Section 2.5 to “RNA Extraction and Quantitative RT-PCR” to ensure that the title is more accurate and clear.

Comments 12: Line 185: 7 as superscript and the word “well” is missing. Line 547: H2O2.

Response 12: Thank you very much for taking the time out of your busy schedule to review our manuscript. We have revised the corresponding parts of the manuscript as per your suggestions.

Comments 13: Materials and methods related to Annexin V-FITC are not mentioned in the text.

Response 13: Thank you very much for your meticulous guidance on our manuscript. We sincerely apologize for the oversights during the manuscript writing process. According to your suggestions, we have added the experimental methods and procedures for Annexin V-FITC in the “Materials and methods” section of the manuscript.

Comments 14: Line 262: I assume the authors meant to refer to the expression of Mfn1 and not shMfn1.

Response 14: We are extremely grateful for the valuable comments and suggestions you’ve offered on our manuscript. We highly agree with your suggestions and have revised the corresponding parts of the manuscript according to your advice.

Comments 15: Since enzyme function is typically reported based on its activity, it is suggested changing “content” to “activity” in the Materials and Methods section, even though they do report enzymatic activity in their results.

Response 15: We are extremely grateful for the valuable suggestions you have put forward. We believe your proposals are very reasonable. We accept your suggestions and, in accordance with them, carefully reviewed the “Materials and Methods” section. We replaced all instances of “content” with “activity” to ensure consistency between this section and the results section, making the article more rigorous in logic and expression. Thank you again for your professional guidance.

Comments 16: In Figure 2E, review and verify the significance between the CON vs. OE-Mfn1 (***?) groups on CAT levels.

Response 16: We are extremely grateful to you for raising valuable questions about our manuscript. In response to your queries, we conducted further analysis of the original data. The results show that the P-value for the CAT level between the CON and OE-Mfn1 cells is 0.000122713, which indeed corresponds to “***”. Once again, we appreciate your guidance, as it has been of great help in improving the quality of our manuscript.

4. Response to Comments on the Quality of English Language

Point 1: The English is fine and does not require any improvement.

Response 1: We would like to express our sincere gratitude for your thorough review of our manuscript and your high recognition of its English expression. Your positive feedback has been a great source of inspiration for us. Although you have affirmed the language quality of our manuscript, we still attach great importance to the scientific rigor and accuracy of the writing. We will carefully consider all the other comments and suggestions you’ve provided and make revisions accordingly. We guarantee that during the subsequent revision process, we will maintain the readability of the manuscript to ensure it better meets the publication requirements of the journal. Once again, we sincerely thank you for your approval and support of our work.

Round 2

Reviewer 1 Report (Previous Reviewer 2)

Comments and Suggestions for Authors

Most of my comments have been addressed adequately. However, regarding the missing appropriate controls (point 1) no remarks have been transfered into the manuscript. That could have been included at least in the study limitation section.

Moreover, the newly added method descriptions should be language edited: Please do not write what is written in the manual, but write what you did: ... we added ...,  ...we mixed...

Comments on the Quality of English Language

Correct the newly introduced method parts.

Author Response

Response to Reviewer 1 Comments

1. Summary

We sincerely thank you for reviewing our manuscript again and providing us with patient guidance and valuable suggestions. We deeply apologize for the oversights during the first-round revision of the manuscript. Based on your suggestions, we have comprehensively checked and revised the manuscript. Specifically, we have modified the “Materials and Methods” section, using the first-person perspective. Meanwhile, we have explained the plasmid selection in the “Research Limitations” section of the manuscript. Thank you again for your patient guidance. Your suggestions and advice have significantly helped us improve the quality of our manuscript, making it more suitable for publication in the journal of Cells. We are grateful once again for your support and recognition of our work.

2. Questions for General Evaluation

Reviewer’s Evaluation

Response and Revisions

Does the introduction provide sufficient background and include all relevant references?

Yes/Can be improved/Must be improved/Not applicable

Thank you very much to the reviewer for the comments and suggestions on our manuscript. We have revised the manuscript according to your feedback.

Are all the cited references relevant to the research?

Yes/Can be improved/Must be improved/Not applicable

Thank you to the reviewer for the comments and suggestions on this manuscript.

Is the research design appropriate?

Yes/Can be improved/Must be improved/Not applicable

Thank you very much to the reviewer for the comments and suggestions on our manuscript. We have revised the manuscript according to your feedback.

Are the methods adequately described?

Yes/Can be improved/Must be improved/Not applicable

Thank you very much to the reviewer for the comments and suggestions on our manuscript. We have revised the manuscript according to your feedback.

Are the results clearly presented?

Yes/Can be improved/Must be improved/Not applicable

Thank you very much to the reviewer for the comments and suggestions on our manuscript. We have revised the manuscript according to your feedback.

Are the conclusions supported by the results?

Yes/Can be improved/Must be improved/Not applicable

Thank you very much to the reviewer for the comments and suggestions on our manuscript. We have revised the manuscript according to your feedback.

Are all figures and tables clear and well-presented?

Yes/Can be improved/Must be improved/Not applicable

Thank you very much to the reviewer for the comments and suggestions on our manuscript. We have revised the manuscript according to your feedback.

3. Point-by-point response to Comments and Suggestions for Authors

Comments 1: Most of my comments have been addressed adequately. However, regarding the missing appropriate controls (point 1) no remarks have been transfered into the manuscript. That could have been included at least in the study limitation section.

Response 1: We sincerely appreciate you carefully reviewing our manuscript once again. We apologize deeply for not addressing the issue of the lack of appropriate controls (point 1) in the manuscript as you pointed out. Based on your feedback, we have revised the manuscript. In the section on research limitations, we have elaborated on the situation of the lack of appropriate controls in the study and clearly indicated that this may have a certain impact on the accuracy and reliability of the research. Meanwhile, we have also analyzed the potential biases that the lack of controls may cause to the research results and mentioned measures such as conducting supplementary control experiments that can be taken in future research to make improvements. Thank you again for your professional suggestions, which have greatly improved the quality of our manuscript.

Comments 2: Moreover, the newly added method descriptions should be language edited: Please do not write what is written in the manual, but write what you did: ... we added ...,  ...we mixed...

Response 2: We sincerely appreciate your review of our manuscript and the valuable comments and suggestions you’ve provided. The issue you pointed out is of great significance in enhancing the quality of our manuscript. In accordance with your suggestions, we will conduct a comprehensive review and revision of the newly - added method descriptions. We’ll ensure that the language used meets the requirements, making the method descriptions more accurate, clear, and closely aligned with the actual research process. Once again, we’re grateful for your professional advice.

4. Response to Comments on the Quality of English Language

Point 1: The English could be improved to more clearly express the research.

Response 1: Thank you very much for your valuable suggestions. We have revised the manuscript according to your comments. In the “Materials and Methods” section, we modified the newly added methods. We used the first - person perspective to describe in detail the actual operations we carried out. Additionally, we invited a professional English - native speaker to review the manuscript to improve the overall quality of our English description. Moreover, all the authors have reviewed the revised version of the manuscript. Thank you again for your professional guidance.

Reviewer 2 Report (New Reviewer)

Comments and Suggestions for Authors

Xu et al., 2025

Authors generally agree with indicated corrections but did not implement them in whole text.

Authors defined the name Mfn1 (italics) for the gene encoding Mfn1 protein and should keep it that way in whole text.

All Authors should read again the text carefully and correct all mistakes. Take your time.

Title

P1, l2  by Mfn1 gene

Abstract

P1, L16 Mfn1 and Mfn2 gene expression

L18 Mfn1 expression (or Mfn1 gene expression)

L19 Mfn1 gene using

L25 Mfn1 gene expression

Introduction

P2, L74 silencing of Mfn1  (Authors agreed that this was not knockdown of Mfn1 what they did in their experiments)

P2, L75 Mfn1

Materials and methods

P2, L83 These cells were used as control cells (CON) in further study. (Better define here what is the abbreviation and the name of cells CON)

P2,L86 10 cm

P2,L90 constructed

P2,L93 Transfection experiments were performed three times and transfection efficiency was statistically analyzed.

P2, L94  Transfected bovine fetus fibroblasts overexpressing Mfn1 gene (OE-Mfn1) and fibroblasts with silenced Mfn1 gene (shMfn1) were used in further study.

P3, L103 removed and

P3, L104 Samples were gently shake on a microplate

P3, L105 10 min

P3, L124 Annexin V-FITC detection of cell apoptosis ability

P3, L126-134  space between lines is different in this part of the text

P4, L154 ATP content assay (ATP is a chemical compound, not an enzyme, like SOD and CAT) ATP content, SOD activity, CAT activity, NADH content. Authors should know the difference.

P4, L178 4oCP5, L202 Each sample of 1x107 cells

P5, L204 30 min

P4, L207 10 min

P5,L209 gel concentration of 12% was selected for electrophoresis.

P5, L213 remove “mouse”. It is a repetition what was in previous line (Mouse … is a mouse)_

P5, L216 , raised…….. anti-histone (remove “this antibody”, is too many times)

P5, L217, 219., 221,224 remove: this antibody is

P5, L235 10 min…Than the developing solutions A and B were added

P5, L238 Each experiment was repeated three times

P6, L247-254 all sentences are in wrong grammatical form in this part of the text

P6, L257 and shMfn1 cell cultures.

P6, L262 Principal Component Analysis (PCA) ( here for the first time abbreviation was introduced) Do not start the sentence with brackets (..)

P6, L279 differentially produced metabolites

Results

P6, L293 expression of Mfn1 and the levels of Mfn1

P7, L296 cells (Figure…)

P7, L300 control cells

P7, L305 following genes: … (if genes are listed, in italics) than following genes : is ok

P7, L6 Cdk2

P8, L319, 321 Mfn1 gene

P9, L344, Mfn1 gene

P9, L345 After the silencing of the Mfn1 gene

P9, L353 CAT activity

P10,L354 SOD activity

P10,L 367 enriched genes involved in cell growth

P10.L370 enriched in genes involved in cell proliferation

P10,L378 enriched in genes involved in cAMP, ..

P11, L389 Metabolome analysis of OE-Mfn1 and ShMfn1 cells (remove sequencing)

P12, L433 remove “modification”

P13, L460 Mfn1 gene

P13, L465 both of them were

Discussion

P15, L491, 492, 493, 494, 495 496 498 Mfn1

P15, L527 SOD activity

P15, L529 silencing of Mfn1

P16, L535 differentially produced

Xu et al., 2025

Authors generally agree with indicated corrections but did not implement them in whole text.

Authors defined the name Mfn1 (italics) for the gene encoding Mfn1 protein and should keep it that way in whole text.

All Authors should read again the text carefully and correct all mistakes. Take your time.

Title

P1, l2  by Mfn1 gene

Abstract

P1, L16 Mfn1 and Mfn2 gene expression

L18 Mfn1 expression (or Mfn1 gene expression)

L19 Mfn1 gene using

L25 Mfn1 gene expression

Introduction

P2, L74 silencing of Mfn1  (Authors agreed that this was not knockdown of Mfn1 what they did in their experiments)

P2, L75 Mfn1

Materials and methods

P2, L83 These cells were used as control cells (CON) in further study. (Better define here what is the abbreviation and the name of cells CON)

P2,L86 10 cm

P2,L90 constructed

P2,L93 Transfection experiments were performed three times and transfection efficiency was statistically analyzed.

P2, L94  Transfected bovine fetus fibroblasts overexpressing Mfn1 gene (OE-Mfn1) and fibroblasts with silenced Mfn1 gene (shMfn1) were used in further study.

P3, L103 removed and

P3, L104 Samples were gently shake on a microplate

P3, L105 10 min

P3, L124 Annexin V-FITC detection of cell apoptosis ability

P3, L126-134  space between lines is different in this part of the text

P4, L154 ATP content assay (ATP is a chemical compound, not an enzyme, like SOD and CAT) ATP content, SOD activity, CAT activity, NADH content. Authors should know the difference.

P4, L178 4oCP5, L202 Each sample of 1x107 cells

P5, L204 30 min

P4, L207 10 min

P5,L209 gel concentration of 12% was selected for electrophoresis.

P5, L213 remove “mouse”. It is a repetition what was in previous line (Mouse … is a mouse)_

P5, L216 , raised…….. anti-histone (remove “this antibody”, is too many times)

P5, L217, 219., 221,224 remove: this antibody is

P5, L235 10 min…Than the developing solutions A and B were added

P5, L238 Each experiment was repeated three times

P6, L247-254 all sentences are in wrong grammatical form in this part of the text

P6, L257 and shMfn1 cell cultures.

P6, L262 Principal Component Analysis (PCA) ( here for the first time abbreviation was introduced) Do not start the sentence with brackets (..)

P6, L279 differentially produced metabolites

Results

P6, L293 expression of Mfn1 and the levels of Mfn1

P7, L296 cells (Figure…)

P7, L300 control cells

P7, L305 following genes: … (if genes are listed, in italics) than following genes : is ok

P7, L6 Cdk2

P8, L319, 321 Mfn1 gene

P9, L344, Mfn1 gene

P9, L345 After the silencing of the Mfn1 gene

P9, L353 CAT activity

P10,L354 SOD activity

P10,L 367 enriched genes involved in cell growth

P10.L370 enriched in genes involved in cell proliferation

P10,L378 enriched in genes involved in cAMP, ..

P11, L389 Metabolome analysis of OE-Mfn1 and ShMfn1 cells (remove sequencing)

P12, L433 remove “modification”

P13, L460 Mfn1 gene

P13, L465 both of them were

Discussion

P15, L491, 492, 493, 494, 495 496 498 Mfn1

P15, L527 SOD activity

P15, L529 silencing of Mfn1

P16, L535 differentially produced

Xu et al., 2025

Authors generally agree with indicated corrections but did not implement them in whole text.

Authors defined the name Mfn1 (italics) for the gene encoding Mfn1 protein and should keep it that way in whole text.

All Authors should read again the text carefully and correct all mistakes. Take your time.

Title

P1, l2  by Mfn1 gene

Abstract

P1, L16 Mfn1 and Mfn2 gene expression

L18 Mfn1 expression (or Mfn1 gene expression)

L19 Mfn1 gene using

L25 Mfn1 gene expression

Introduction

P2, L74 silencing of Mfn1  (Authors agreed that this was not knockdown of Mfn1 what they did in their experiments)

P2, L75 Mfn1

Materials and methods

P2, L83 These cells were used as control cells (CON) in further study. (Better define here what is the abbreviation and the name of cells CON)

P2,L86 10 cm

P2,L90 constructed

P2,L93 Transfection experiments were performed three times and transfection efficiency was statistically analyzed.

P2, L94  Transfected bovine fetus fibroblasts overexpressing Mfn1 gene (OE-Mfn1) and fibroblasts with silenced Mfn1 gene (shMfn1) were used in further study.

P3, L103 removed and

P3, L104 Samples were gently shake on a microplate

P3, L105 10 min

P3, L124 Annexin V-FITC detection of cell apoptosis ability

P3, L126-134  space between lines is different in this part of the text

P4, L154 ATP content assay (ATP is a chemical compound, not an enzyme, like SOD and CAT) ATP content, SOD activity, CAT activity, NADH content. Authors should know the difference.

P4, L178 4oCP5, L202 Each sample of 1x107 cells

P5, L204 30 min

P4, L207 10 min

P5,L209 gel concentration of 12% was selected for electrophoresis.

P5, L213 remove “mouse”. It is a repetition what was in previous line (Mouse … is a mouse)_

P5, L216 , raised…….. anti-histone (remove “this antibody”, is too many times)

P5, L217, 219., 221,224 remove: this antibody is

P5, L235 10 min…Than the developing solutions A and B were added

P5, L238 Each experiment was repeated three times

P6, L247-254 all sentences are in wrong grammatical form in this part of the text

P6, L257 and shMfn1 cell cultures.

P6, L262 Principal Component Analysis (PCA) ( here for the first time abbreviation was introduced) Do not start the sentence with brackets (..)

P6, L279 differentially produced metabolites

Results

P6, L293 expression of Mfn1 and the levels of Mfn1

P7, L296 cells (Figure…)

P7, L300 control cells

P7, L305 following genes: … (if genes are listed, in italics) than following genes : is ok

P7, L6 Cdk2

P8, L319, 321 Mfn1 gene

P9, L344, Mfn1 gene

P9, L345 After the silencing of the Mfn1 gene

P9, L353 CAT activity

P10,L354 SOD activity

P10,L 367 enriched genes involved in cell growth

P10.L370 enriched in genes involved in cell proliferation

P10,L378 enriched in genes involved in cAMP, ..

P11, L389 Metabolome analysis of OE-Mfn1 and ShMfn1 cells (remove sequencing)

P12, L433 remove “modification”

P13, L460 Mfn1 gene

P13, L465 both of them were

Discussion

P15, L491, 492, 493, 494, 495 496 498 Mfn1

P15, L527 SOD activity

P15, L529 silencing of Mfn1

P16, L535 differentially produced

Comments on the Quality of English Language

Still needs corrections. Some parts in the Matrials and Methods are not in correct grammatical form. Should be past tense. See above.

Author Response

Response to Reviewer 2 Comments

1. Summary

We sincerely thank you for your patient guidance and valuable suggestions during the review process. First of all, we deeply apologize for the oversights during the manuscript revision. Based on your suggestions, we have comprehensively revised the manuscript again.

Specifically, we have carefully examined the entire manuscript, further verified and standardized the use of gene and protein names in the manuscript to make the presentation more scientific and standardized. We have also corrected the tenses in the “Materials and Methods” section to ensure that the correct tenses are used to reflect our research content.

In addition, we have invited a professional whose native language is English to review the language of the manuscript, making our English expressions more scientific and standardized to better present our research content. Moreover, all the authors have reviewed the manuscript before resubmitting the revised version.

Thank you again for your patient guidance. Your comments and suggestions have greatly helped us improve the quality of the manuscript, making it more suitable for publication in the journal Cells. We are grateful once again for your support and recognition of our work.

2. Questions for General Evaluation

Reviewer’s Evaluation

Response and Revisions

Does the introduction provide sufficient background and include all relevant references?

Yes/Can be improved/Must be improved/Not applicable

Thank you to the reviewer for the comments and suggestions on this manuscript.

Are all the cited references relevant to the research?

Yes/Can be improved/Must be improved/Not applicable

Thank you to the reviewer for the comments and suggestions on this manuscript.

Is the research design appropriate?

Yes/Can be improved/Must be improved/Not applicable

Thank you very much to the reviewer for the comments and suggestions on our manuscript.

Are the methods adequately described?

Yes/Can be improved/Must be improved/Not applicable

Thank you to the reviewer for the comments and suggestions on this manuscript.

Are the results clearly presented?

Yes/Can be improved/Must be improved/Not applicable

Thank you very much to the reviewer for the comments and suggestions on our manuscript. We have revised the manuscript according to your feedback.

Are the conclusions supported by the results?

Yes/Can be improved/Must be improved/Not applicable

Thank you to the reviewer for the comments and suggestions on this manuscript

Are all figures and tables clear and well-presented?

Yes/Can be improved/Must be improved/Not applicable

Thank you to the reviewer for the comments and suggestions on this manuscript

3. Point-by-point response to Comments and Suggestions for Authors

General comments

Comments 1: Authors generally agree with indicated corrections but did not implement them in whole text.

Response 1: We are extremely grateful to you for pointing out the issues in our manuscript during the revision process. We sincerely apologize for our carelessness and oversight, which led to the failure to implement the corrections you suggested throughout the manuscript in a timely manner. We fully agree with the revision suggestions you put forward for our manuscript. In accordance with your advice, we have conducted a detailed and comprehensive check of the entire manuscript, with a focus on making thorough revisions to ensure that the quality of the manuscript meets a higher standard. Once again, thank you for your professional guidance on our manuscript.

Comments 2: Authors defined the name Mfn1 (italics) for the gene encoding Mfn1 protein and should keep it that way in whole text.

Response 2: We are deeply grateful to you for your meticulous review of our manuscript and the invaluable revision suggestions you’ve offered. The issues you pointed out regarding the naming conventions of genes and proteins in our manuscript have played a crucial role in guiding us to further improve the quality of our work. Indeed, we had defined in the manuscript that the gene encoding Mfn1 should be presented in italic format. We sincerely apologize for failing to maintain this convention consistently throughout the manuscript during the revision process. This oversight was due to our carelessness during the revision. In accordance with your suggestions, we have conducted a line-by-line check of the entire manuscript, with particular attention paid to all instances of the Mfn1 gene and protein names. We have changed all gene names to italic format to better comply with academic standards. Moreover, we will carefully review the entire manuscript to ensure that the revised content is clear and coherent in terms of language and logic, and to avoid any new errors or ambiguities arising from the format adjustment. Thank you again for your professional guidance.

Comments 3: All Authors should read again the text carefully and correct all mistakes. Take your time.

Response 3: We are extremely grateful for your support of our work and the professional guidance you’ve provided. We also really appreciate your suggestions and reminders. We are fully aware of the issues in the manuscript. We sincerely apologize for the oversights during the manuscript revision process. After completing this round of revisions, all the authors will carefully re-read the manuscript and earnestly address the existing problems. Thank you again for your professional advice and patient guidance.

Title

Comments 4: P1, l2 by Mfn1 gene

Response 4: We are truly grateful to you for your meticulous review of our manuscript and for offering valuable revision suggestions. We sincerely apologize for the oversights during the manuscript revision process. In accordance with your suggestions, we have made revisions to the Mfn1 in the title of the manuscript, changing its format to italic.

Abstract

Comments 5: P1, L16 Mfn1 and Mfn2 gene expression

Response 5: We are extremely grateful to you for reviewing our manuscript and providing valuable revision comments and suggestions. We sincerely apologize for the oversights during the manuscript preparation. We have accepted your suggestions and revised the manuscript accordingly.

Comments 6: L18 Mfn1 expression (or Mfn1 gene expression)

Response 6: Thank you very much for your meticulous guidance on our manuscript and the valuable revision suggestions you’ve put forward. Based on your suggestions, we’ve revised the manuscript and changed the format of Mfn1 in this part to italic.

Comments 7: L19 Mfn1 gene using

Response 7: We sincerely appreciate you taking the time out of your busy schedule to review our manuscript and offer valuable revision suggestions. We have made revisions to the manuscript in accordance with your advice. We believe these modifications will greatly contribute to enhancing the quality of our manuscript.

Comments 8: L25 Mfn1 gene expression

Response 8: Thank you very much for your careful guidance on our manuscript. We have revised the manuscript according to your suggestions.

Introduction

Comments 9: P2, L74 silencing of Mfn1 (Authors agreed that this was not knockdown of Mfn1 what they did in their experiments)

Response 9: We are extremely grateful for your meticulous review of our manuscript and the professional suggestions you put forward. Thank you for pointing out this issue. We agree that what was done in the experiment was not the knockdown operation of Mfn1. Therefore, we have revised the manuscript and changed it to “silencing of Mfn1”.

Comments 10: P2, L75 Mfn1

Response 10: We sincerely appreciate your professional opinions and suggestions. We have accepted your suggestions and revised the manuscript by changing the format of Mfn1 to italic.

Materials and methods

Comments 11: P2, L83 These cells were used as control cells (CON) in further study. (Better define here what is the abbreviation and the name of cells CON)

Response 11: We are extremely grateful for the valuable suggestions you put forward regarding our manuscript. We fully agree with the issue you pointed out about clarifying the abbreviation and name of the CON cells. In accordance with your suggestions, we have revised the manuscript and added an explanation of the abbreviation “CON”, the name of the cells it represents, and their specific definitions in the corresponding part of the manuscript. Once again, thank you for your meticulous review and professional advice.

Comments 12: P2, L86 10 cm

Response 12: Thank you very much for the valuable revision suggestions you provided for our manuscript. We have revised the manuscript according to your advice by adding a space between “10” and “cm”.

Comments 13: P2, L90 constructed

Response 13: Thank you for your support, careful review, and guidance regarding our manuscript. We sincerely apologize for the oversight during the manuscript writing process. We accept your suggestion that the tense here should be in the past tense, and we have revised the manuscript accordingly.

Comments 14: P2, L93 Transfection experiments were performed three times and transfection efficiency was statistically analyzed.

Response 14: We truly appreciate you for putting forward valuable revision suggestions for our manuscript. We’ve adopted your suggestions and made corresponding revisions to the manuscript.

Comments 15: P2, L94 Transfected bovine fetus fibroblasts overexpressing Mfn1 gene (OE-Mfn1) and fibroblasts with silenced Mfn1 gene (shMfn1) were used in further study.

Response 15: We sincerely thank you for your meticulous guidance on our manuscript and for providing valuable suggestions for revision. We have accepted your suggestions and revised the manuscript accordingly.

Comments 16: P3, L103 removed and

Response 16: We truly appreciate your professional guidance and suggestions. We have accepted your advice and revised the manuscript by deleting the “and” in Line 103.

Comments 17: P3, L104 Samples were gently shake on a microplate

Response 17: We sincerely appreciate you for offering valuable revision suggestions for our manuscript. We have revised the manuscript according to your suggestions.

Comments 18: P3, L105 10 min

Response 18: We sincerely appreciate your valuable suggestions for revision. We have accepted your advice and revised the manuscript. A space has been added between “10” and “min”.

Comments 19: P3, L124 Annexin V-FITC detection of cell apoptosis ability

Response 19: Thank you for reviewing our manuscript and providing valuable suggestions for revision. We have accepted your advice and revised the manuscript accordingly. We have modified the content in Line 124 to “Annexin V-FITC detection of cell apoptosis ability”.

Comments 20: P3, L126-134 space between lines is different in this part of the text

Response 20: We are truly grateful for your meticulous guidance on our manuscript and the valuable revision suggestions you’ve offered. In accordance with your suggestions, we’ve checked the corresponding parts of the manuscript. The line spacing in the section from L126 to L134 is set to a minimum of 14 points, which is consistent with the rest of the manuscript. Thank you again for your careful guidance.

Comments 21: P4, L154 ATP content assay (ATP is a chemical compound, not an enzyme, like SOD and CAT) ATP content, SOD activity, CAT activity, NADH content. Authors should know the difference.

Response 21: We are extremely grateful for your meticulous review and professional suggestions. Based on your feedback, we have fully recognized the confusion in our manuscript’s presentation during the writing process. Your point about ATP being a chemical compound, which is fundamentally different from enzymes like SOD and CAT, is extremely crucial. Following your guidance, we have revised the manuscript. We adopted more accurate and clear expressions to separately elaborate on the items we measured and their properties. Once again, we sincerely thank you for your corrections.

Comments 22: P4, L178 4oCP5, L202 Each sample of 1x107 cells

Response 22: We sincerely appreciate your meticulous guidance on our manuscript and the valuable suggestions you’ve provided. We’ve accepted your advice and revised the manuscript accordingly.

Comments 23: P5, L204 30 min

Response 23: Thank you for reviewing and carefully guiding us on our manuscript. We have revised the manuscript according to your suggestions, adding a space between “30” and “min”.

Comments 24: P4, L207 10 min

Response 24: We are extremely grateful for your support of our work and for the valuable revision suggestions you’ve provided. We have revised the manuscript in accordance with your suggestions.

Comments 25: P5, L209 gel concentration of 12% was selected for electrophoresis.

Response 25: Thank you very much for your careful review of our manuscript and for providing valuable revision suggestions. We have accepted your suggestions and revised the manuscript accordingly, changing it to “gel concentration of 12% was selected for electrophoresis”.

Comments 26: P5, L213 remove “mouse”. It is a repetition what was in previous line (Mouse … is a mouse).

Response 26: Thank you for your support for our manuscript and for offering valuable revision suggestions. We have accepted your advice and deleted the word “mouse”.

Comments 27: P5, L216 , raised…….. anti-histone (remove “this antibody”, is too many times)

Response 27: We are extremely grateful to you for putting forward valuable revision suggestions for our manuscript. We have revised the manuscript according to your suggestions.

Comments 28: P5, L217, 219., 221,224 remove: this antibody is

Response 28: We are extremely grateful to you for reviewing our manuscript and providing patient guidance. We fully agree with your suggestions and have revised the manuscript accordingly.

Comments 29: P5, L235 10 min…Than the developing solutions A and B were added

Response 29: Thank you for your meticulous guidance on our manuscript. We fully agree with your suggestions and have revised the manuscript accordingly.

Comments 30: P5, L238 Each experiment was repeated three times

Response 30: We are very grateful for your support of our work. We have accepted your suggestion and revised the manuscript, changing it to “Each experiment was repeated three times”.

Comments 31: P6, L247-254 all sentences are in wrong grammatical form in this part of the text

Response 31: We are extremely grateful to you for providing valuable revision suggestions for our manuscript. We sincerely apologize for the oversights made during the manuscript writing process. We fully agree with the points you’ve raised. In accordance with your suggestions, we have conducted a detailed review and correction of the grammar in this part of the manuscript to make its descriptions more scientific and reasonable. Thank you once again for your meticulous guidance on our manuscript. Your suggestions have significantly improved the quality of our work.

Comments 32: P6, L257 and shMfn1 cell cultures.

Response 32: Thank you for your support regarding our manuscript and for offering valuable revision suggestions. We have accepted your advice and revised the manuscript accordingly, making the descriptions in it clearer and more standardized.

Comments 33: P6, L262 Principal Component Analysis (PCA) ( here for the first time abbreviation was introduced) Do not start the sentence with brackets (..)

Response 33: We are truly grateful to you for pointing out the issues in our manuscript. Based on your suggestions, we have revised the relevant sentence to avoid starting a sentence with parentheses, making the description in our manuscript more in line with the requirements of academic writing. Meanwhile, we will pay more attention to such detailed problems during the revision process to ensure the standardization of the manuscript writing. Thank you again for your careful guidance.

Comments 34: P4, P6, L279 differentially produced metabolites

Response 34: We are extremely grateful to you for reviewing our manuscript and offering valuable revision suggestions. We have revised the manuscript in accordance with your comments.

Results

Comments 35: P6, L293 expression of Mfn1 and the levels of Mfn1

Response 35: We sincerely appreciate your professional comments and suggestions on our manuscript. We have accepted your suggestions and revised the manuscript accordingly.

Comments 36: P7, L296 cells (Figure…)

Response 36: Thank you for your professional guidance. We have accepted your suggestions and revised the manuscript accordingly.

Comments 37: P7, L300 control cells

Response 37: Thank you very much for your patient review and guidance on our manuscript. We have revised the manuscript according to your suggestions, changing “groups” to “cells”.

Comments 38: P7, L305 following genes: … (if genes are listed, in italics) than following genes : is ok

Response 38: Thank you for your guidance on our manuscript and for patiently offering revision suggestions. We have accepted your suggestions and made corresponding revisions to the manuscript based on your advice.

Comments 39: P7, L6 Cdk2

Response 39: Thank you very much for your patient guidance. We have revised the manuscript according to your suggestions and changed the format of Cdk2 to italic.

Comments 40: P8, L319, 321 Mfn1 gene

Response 40: Thank you for your support of our manuscript and for providing valuable revision suggestions. In accordance with your advice, we have revised the manuscript by changing the format of Mfn1 to italic.

Comments 41: P9, L344, Mfn1 gene

Response 41: Thank you very much for your meticulous guidance. According to your suggestions, we have revised the manuscript and changed the format of Mfn1 to italic.

Comments 42: P9, L345 After the silencing of the Mfn1 gene

Response 42: Thank you for your patient guidance and for offering valuable revision suggestions. Based on your advice, we have revised the manuscript. Specifically, we’ve changed it to “After the silencing of the Mfn1 gene”.

Comments 43: P9, L353 CAT activity

Response 43: Thank you for your support of our manuscript and for putting forward valuable revision opinions and suggestions. In accordance with your suggestions, we have revised the manuscript and changed “CAT content” to “CAT activity”.

Comments 44: P10, L354 SOD activity

Response 44: Thank you very much for your patient guidance on our manuscript. We have accepted your suggestions and revised the manuscript accordingly.

Comments 45: P10, L 367 enriched genes involved in cell growth

Response 45: Thank you for your support and dedicated guidance. We fully endorse your suggestions and have made revisions to the manuscript in accordance with them.

Comments 46: P10. L370 enriched in genes involved in cell proliferation

Response 46: Thank you for your support and careful guidance. We fully agree with your suggestions and have revised the manuscript accordingly.

Comments 47: P10, L378 enriched in genes involved in cAMP, ..

Response 47: Thank you for your support and meticulous guidance. We fully agree with your suggestions and have revised the manuscript accordingly.

Comments 48: P11, L389 Metabolome analysis of OE-Mfn1 and ShMfn1 cells (remove sequencing)

Response 48: Thank you very much for your professional guidance. We accept your suggestions and have revised the manuscript accordingly by deleting “sequencing”.

Comments 49: P12, L433 remove “modification”

Response 49: Thank you very much for your professional guidance. We accept your suggestions and have revised the manuscript accordingly, deleting the word “modification”.

Comments 50: P13, L460 Mfn1 gene

Response 50: Thank you for your professional guidance and the suggestions you’ve put forward. We’ve revised the manuscript in line with your advice, changing the format of Mfn1 to italic.

Comments 51: P13, L465 both of them were

Response 51: Thank you for your support and careful guidance. We fully agree with your suggestions and have revised the manuscript accordingly.

Discussion

Comments 52: P15, L491, 492, 493, 494, 495 496 498 Mfn1

Response 52: Thank you for reviewing our manuscript and providing valuable comments and suggestions. We have revised the manuscript according to your suggestions and changed the format of Mfn1 to italic.

Comments 53: P15, L527 SOD activity

Response 53: Thank you for your support and meticulous guidance. We fully agree with your comments and have revised the manuscript accordingly.

Comments 54: P15, L529 silencing of Mfn1

Response 54: Thank you for your support of our work and for putting forward valuable revision comments and suggestions. We have revised the manuscript according to your comments, changing “knockdown” to “silencing”.

Comments 55: P16, L535 differentially produced

Response 55: Thank you for your support of our work and for providing valuable revision comments and suggestions. In accordance with your advice, we have revised the manuscript. Specifically, we changed “expressed” to “produced” to reflect the synthesis of metabolites. Once again, we appreciate your meticulous guidance. These revisions have made our manuscript more scientific and standardized.

4. Response to Comments on the Quality of English Language

Point 1: The English could be improved to more clearly express the research.

Still needs corrections. Some parts in the Matrials and Methods are not in correct grammatical from. Should be part tense. See above.

Response 1: We sincerely appreciate the valuable comments you provided on our manuscript. In light of the issues you raised earlier, we have fully recognized the problems in the English expression and grammar of the manuscript. Based on your suggestions, we have carefully examined and revised the manuscript. Meanwhile, we invited a native English speaker to review the language description of the manuscript to ensure accurate grammar and clear presentation of the research content. Moreover, all the authors of the manuscript have read through it to ensure that all errors have been further corrected. Regarding the grammar issues in the Materials and Methods section, we have carefully checked and modified them, changing the corresponding content to the correct tense. Once again, we are grateful for your professional guidance and patient review. Your comments have significantly improved the quality of our manuscript.

This manuscript is a resubmission of an earlier submission. The following is a list of the peer review reports and author responses from that submission.

Round 1

Reviewer 1 Report

Comments and Suggestions for Authors

The manuscript by Xu et al. is a potentially important study on how mitofusin-1 expression levels affect mitochondrial metabolism and corresponding changes in related epigenetic modifications in cells.  While the data are overall relatively convincing, there are a number of deficiencies that can be addressed as follows.

While information on the expression constructs is given in supplemental figures, this information should also appear in materials and methods.  In particular, description of what is meant by “control” specimens is required.  Are these control cells also transfected with empty or otherwise control expression vectors.  If so what are the control vectors used for appropriate comparison with expression vectors that affect mfn1 levels. It appears from Figure S1 that the controls are not transfected with control vectors but this is not clear.  If not, this needs to be performed for appropriate comparison at least in one relevant validation experiment that could appear in supplemental data.  Also, Figure S1 does not have sufficient resolution to view details of the plasmid expression constructs used in this report.

Data showing that the expression constructs affect Mfn1 levels had the intended structural effects on mitochondrial morphology should be included.  A standard mitochondrial stain with statistical analysis showing elongation or fragmentation of mitochondria with mitofusin-1 overexpression or depletion, respectively, would suffice and should be shown in supplemental data.

In figure 2, please explain why chloramphenicol acetyltransferase (CAT) is expressed in these cells as this is usually a nonmammalian reporter gene . If this is not a mistake, please explain how the expression of CAT occurs and how this is significant to  the study.

Section 3.3, line 287.  Please define “PCA data” and how these are generated in materials and methods.

In section 3.3, text referring to Fgure 3F should be inserted somewhere in the paragraph around line 308.

Please further described the company supplying the metabolite ELISA kits used in these studies with information on the validity of the assay. Measures of alpha-ketoglutarate, acetyl-CoA, and SAM are a main finding of the present report and therefore assays for these molecules are of critical importance. The additional independent approach  using metabolomics in figure 4 should include a focused analysis of alpha-ketoglutarate, acetyl-CoA, and SAM to support the ELISA data (also in Figure 4).    Finally , what do the colored bars in figure 4E represent?  This needs to be added to the figure legend or in a figure key.

In figure 5, the relationships between expression of histone modification genes and metabolite levels is not sufficiently explained.  For instance, it is not explained why increased alpha-ketoglutarate should correlate with expression levels of Tet or any other histone modification enzymes.  Explain whether TET activity and/or TET expression is expected to increase by higher levels of alpha-ketoglutarate.  In the case of the latter, it will be important to add references and text that the expression levels of these enzymes are affected by substrate levels for these data to make sense.  Finally, the westerns in Figure 5I-P show extremely small changes and don’t clearly reflect quantitation of data in the accompanying histograms.  Please specify the n’s of each treatment group in these experiments in the figure legend and in materials and methods.  Also, this type of analysis is usually performed with a Chip or Chip-seq assay instead of analysis of whole cell lysate which should give adequate signals and changes between groups for statistical analysis. 

Typos:

Page 2: oxidative phosphorylation is repeated twice. Delete one.

Line 77: “this study try”…   needs correction.

Author Response

3. Point-by-point response to Comments and Suggestions for Authors

Comments 1: While information on the expression constructs is given in supplemental figures, this information should also appear in materials and methods. In particular, description of what is meant by “control” specimens is required. Are these control cells also transfected with empty or otherwise control expression vectors. If so what are the control vectors used for appropriate comparison with expression vectors that affect mfn1 levels. It appears from Figure S1 that the controls are not transfected with control vectors but this is not clear. If not, this needs to be performed for appropriate comparison at least in one relevant validation experiment that could appear in supplemental data.  Also, Figure S1 does not have sufficient resolution to view details of the plasmid expression constructs used in this report.

Response 1: We are extremely grateful to the reviewers for their comments and suggestions. We accept your comments and will supplement the information about the vectors used in this study in the Materials and Methods section of the manuscript. We will answer the reviewers’ question regarding the meaning of the control samples. The control samples used in this study are cells that have not been transfected with any vectors. The vectors used in this study were obtained from previous experiments, and corresponding experimental verifications have been carried out using these vectors, and relevant articles have been published [1]. Based on this premise, in this study, we selected these two vectors to transfect the cells in order to achieve the purpose of over-expressing and knocking down the Mfn1 gene. Your comments have made us fully aware of our shortcomings.。We sincerely apologize for the unclear plasmid vector diagram in Figure S1. In order to better present the information about the vectors we used, we have redrawn Figure S1 to improve the clarity of the vector images, so as to better reflect our research.

[1] Wu S, Zhao X, Wu M, Yang L, Liu X, Li D, Xu H, Zhao Y, Su X, Wei Z, Bai C, Su G, Li G. Low Expression of Mitofusin 1 Gene Leads to Mitochondrial Dysfunction and Embryonic Genome Activation Failure in Ovine-Bovine Inter-Species Cloned Embryos. Int J Mol Sci. 2022 Sep 4;23(17):10145.

Comments 2: Data showing that the expression constructs affect Mfn1 levels had the intended structural effects on mitochondrial morphology should be included. A standard mitochondrial stain with statistical analysis showing elongation or fragmentation of mitochondria with mitofusin-1 overexpression or depletion, respectively, would suffice and should be shown in supplemental data.

Response 2: We sincerely appreciate that you took the time out of your busy schedule to review our manuscript and provided constructive comments and suggestions. We fully agree with your idea of studying mitochondrial morphological changes by regulating the expression of the Mfn1 gene through vector transfection. Current research has demonstrated that alterations in the expression level of the Mfn1 gene can affect mitochondrial morphology. High expression of the Mfn1 gene in cells can lead to the formation of a characteristic mitochondrial network, thereby influencing mitochondrial morphology. When the expression of the Mfn1 gene in cells is reduced, this abnormal mitochondrial network phenomenon can be improved [1,2]. Based on the above research background, this study did not detect the impact of Mfn1 expression on mitochondrial morphology. Once again, thank you for reviewing this manuscript.

[1] Santel A, Frank S, Gaume B, Herrler M, Youle RJ, Fuller MT. Mitofusin-1 protein is a generally expressed mediator of mitochondrial fusion in mammalian cells. J Cell Sci. 2003 Jul 1;116(Pt 13):2763-74.

[2] Cao YL, Meng S, Chen Y, Feng JX, Gu DD, Yu B, Li YJ, Yang JY, Liao S, Chan DC, Gao S. MFN1 structures reveal nucleotide-triggered dimerization critical for mitochondrial fusion. Nature. 2017 Feb 16;542(7641):372-376.

Comments 3: In figure 2, please explain why chloramphenicol acetyltransferase (CAT) is expressed in these cells as this is usually a nonmammalian reporter gene. If this is not a mistake, please explain how the expression of CAT occurs and how this is significant to the study.

Response 3: We would like to express our sincere gratitude to the reviewers for their valuable comments and suggestions. We will answer the reviewers’ questions. In this study, the CAT we detected is not acetyltransferase. Here, CAT refers to catalase. Catalase is distributed in all tissues of all known animals, especially in the liver. It is an enzyme that catalyzes the decomposition of hydrogen peroxide into oxygen and water and is located in the peroxisomes of cells. While mitochondria produce energy through oxidative phosphorylation, they also generate superoxide and reactive oxygen species (ROS). Since the membrane is impermeable to superoxide, superoxide can be easily dismutated by mitochondrial superoxide dismutase (SOD) into hydrogen peroxide (H₂O₂) and O₂, converting it into non - free radicals, which can cause oxidative damage and ultimately initiate the apoptotic pathway. At the same time, this study found that changes in the expression of the Mfn1 gene can affect the content of ROS in cells. Based on the above reasons, we detected the expression level of CAT.

Comments 4: Section 3.3, line 287.  Please define “PCA data” and how these are generated in materials and methods.

Response 4: We sincerely appreciate the constructive comments and suggestions you provided on our manuscript. We have accepted your suggestions and added and elaborated on the method of generating PCA data in the Materials and Methods section of the manuscript.

Comments 5: In section 3.3, text referring to Fgure 3F should be inserted somewhere in the paragraph around line 308.

Response 5: We sincerely appreciate your valuable comments on our manuscript. First of all, we would like to express our deep apologies for the mistakes made during the manuscript writing process. We accept your suggestions and have revised the manuscript. We have inserted text referring to Fig 3F in the manuscript.

Comments 6: Please further described the company supplying the metabolite ELISA kits used in these studies with information on the validity of the assay. Measures of alpha-ketoglutarate, acetyl-CoA, and SAM are a main finding of the present report and therefore assays for these molecules are of critical importance. The additional independent approach using metabolomics in figure 4 should include a focused analysis of alpha-ketoglutarate, acetyl-CoA, and SAM to support the ELISA data (also in Figure 4). Finally, what do the colored bars in figure 4E represent? This needs to be added to the figure legend or in a figure key.

Response 6: We would like to express our sincere gratitude to the reviewers for their valuable comments and suggestions. In Section 2.11 of the manuscript, we have described the companies and catalog numbers of the ELISA kits used for the detection of alpha-ketoglutarate, acetyl-CoA, and SAM, as well as the detailed methods for detecting metabolites using these kits. We highly agree with the reviewers’ suggestion regarding the detection of the expression levels of α-ketoglutarate, acetyl-CoA, and SAM by metabolomics. We also gave full consideration to this at the initial stage of the experiment, and analyzed these metabolites using metabolomics. However, these metabolites were not enriched. Therefore, we shifted the focus of the metabolomics research to the GO and KEGG functional enrichment analysis of differential metabolites. We also fully agree with your suggestion regarding Figure 4E. According to your advice, we have added an explanation of the meaning represented by the color bar in the legend of Figure 4E.

Comments 7: In figure 5, the relationships between expression of histone modification genes and metabolite levels is not sufficiently explained.  For instance, it is not explained why increased alpha-ketoglutarate should correlate with expression levels of Tet or any other histone modification enzymes.  Explain whether TET activity and/or TET expression is expected to increase by higher levels of alpha-ketoglutarate.  In the case of the latter, it will be important to add references and text that the expression levels of these enzymes are affected by substrate levels for these data to make sense.  Finally, the westerns in Figure 5I-P show extremely small changes and don’t clearly reflect quantitation of data in the accompanying histograms. Please specify the n’s of each treatment group in these experiments in the figure legend and in materials and methods.  Also, this type of analysis is usually performed with a Chip or Chip-seq assay instead of analysis of whole cell lysate which should give adequate signals and changes between groups for statistical analysis.

Response 7: We are extremely grateful to the reviewers for their comments and suggestions. Regarding the issue of metabolites affecting the expression of DNA methylation - modifying enzymes, we have cited relevant literature in lines 403 - 405 of the manuscript. Although the changes shown in the Western blots in Figures 5 I-P are very small, we used ImageJ software to perform grayscale analysis on the protein expression in the Western blots. Each group was repeated three times, and we analyzed the significance of the differences. A P-value < 0.05 was considered to indicate a significant difference. We have supplemented a detailed description of the grayscale analysis method in Section 2.10 of the Materials and Methods section. We also highly agree with the reviewers’ suggestion of using ChIP to detect histone methylation modifications. However, due to funding limitations in this study, we did not use ChIP for histone methylation modification detection but instead used Western blot. Once again, thank you for your comments and suggestions. We have benefited greatly from them, which have been of great help in improving the quality of our manuscript. They have also provided new ideas and directions for our subsequent research.

Comments 8: Page 2: oxidative phosphorylation is repeated twice. Delete one.

Response 8:  We sincerely appreciate the valuable comments and suggestions you provided for our manuscript. Your insights have been of great assistance in enhancing the quality of our work. In accordance with your suggestions, we have made revisions to the manuscript to ensure it better meets the publication requirements of the journal.

Comments 9: Line 77: “this study try”…   needs correction.

Response 9: Thank you very much for the valuable comments and suggestions you provided for our manuscript. We have revised the manuscript according to your advice and changed “try” to “tries”.

4. Response to Comments on the Quality of English Language

Point 1: The English is fine and does not require any improvement.

Response 1: Thank you very much for reviewing our manuscript and providing feedback on its English quality. We are delighted that you have affirmed the English quality of our manuscript. In the subsequent revised version, we will continue to maintain the clarity and accuracy of the manuscript’s language. Once again, we appreciate your support for our work and the valuable comments and suggestions you have put forward.

Reviewer 2 Report

Comments and Suggestions for Authors

The following points should be considered by the authors:

1. Section 2.9: Here the individual steps describing the kit procedure can be omitted.

2. Transfection efficiency was quite different and resulting in very modest changes in protein expression (1.4 fold or 0.8 fold). That should be commented on.

3. Line 52: acetyl CoA is missing

4. Fig. 1 panels A and D: The fluorescence micrographs are not visible (black images).

5. It is not clear how the fluorescence intensity in Figs. 1B and E was obtained.

How many experiments were performed (this information should be added to all legends)?

6. Fig. 2: Why the authors write in the ordinate to panels A, B, E and F 'relative'?

7. Fig.3: Why the KEGG enrichment analysis (no mitochondrial pathways detected) did not reproduce the results of the GO enrichment analysis?

8. Did the metabolomics investigation also show significant changes in alpha-KG and acetyl-CoA?

9. How the authors explain upregulation of Kdm1a in OE-Mfn1 as well as in shMfn1 (Fig. 5C and D)?

Author Response

3. Point-by-point response to Comments and Suggestions for Authors

Comments 1: Section 2.9: Here the individual steps describing the kit procedure can be omitted.

Response 1: We sincerely thank you for your support of our work and for your valuable comments and suggestions. We agree with your opinions, but we believe that the description of the experimental procedure in 2.9 is to better present the content of our research, so we did not delete it.

Comments 2: Transfection efficiency was quite different and resulting in very modest changes in protein expression (1.4 fold or 0.8 fold). That should be commented on.

Response 2: We are extremely grateful to the reviewers for their comments and suggestions on our manuscript. We fully agree with the opinions you put forward. We also recognize that there were significant differences in transfection efficiency, which led to only very minor changes in protein expression. At the mRNA level, we detected relatively large changes in Mfn1 levels, but the changes at the protein expression level were relatively small. However, our analysis of the fluorescence intensity using Image J software showed that there were significant differences in the expression level of Mfn1 protein. This might be because we did not screen the successfully transfected cells. Once again, thank you for your valuable comments, which have enabled us to more clearly recognize the limitations in our research. They also provide valuable guiding suggestions for our future research.

Comments 3: Line 52: acetyl CoA is missing.

Response 3: We would like to express our sincere gratitude to the reviewers for taking the time out of their busy schedules to review our manuscript. We apologize deeply for the mistakes in the manuscript writing. In accordance with your suggestions, we have revised the manuscript to make it more compliant with the publication requirements of the journal.

Comments 4: Fig. 1 panels A and D: The fluorescence micrographs are not visible (black images).

Response 4: We sincerely appreciate the comments and suggestions you provided regarding our manuscript. In accordance with your feedback, we have adjusted the immunofluorescence images in Figs. 1A and 1D to make them clearer.

Comments 5: It is not clear how the fluorescence intensity in Figs 1B and E was obtained. How many experiments were performed (this information should be added to all legends)?

Response 5: We are extremely grateful to the reviewers for their comments and suggestions. First, we will answer the questions you raised. We used ImageJ software to analyze the fluorescence intensities in Figs 1B and E. We respectively selected 5 fields of view to count their relative fluorescence intensities. Each field of view was detected 3 times, and the average value was calculated. Finally, we performed a statistical analysis on the fluorescence intensities of different treatment groups. We highly agree with your suggestion and have added the method of obtaining the fluorescence intensities in the figure legends. Thank you again for your comments, which are of great help in improving the quality of our manuscript.

Comments 6: Fig. 2: Why the author write in the ordinate to panels A, B, E and F ‘relative’?

Response 6: We sincerely appreciate the reviewers for taking the time out of their busy schedules to review our manuscript. We would like to address the questions raised by the reviewers. Regarding the detection of JC-1 content in Fig 2A, we used the JC-1 Detection Kit (C2003S) from Beyotime to measure the JC-1 content in the cells. When the mitochondrial membrane potential is high, red fluorescence is produced, and when the mitochondrial membrane potential is low, green fluorescence is generated. The JC-1 level in the cells is analyzed by the ratio of red to green fluorescence. Therefore, “relative” is added to the y-axis of Fig 2A. For Figures 2B, E, and F, we have modified the y-axes to make them more scientific and reasonable. Moreover, in the Methods section, we have added explanations about the detection methods for SOD and CAT. Once again, we are grateful for your comments and suggestions on our manuscript. Your input has greatly improved the quality of our work.

Comments 7: Fig. 3: Why the KEGG enrichment analysis (no mitochondrial pathways detected) did not reproduce the results of the GO enrichment analysis?

Response 7: We are extremely grateful to you for taking the time out of your busy schedule to review our manuscript and for providing valuable comments and suggestions. We would like to address the questions raised by the reviewers regarding Fig. 3. In this study, we conducted GO and KEGG biological function enrichment analyses on the differentially expressed genes screened after regulating the expression of the Mfn1 gene. Our results are all based on RNA-seq data, and they are real and reliable. The mitochondrial pathway was not enriched. However, both the GO and KEGG functional enrichment analyses of the differentially expressed genes in this study were enriched in functions such as cell growth and apoptosis, as well as energy, glucose and lipid metabolism, indicating that the change in Mfn1 gene expression has an impact on cell proliferation and metabolic capacity, which further validates our previous conclusions. Once again, we thank you for your constructive comments and suggestions on our manuscript.

Comments 8: Did the metabolomics investigation also show significant changes in alpha-KG and acetyl-CoA?

Response 8: We sincerely appreciate your assistance and support for our work. The issue raised by our reviewers was also analyzed during the preliminary stage of our experiment. However, in this study, we did not directly detect the changes in the content of alpha-KG and acetyl-CoA through metabolomics. We focused on analyzing the differential metabolites among different treatment groups and the GO and KEGG pathways they were enriched in by metabolomics, aiming to clarify the changes in metabolism - related pathways due to the regulation of the Mfn1 gene. We used ELISA kits to detect the content of alpha-KG and acetyl-CoA. The results showed that the regulation of Mfn1 gene expression could affect the content of the metabolites alpha-KG and acetyl-CoA. Once again, we thank you for reviewing and supporting our manuscript.

Comments 9: How the authors explain upregulation of Kdm1a in OE-Mfn1 as well as in shMfn1 (Fig. 5C and D)?

Response 9: We are very grateful for the comments and suggestions you provided on this manuscript. Regarding the issue raised by the reviewer, we also gave it full consideration during the manuscript writing process. As is well-known, kdm1a and kdm1b are enzymes that regulate H3K9 methylation modification. In addition, the kmt1, kdm3, and kdm7 families can also regulate the level of H3K9 methylation modification. The increased expression of kdm1a detected in OE-Mfn1 cells may be the main reason for the decrease in the degree of H3K9me3 modification. In the shMfn1 group of cells, the level of H3K9me3 modification was found to increase, but the expression of kdm1a also increased to some extent. This indicates that kdm1a may not be the main cause of the change in the H3K9me3 modification level in the shMfn1 group of cells, and comprehensive multi-aspect detection and analysis should be carried out. However, due to the funding issue of this study, we did not detect the expression levels of other enzymes involved in the regulation of H3K9me3 modification. The issue you raised has also made us aware of the deficiencies in our experimental design. In the subsequent research process, we will conduct in - depth thinking and design. Once again, we thank you for your constructive comments and suggestions, which are of great help in improving the quality of our manuscript and future scientific research.

4. Response to Comments on the Quality of English Language

Point 1: The English is fine and does not require any improvement.

Response 1: We would like to express our sincere gratitude for your thorough review of our manuscript and your high recognition of its English expression. Your positive feedback has been a great source of inspiration for us. Although you have affirmed the language quality of our manuscript, we still attach great importance to the scientific rigor and accuracy of the writing. We will carefully consider all the other comments and suggestions you’ve provided and make revisions accordingly. We guarantee that during the subsequent revision process, we will maintain the readability of the manuscript to ensure it better meets the publication requirements of the journal. Once again, we sincerely thank you for your approval and support of our work.

Round 2

Reviewer 1 Report

Comments and Suggestions for Authors

The authors did not respond in a substantial way to this reviewer's concerns on the data and approaches used in the study.

Author Response

Response to Reviewer 1 Comments

1. Summary

We are extremely grateful to you for your patient guidance and selfless assistance throughout the manuscript submission process. However, we deeply apologize for failing to provide a substantial response to your concerns regarding the data and methods in our manuscript. This was because we failed to fully understand the suggestions you put forward during the first-round review. There was a significant deviation in our understanding of your suggestions, resulting in our failure to precisely address your questions. Undoubtedly, this was an oversight in our work. When responding to the review comments this time, we revisited and thoroughly analyzed your first-round suggestions. With a rigorous and responsible attitude, we reorganized our responses, aiming to present the data and methods in the manuscript more clearly. We strive to ensure that every detail can accurately reflect the content of our research, in the hope of meeting your requirements and expectations. We sincerely hope that this response can make up for our previous deficiencies and gain your approval. Once again, thank you for your continuous support and assistance. We would like to express our gratitude again for your support and recognition of our work.

2. Questions for General Evaluation

Reviewer’s Evaluation

Response and Revisions

Does the introduction provide sufficient background and include all relevant references?

Yes/Can be improved/Must be improved/Not applicable

Thank you very much to the reviewer for the comments and suggestions on our manuscript. We have revised the manuscript according to your feedback.

Are all the cited references relevant to the research?

Yes/Can be improved/Must be improved/Not applicable

Thank you to the reviewer for the comments and suggestions on this manuscript.

Is the research design appropriate?

Yes/Can be improved/Must be improved/Not applicable

Thank you very much to the reviewer for the comments and suggestions on our manuscript. We have revised the manuscript according to your feedback.

Are the methods adequately described?

Yes/Can be improved/Must be improved/Not applicable

Thank you very much to the reviewer for the comments and suggestions on our manuscript. We have revised the manuscript according to your feedback.

Are the results clearly presented?

Yes/Can be improved/Must be improved/Not applicable

Thank you very much to the reviewer for the comments and suggestions on our manuscript. We have revised the manuscript according to your feedback.

Are the conclusions supported by the results?

Yes/Can be improved/Must be improved/Not applicable

Thank you very much to the reviewer for the comments and suggestions on our manuscript. We have revised the manuscript according to your feedback.

3. Point-by-point response to Comments and Suggestions for Authors

Comments 1: The authors did not respond in a substantial way to this reviewer's concerns on the data and approaches used in the study.

Response 1: We would like to express our sincere gratitude for your patient guidance and assistance during the submission process of our manuscript. We deeply regret that, as you pointed out, we failed to provide a substantial response to your concerns regarding the data and methods used in our manuscript. First and foremost, we apologize for not fully understanding the suggestions you provided in the first round of review. There was a misinterpretation on our part of your comments and recommendations, which led to an inaccurate response to your feedback. In this round of responding to the review comments, we have revisited and thoroughly analyzed your first - round feedback. We have crafted a new response with the aim of making the data and methods in the manuscript more clearly reflect the content of our research. Thank you again for your valuable input and support.

You previously put forward Comments 1: While information on the expression constructs is given in supplemental figures, this information should also appear in materials and methods. In particular, description of what is meant by “control” specimens is required. Are these control cells also transfected with empty or otherwise control expression vectors. If so what are the control vectors used for appropriate comparison with expression vectors that affect mfn1 levels. It appears from Figure S1 that the controls are not transfected with control vectors but this is not clear. If not, this needs to be performed for appropriate comparison at least in one relevant validation experiment that could appear in supplemental data.  Also, Figure S1 does not have sufficient resolution to view details of the plasmid expression constructs used in this report.

Response 1: We highly agree with your viewpoints. First, we have supplemented the information about the vectors used in the Materials and Methods section of the manuscript (Line 93-99). We fully concur with your point that if there is no transfected control vector in the control group, at least one relevant validation experiment is needed to prove this. In this study, we used two vectors, namely the Mfn1 overexpression vector and the Mfn1 knockdown vector. Since these two vectors were constructed from different backbone vectors, we directly selected untransfected cells as the control group. The effects of the two vectors in this study have been verified in previous research, and relevant articles using these two vector tools have been published. In the previous articles, we also conducted verification and analysis on the functions of the vectors, and we have cited these previous articles in the Materials and Methods section of this manuscript. Once again, we deeply apologize that our previous response did not meet your expectations. We hope that this response has been improved and can more clearly reflect the results of our research.

Comments 2: Data showing that the expression constructs affect Mfn1 levels had the intended structural effects on mitochondrial morphology should be included. A standard mitochondrial stain with statistical analysis showing elongation or fragmentation of mitochondria with mitofusin-1 overexpression or depletion, respectively, would suffice and should be shown in supplemental data.

Response 2: We sincerely appreciate the valuable comments you provided. Regarding your suggestion to supplement the data on the changes in mitochondrial morphology after Mfn1 overexpression and knockout in this study, we have given it serious consideration. Currently, due to the limitations in our project funds, time, and resources, we are temporarily unable to conduct this experiment. However, we have conducted in - depth research on other relevant characteristics of mitochondria from multiple perspectives. We measured several mitochondrial function indicators, such as mitochondrial membrane potential, ATP levels, and antioxidant capacity, to explore the effects of changes in Mfn1 gene expression on mitochondria. These results indirectly but strongly support our view that the Mfn1 gene can influence mitochondrial function. There has been some progress in the research on the impact of the Mfn1 gene on mitochondrial morphology. High expression of the Mfn1 gene in cells can lead to the formation of mitochondrial network characteristics, thereby affecting mitochondrial morphology. When the expression level of the Mfn1 gene in cells decreases, this abnormal mitochondrial network phenomenon can be improved [1,2]. Although direct mitochondrial morphological staining and statistical analysis data would add more details to the study, we believe that the existing data are sufficient to support the core conclusions of this research. Once again, thank you for your understanding and support.

[1] Santel A, Frank S, Gaume B, Herrler M, Youle RJ, Fuller MT. Mitofusin-1 protein is a generally expressed mediator of mitochondrial fusion in mammalian cells. J Cell Sci. 2003 Jul 1;116(Pt 13):2763-74.

[2] Cao YL, Meng S, Chen Y, Feng JX, Gu DD, Yu B, Li YJ, Yang JY, Liao S, Chan DC, Gao S. MFN1 structures reveal nucleotide-triggered dimerization critical for mitochondrial fusion. Nature. 2017 Feb 16;542(7641):372-376.

Comments 3: In figure 2, please explain why chloramphenicol acetyltransferase (CAT) is expressed in these cells as this is usually a nonmammalian reporter gene. If this is not a mistake, please explain how the expression of CAT occurs and how this is significant to the study.

Response 3: We sincerely appreciate the valuable comments and suggestions you’ve provided. We highly value your feedback and are now addressing your concerns. In this study, the CAT we detected is not acetyltransferase but catalase. Catalase is widely present in all tissues of all known animals and is mainly located in the peroxisomes of cells. Its content is particularly significant in liver tissue. Its main function is to catalyze the decomposition of hydrogen peroxide into oxygen and water. While mitochondria generate energy through the process of oxidative phosphorylation, they inevitably produce superoxides and reactive oxygen species (ROS). Since the mitochondrial membrane is impermeable to superoxides, superoxides can be readily catalyzed by mitochondrial superoxide dismutase (SOD) to form hydrogen peroxide (H₂O₂) and oxygen, thereby converting highly oxidizing free radicals into a relatively stable non - radical form. If these free radicals are not converted in a timely manner, they can cause oxidative damage to cells and may even trigger the cell apoptosis pathway in severe cases. It’s worth noting that an important finding in this study is that changes in the expression level of the Mfn1 gene can affect the intracellular ROS content. Based on these physiological mechanisms and research findings, we believe that detecting the expression level of CAT is of great significance for a deeper understanding of the intracellular redox balance and the function of the Mfn1 gene. Therefore, we carried out relevant detection work. Thank you again for your support and guidance.

Comments 4: Section 3.3, line 287.  Please define “PCA data” and how these are generated in materials and methods.

Response 4: We sincerely appreciate the comments you provided on our manuscript. We accept your suggestions and have supplemented the method for PCA data in the Materials and Methods section of the manuscript (Line 213 - 215).

Comments 5: In section 3.3, text referring to Fgure 3F should be inserted somewhere in the paragraph around line 308.

Response 5: We sincerely appreciate your valuable comments on our manuscript. First of all, we would like to express our deep apologies for the mistakes made during the manuscript writing process. We accept your suggestions and have revised the manuscript. We have inserted text referring to Fig 3F in the manuscript (Line 348).

Comments 6: Please further described the company supplying the metabolite ELISA kits used in these studies with information on the validity of the assay. Measures of alpha-ketoglutarate, acetyl-CoA, and SAM are a main finding of the present report and therefore assays for these molecules are of critical importance. The additional independent approach using metabolomics in figure 4 should include a focused analysis of alpha-ketoglutarate, acetyl-CoA, and SAM to support the ELISA data (also in Figure 4). Finally, what do the colored bars in figure 4E represent? This needs to be added to the figure legend or in a figure key.

Response 6: We would like to express our sincere gratitude to the reviewers for their valuable comments and suggestions. In Section 2.11 of the manuscript (Line 186-204), we have described the companies and catalog numbers of the ELISA kits used for the detection of alpha-ketoglutarate, acetyl-CoA, and SAM, as well as the detailed methods for detecting metabolites using these kits. We highly agree with the reviewers’ suggestion regarding the detection of the expression levels of α-ketoglutarate, acetyl-CoA, and SAM by metabolomics. We also gave full consideration to this at the initial stage of the experiment, and analyzed these metabolites using metabolomics. However, these metabolites were not enriched. Therefore, we shifted the focus of the metabolomics research to the GO and KEGG functional enrichment analysis of differential metabolites. We also fully agree with your suggestion regarding Figure 4E. According to your advice, we have added an explanation of the meaning represented by the color bar in the legend of Figure 4E.

Comments 7: In figure 5, the relationships between expression of histone modification genes and metabolite levels is not sufficiently explained.  For instance, it is not explained why increased alpha-ketoglutarate should correlate with expression levels of Tet or any other histone modification enzymes.  Explain whether TET activity and/or TET expression is expected to increase by higher levels of alpha-ketoglutarate.  In the case of the latter, it will be important to add references and text that the expression levels of these enzymes are affected by substrate levels for these data to make sense.  Finally, the westerns in Figure 5I-P show extremely small changes and don’t clearly reflect quantitation of data in the accompanying histograms. Please specify the n’s of each treatment group in these experiments in the figure legend and in materials and methods.  Also, this type of analysis is usually performed with a Chip or Chip-seq assay instead of analysis of whole cell lysate which should give adequate signals and changes between groups for statistical analysis.

Response 7: We sincerely appreciate the valuable comments and suggestions you have provided. Regarding the issue of insufficient explanation of the relationship between histone modification gene expression and metabolite levels that you pointed out, we have supplemented relevant references in the main text of the manuscript to illustrate that an increase in α-KG levels will enhance the activity and expression of TET enzymes. This is because α-KG is the substrate of TET enzymes, and an increase in substrate levels generally promotes enzyme activity and expression (Line 410-413). Although the differences in the Western blots in Figure 5I-P are not visually significant, we used Image J software to perform grayscale analysis on the protein expression levels in the Western blots. Each group was repeated three times to analyze the significance of the differences, and a P-value < 0.05 was considered statistically significant. We have provided a detailed description of the method of using Image J software for grayscale analysis in Section 2.10 of the Materials and Methods section of the manuscript (Line 179 - 180). We also highly agree with your suggestion to detect histone methylation modifications using CHIP or CHIP - seq. We fully understand that these two techniques have significant advantages in such analyses and can provide more detailed and accurate data. We have carefully and fully considered your proposal. However, due to the limitations of our research funds, we are currently unable to use CHIP to detect the modification levels of histone methylation. After comprehensively considering various factors, we finally chose Western blot technology to detect histone modification levels. This technology also has good effects in detecting histone modification levels and can provide valuable reference data for our research. We are well aware of its possible limitations, but under the current conditions, this is the most appropriate choice after careful consideration. Thank you again for your professional suggestions. We will continue to pay attention to the technological development in this field and will consider using more advanced detection methods in future research if possible.

Comments 8: Page 2: oxidative phosphorylation is repeated twice. Delete one.

Response 8:  We sincerely appreciate the valuable comments and suggestions you provided for our manuscript. Your insights have been of great assistance in enhancing the quality of our work. In accordance with your suggestions, we have made revisions to the manuscript to ensure it better meets the publication requirements of the journal.

Comments 9: Line 77: “this study try”…   needs correction.

Response 9: Thank you very much for the valuable comments and suggestions you provided for our manuscript. We have revised the manuscript according to your advice and changed “try” to “tries”.’

4. Response to Comments on the Quality of English Language

Point 1: The English is fine and does not require any improvement.

Response 1: Thank you very much for reviewing our manuscript and providing feedback on its English quality. We are delighted that you have affirmed the English quality of our manuscript. In the subsequent revised version, we will continue to maintain the clarity and accuracy of the manuscript’s language. Once again, we appreciate your support for our work and the valuable comments and suggestions you have put forward.

Reviewer 2 Report

Comments and Suggestions for Authors

The authors have addressed in their reply most of my comments. However, very little of this was leading to a revision of the manuscript text. In particular, the numbers of performed experiments has not been added to the legends of figures.

Author Response

Response to Reviewer 2 Comments

1. Summary

We sincerely appreciate your patient guidance and valuable suggestions during the manuscript review process. We carefully considered your comments and made thorough revisions to the manuscript. Specifically, we added the number of experimental samples and the number of experimental repetitions to the figure captions. The comments and suggestions you provided have greatly helped us improve the quality of the manuscript, making it more suitable for publication in the Journal of Cells. Once again, we are grateful for your support and recognition of our work.

2. Questions for General Evaluation

Reviewer’s Evaluation

Response and Revisions

Does the introduction provide sufficient background and include all relevant references?

Yes/Can be improved/Must be improved/Not applicable

Thank you very much to the reviewer for the comments and suggestions on our manuscript. We have revised the manuscript according to your feedback.

Are all the cited references relevant to the research?

Yes/Can be improved/Must be improved/Not applicable

Thank you to the reviewer for the comments and suggestions on this manuscript.

Is the research design appropriate?

Yes/Can be improved/Must be improved/Not applicable

Thank you very much to the reviewer for the comments and suggestions on our manuscript. We have revised the manuscript according to your feedback.

Are the methods adequately described?

Yes/Can be improved/Must be improved/Not applicable

Thank you to the reviewer for the comments and suggestions on this manuscript.

Are the results clearly presented?

Yes/Can be improved/Must be improved/Not applicable

Thank you to the reviewer for the comments and suggestions on this manuscript.

Are the conclusions supported by the results?

Yes/Can be improved/Must be improved/Not applicable

Thank you very much to the reviewer for the comments and suggestions on our manuscript. We have revised the manuscript according to your feedback.

3. Point-by-point response to Comments and Suggestions for Authors

Comments 1: The authors have addressed in their reply most of my comments. However, very little of this was leading to a revision of the manuscript text. In particular, the numbers of performed experiments has not been added to the legends of figures.

Response 1: We sincerely appreciate your second review of our manuscript and for pointing out the issues. Regarding the problem you mentioned about the lack of experiment repetition times in the figure captions, we immediately made corrections. We have added the repetition times of each experiment to all figure captions to ensure the integrity and transparency of the data. The revised captions for Figure 1 can be found in Lines 271-274, Lines 277-279, and Lines 291-292. For Figure 2, the revised caption is in Lines 321-322. The revised caption for Figure 3 is in Lines 355-356. For Figure 4, it is in Lines 400-403. And for Figure 5, the revised caption is in Lines 463-465. Meanwhile, we have conducted a comprehensive check of the manuscript. In accordance with the suggestions you previously put forward, we have further improved the content of the manuscript. We have incorporated more of the content elaborated in our response into the main text to enhance the quality and readability of the manuscript. Once again, we are grateful for your meticulous and professional review. We will continuously improve our research and the manuscript based on your suggestions.

4. Response to Comments on the Quality of English Language

Point 1: The English is fine and does not require any improvement.

Response 1: We would like to express our sincere gratitude for your thorough review of our manuscript and your high recognition of its English expression. Your positive feedback has been a great source of inspiration for us. Although you have affirmed the language quality of our manuscript, we still attach great importance to the scientific rigor and accuracy of the writing. We will carefully consider all the other comments and suggestions you’ve provided and make revisions accordingly. We guarantee that during the subsequent revision process, we will maintain the readability of the manuscript to ensure it better meets the publication requirements of the journal. Once again, we sincerely thank you for your approval and support of our work.